# DNA damage-induced PARP1 activation confers cardiomyocyte dysfunction through NAD$^+$ depletion in experimental atrial fibrillation

Deli Zhang[1], Xu Hu[1], Jin Li[1], Jia Liu[2], Luciënne Baks-te Bulte[1], Marit Wiersma[1], Noor-ul-Ann Malik[1], Denise M.S. van Marion[1], Marziyeh Tolouee[3], Femke Hoogstra-Berends[3], Eva A.H. Lanters[4], Arie M. van Roon[5], Antoine A.F. de Vries[2], Daniël A. Pijnappels[2], Natasja M.S. de Groot[4], Robert H. Henning[3] & Bianca J.J.M. Brundel [1]

Atrial fibrillation (AF) is the most common clinical tachyarrhythmia with a strong tendency to progress in time. AF progression is driven by derailment of protein homeostasis, which ultimately causes contractile dysfunction of the atria. Here we report that tachypacing-induced functional loss of atrial cardiomyocytes is precipitated by excessive poly(ADP)-ribose polymerase 1 (PARP1) activation in response to oxidative DNA damage. PARP1-mediated synthesis of ADP-ribose chains in turn depletes nicotinamide adenine dinucleotide (NAD$^+$), induces further DNA damage and contractile dysfunction. Accordingly, NAD$^+$ replenishment or PARP1 depletion precludes functional loss. Moreover, inhibition of PARP1 protects against tachypacing-induced NAD$^+$ depletion, oxidative stress, DNA damage and contractile dysfunction in atrial cardiomyocytes and *Drosophila*. Consistently, cardiomyocytes of persistent AF patients show significant DNA damage, which correlates with PARP1 activity. The findings uncover a mechanism by which tachypacing impairs cardiomyocyte function and implicates PARP1 as a possible therapeutic target that may preserve cardiomyocyte function in clinical AF.

[1] Department of Physiology, Amsterdam UMC, Vrije Universiteit Amsterdam, Amsterdam Cardiovascular Sciences, 1081 HZ Amsterdam, The Netherlands. [2] Department of Cardiology, Laboratory of Experimental Cardiology, Leiden University Medical Center, 2300 RC Leiden, The Netherlands. [3] Department of Clinical Pharmacy and Pharmacology, University Medical Centre Groningen, University of Groningen, 9700 RB Groningen, The Netherlands. [4] Department of Cardiology, Erasmus Medical Center, 3015 GD Rotterdam, The Netherlands. [5] Department of Internal Medicine, Division of Vascular Medicine, University of Groningen, University Medical Center Groningen, 9700 RB Groningen, The Netherlands. Correspondence and requests for materials should be addressed to D.Z. (email: d.zhang@vumc.nl) or to B.J.J.M.B. (email: b.brundel@vumc.nl)

Atrial fibrillation (AF) is the most common clinical tachyarrhythmia. Over the past years, considerable progress has been made in unraveling mechanisms driving the initiation and perpetuation of AF, providing targets to ground novel therapeutic options in AF. The current insight is that progression of AF is driven by the high-activation rate of atrial cardiomyocytes, inducing their electrical, structural, and functional remodeling, which renders them increasingly permissive to the arrhythmia[1]. Principle factors governing the cardiomyocyte remodeling, include derailments of $Ca^{2+}$ homeostasis, proteostasis, and the protein quality control system[1–7]. We recently disclosed a prominent role of histone deacetylase 6 (HDAC6)-driven deacetylation of cytoskeletal α-tubulin in structural and functional remodeling of AF cardiomyocytes[8]. In the course of this study, we also observed that nicotinamide (vitamin B₃), an HDAC class III (sirtuins) inhibitor[9,10], offers complete protection against cardiomyocyte remodeling in tachypaced cardiomyocytes and *Drosophila* prepupae, by a mechanism unrelated to its inhibition of sirtuins[8]. Thus, we set out to disclose nicotinamide's mode of action.

In addition to sirtuins, nicotinamide is a known inhibitor of poly(ADP-ribose) polymerases (PARPs)[11,12]. PARPs constitute a family of six nuclear enzymes whose activation is precipitated by single- and double-stranded DNA breaks (SSBs and DSBs, respectively), serving to recruit the DNA repair machinery by synthesis of poly(ADP-ribose) chains (PAR)[13]. During the synthesis of PAR chains, nicotinamide adenine dinucleotide ($NAD^+$) is consumed by PARP up to an extent that it depletes cellular $NAD^+$, leading to a progressive decline in ATP levels, energy loss and cell death in case of excessive PARP activation[14]. Moreover, PARP activation, especially of PARP1, was previously found to be involved in various cardiovascular diseases other than AF[11,12,15–17], and both pharmacological and genetic inhibition of PARP1 provides significant benefits in animal models of such cardiovascular disorders[12,18].

In the current study, we investigate the origin and consequences of PARP activation in experimental AF by characterizing the pathways involved and examine the therapeutic effects of PARP inhibitors and $NAD^+$ replenishment. The findings reveal that tachypacing (TP)-induced cardiomyocyte dysfunction is a consequence of DNA damage-modulated PARP1 activation, which leads to depletion of nicotinamide adenine dinucleotide ($NAD^+$) and further oxidative stress and DNA damage and implicate PARP1 as a possible therapeutic target that may preserve cardiomyocyte function in AF.

## Results

**TP causes DNA damage, PARP activation, and $NAD^+$ loss**. Previously, we observed nicotinamide to protect against contractile dysfunction in tachypaced HL-1 cardiomyocytes and *Drosophila* prepupae, independent of its ability to inhibit sirtuin activity[8]. As nicotinamide is also known to inhibit the activation of PARP[11,12], we tested the level of PARP activity by measuring the amount of PAR synthesis in normal and tachypaced cardiomyocytes. A gradual increase in PAR levels was observed upon TP, which reached significance after 8 h of TP and remained increased afterward (Fig. 1a–d, Supplementary Figure 1a), while PARP1 protein expression was unchanged during TP (Fig. 1a, Supplementary Figure 1b, c). This observation indicates that TP induces PAR synthesis, suggesting induction of PARP activation. Since PARP gets activated by SSB and DSB in the DNA[19], the level of DNA damage was determined by comet assay (single-cell gel electrophoresis)[20], and by measurement of phosphorylation of the Ser-139 residue of the histone variant H2AX, forming γH2AX. Four hours

of TP significantly increased both the amount of DNA in the comet tail (Fig. 1e, f) and γH2AX levels of cardiomyocytes (Fig. 1g–j).

Upon activation, PARP consumes $NAD^+$ to synthesize PAR. Therefore, progressive and excessive activation of PARP results in reductions in $NAD^+$ levels, which finally results in the energy loss and functional impairment of cardiomyocytes[12]. To study whether TP-induced PARP activation depleted $NAD^+$ levels in HL-1 cardiomyocytes, $NAD^+$ levels were measured in HL-1 cardiomyocytes in the course of TP. Eight hours of TP induced a significant reduction in $NAD^+$ levels (Fig. 1k). Normal pacing at 1 Hz did not reveal changes at PAR, γH2AX, or $NAD^+$ levels (Supplementary Figure 2a–e). Together, these findings reveal that TP induces substantial DNA damage and consequently the activation of PARP, resulting in depletion of the cellular content of $NAD^+$ in cardiomyocytes.

**PARP1 is a key enzyme instigating contractile dysfunction**. Since $NAD^+$ is an important constituent for proper cell function and health[21], we next investigated whether the decline in $NAD^+$ levels is a driving mechanism for functional loss by testing the effect of replenishment of $NAD^+$ on contractile function in tachypaced HL-1 cardiomyocytes. TP resulted in a significant $Ca^{2+}$ transients (CaT) loss, which was dose-dependently abrogated by preserving cellular $NAD^+$ levels through exogenous supplementation (Fig. 2a, b, Supplementary Figure 3a, b). This observation was confirmed in tachypaced *Drosophila* prepupae[5], where TP resulted in loss of heart wall contractions and an increase of arrhythmia incidence, which was dose-dependently prevented by replenishment of $NAD^+$ (Fig. 3c–e). Next, we examined whether PARP mediates the $NAD^+$ depletion, since particularly PARP1 and to a lesser extent PARP2 isoforms consume $NAD^+$[13]. Hereto, HL-1 cardiomyocytes were transfected with siRNA targeting PARP1 or PARP2, resulting in specific and effective suppression of their expression in the cardiomyocytes (Supplementary Figure 4a, b). Subsequent TP of siRNA treated cardiomyocytes demonstrated that downregulation of PARP1 significantly protected cardiomyocytes against CaT loss, whereas downregulation of PARP2 did not (Fig. 3a, b).

To confirm that PARP1 is the key PARP enzyme driving TP-induced contractile dysfunction, PARP1 expression was suppressed specifically in the heart of *Drosophila* in two RNAi lines, as confirmed by Western blotting (Supplementary Figure 4c, d). In line with the findings in HL-1 cardiomyocytes, suppression of PARP1 resulted in protection against TP-induced heart wall dysfunction (Fig. 3c–e, Supplementary Figure 4e and Supplementary Figure 5).

These results demonstrate that PARP1 is the key PARP enzyme instigating TP-induced contractile dysfunction in cardiomyocytes.

**PARP1 inhibition prevents $NAD^+$ depletion and functional loss**. To further substantiate that PARP1 represents a drug target to mitigate TP-induced functional remodeling, the action of PARP1 inhibitors was examined in HL-1 cardiomyocytes. PARP1 inhibitors comprised the general inhibitors, nicotinamide and 3-AB, and the specific PARP1/2 inhibitors ABT-888 and olaparib. Both general and specific inhibition of PARP1/2 precluded TP-induced PARylation of proteins and decrease in $NAD^+$ levels (Fig. 4a, b, and Supplementary Figure 6). Furthermore, the PARP1 inhibitors ABT-888 and olaparib also significantly attenuated TP-induced contractile dysfunction in HL-1 cardiomyocytes and *Drosophila* without influencing the baseline contractile function in cardiomyocytes (Fig. 4c–i, Supplementary Figure 3 and Supplementary Figure 7), as previously observed for nicotinamide[8]. In addition, TP of HL-1 cardiomyocytes resulted in significant electrophysiological deteriorations, including

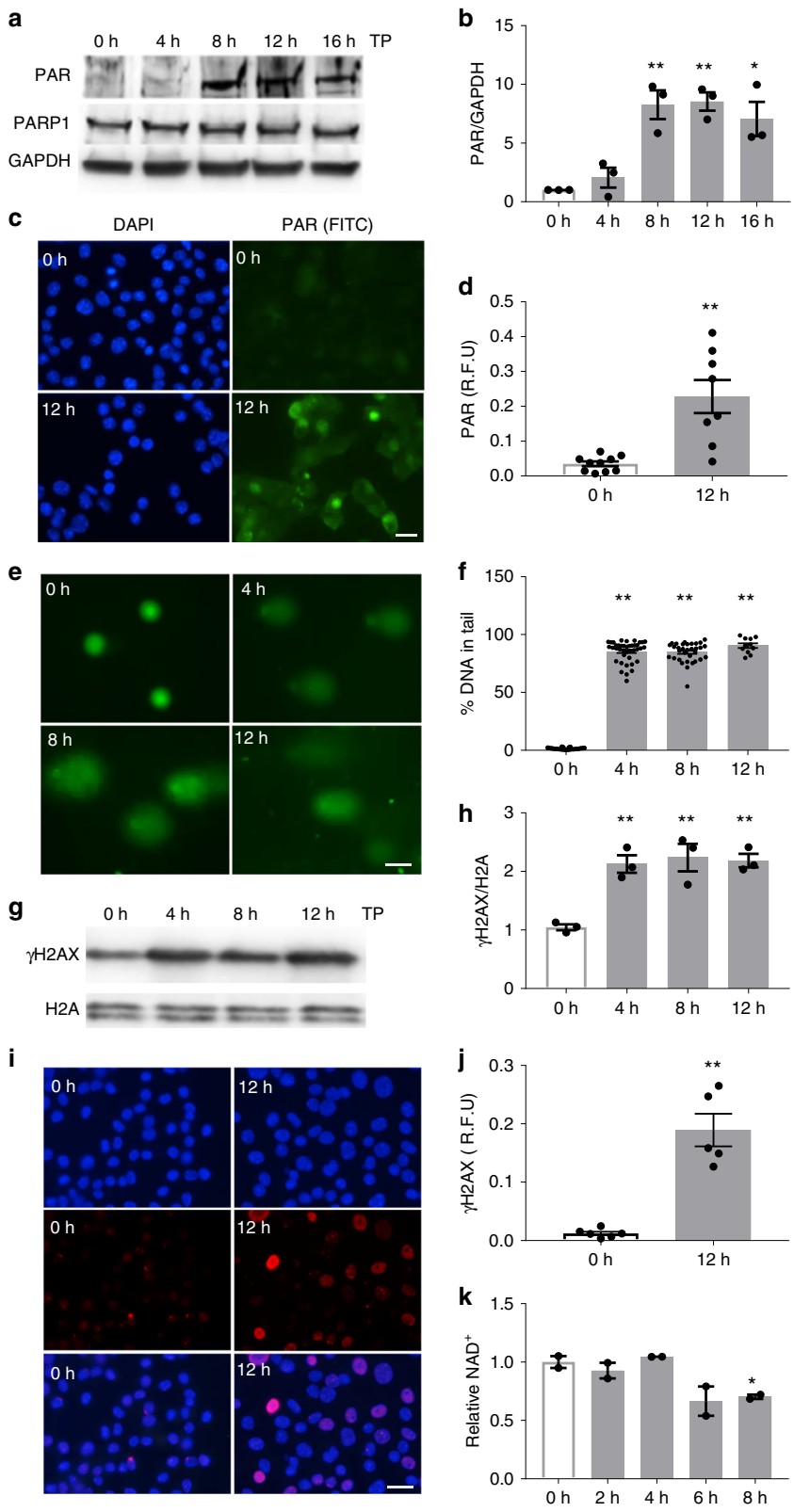

alterations in action potential duration (APD), increased APD dispersions, decreased area of excitability and ion channel remodeling. All TP-induced electrophysiological alterations were prevented by PARP1 inhibitors olaparib and/or ABT-888 (Fig. 5, Supplementary Methods and Supplementary Figure 11). Since AF is a progressive disease, it is of interest to study whether PARP1

inhibition accelerates recovery from TP-induced $NAD^+$ depletion and contractile dysfunction. Hereto, HL-1 cardiomyocytes were tachypaced, followed by 24 h recovery under no pacing conditions. In vehicle treated cardiomyocytes, no recovery from TP induced CaT loss, $NAD^+$ depletion or increased PAR levels was observed. In contrast, tachypaced HL-1 cardiomyocytes post-

**Fig. 1** Tachypacing induces PARP activation, DNA damage, and $NAD^+$ depletion in HL-1 cardiomyocytes. **a** Representative Western blot of PAR and PARP1 levels in control nonpaced (0 h) and tachypaced (TP) HL-1 cardiomyocytes for durations as indicated. **b** Quantified data of PAR expression levels from three independent experiments. $^*P < 0.05$ vs. 0 h, $^{**}P < 0.01$ vs. 0 h. **c**, **d** Immunofluorescent staining and quantified data of PAR levels in control (0 h), and in 12 h TP of HL-1 cardiomyocytes. $^{**}P < 0.01$ vs. 0 h, $n = 10$ images for 0 h, $n = 8$ images for 12 h from over 200 cardiomyocytes. **e** Representative immunofluorescence images of HL-1 cardiomyocytes with time-course TP (0–12 h), showing tail DNA. **f** Quantified percentage of tail DNA in HL-1 cardiomyocytes $^{**}P < 0.01$ vs. 0 h, $n = 49$ cardiomyocytes for 0 h, $n = 40$ for 4 h, $n = 33$ for 8 h, $n = 11$ for 12 h. **g**, **h** Representative Western blot of γH2AX, H2A, and quantified data of γH2AX during time-course of TP in HL-1 cardiomyocytes. $^{**}P < 0.01$ vs. 0 h, $n = 3$ independent experiments. **i**, **j** Representative immunofluorescent staining and quantified data of γH2AX levels in NP (0 h) and TP (12 h) HL-1 cardiomyocytes. $^{**}P < 0.01$ vs. 0 h, $n = 7$ images for 0 h, $n = 6$ images for 12 h from over 200 cardiomyocytes. **k** Relative $NAD^+$ levels in HL-1 cardiomyocytes during time-course of TP (2–8 h) compared to control (0 h). $^*P < 0.05$ vs. 0 h. $n = 2$ independent experiments. Scalebar is 15 μm for **c**, **e** and **i**. Data are all expressed as mean ± s.e.m. Individual group mean differences were evaluated with the two-tailed Student's $t$ test

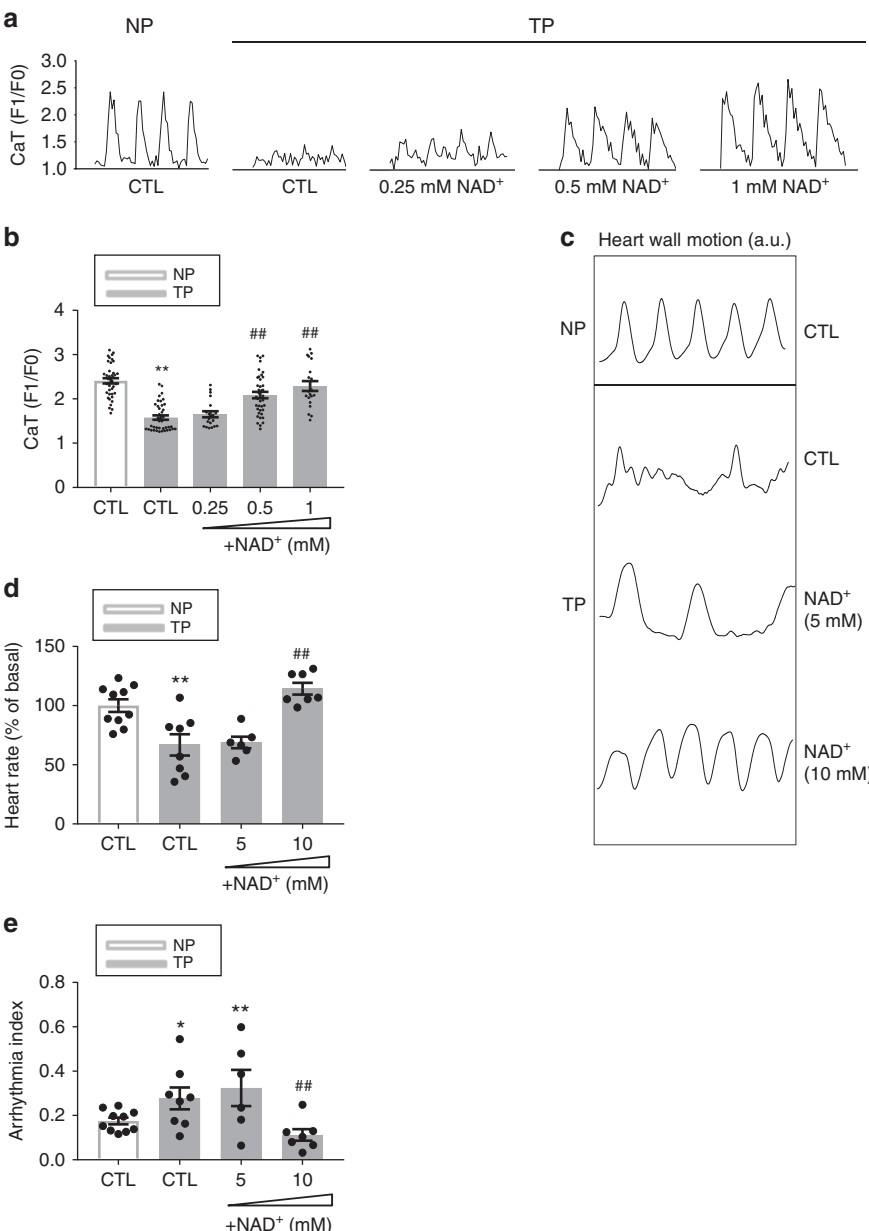

**Fig. 2** Repletion of $NAD^+$ dose-dependently attenuates contractile dysfunction in HL-1 cardiomyocytes and *Drosophila*. **a**, **b** Representative CaT traces and quantified CaT amplitude data of control non-paced (NP) and tachypaced (TP) HL-1 cardiomyocytes pretreated with or without different doses of $NAD^+$ (0.25, 0.5, 1 mM). $^{**}P < 0.01$ vs. Control (CTL) NP $^{##}P < 0.01$ vs. CTL TP, $n = 40$ cardiomyocytes CTL NP, $n = 40$ for CTL TP, $n = 20$ for $NAD^+$ (0.25 mM) TP, $n = 40$ for $NAD^+$ (0.5 mM) TP, and $n = 20$ for $NAD^+$ (1 mM) TP. **c** Representative heart wall motions (during 3.3 s). **d**, **e** Quantified data of relative heart rate and arrhythmicity index to control NP *Drosophila*. *Drosophila* were treated with or without $NAD^+$ (5 or 10 mM). $^*P < 0.05$, $^{**}P < 0.01$ vs. CTL NP $^{##}P < 0.01$ vs. CTL TP. $n = 10$ *Drosophila* prepupae for CTL NP, $n = 8$ for CTL TP, $n = 6$ for $NAD^+$ (5 mM) TP, $n = 7$ for $NAD^+$ (10 mM). Data are all expressed as mean ± s.e.m. Individual group mean differences were evaluated with the two-tailed Student's $t$ test

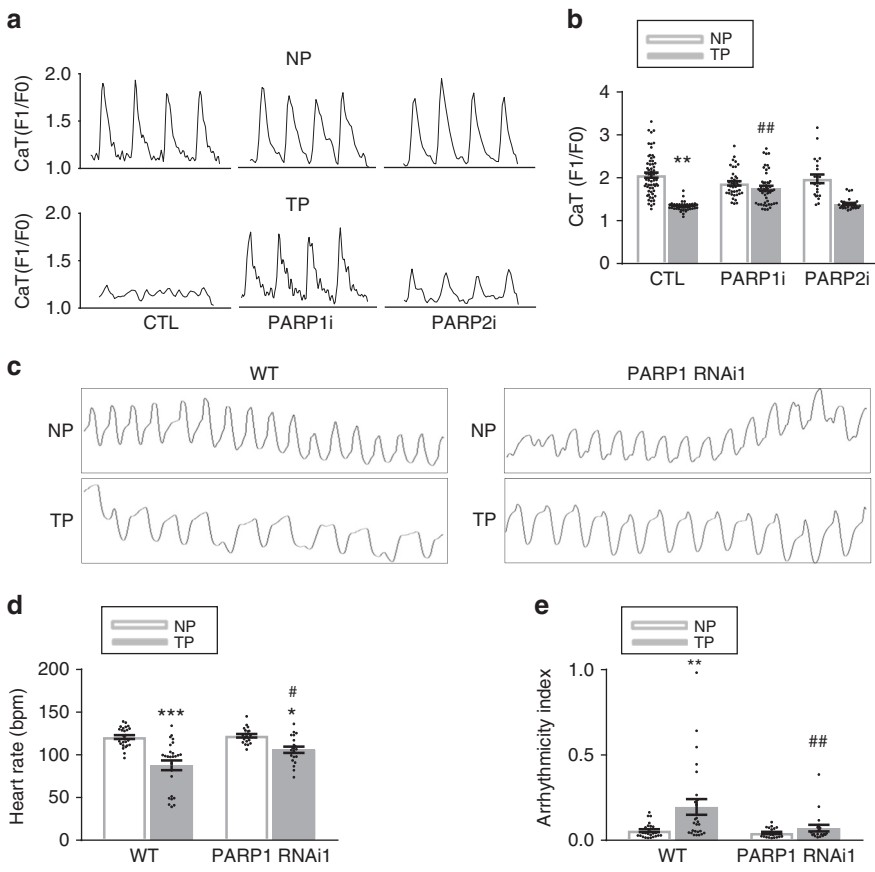

**Fig. 3** PARP1, not PARP2, is the key enzyme mediating tachypacing-induced contractile dysfunction in HL-1 cardiomyocytes and *Drosophila*.
**a**, **b** Representative CaT traces and quantified CaT amplitude data in control nonpaced (NP) or tachypaced (TP) HL-1 cardiomyocytes transfected with scrambled siRNA (CTL), PARP1 siRNA (PARP1i), and PARP2 si RNA (PARP2i). ** $P < 0.01$ vs. CTL NP, ## $P < 0.01$ vs. CTL TP. $n = 62$ cardiomyocytes for CTL NP, $n = 39$ for PARP1i NP, $n = 22$ for PARP2i NP, $n = 56$ for CTL TP, $n = 47$ for PARP1i TP, $n = 27$ for PARP2i TP. **c** Representative traces (10 s) prepared from high-speed movies of *Drosophila* prepupae. Movies were made from nonpaced (NP) and tachypaced (TP) *Drosophila* prepupae in wild-type (WT) and PARP1 knockdown (PARP1 RNAi1) strains. **d**, **e** Quantified heart rate (bpm: beats per minute) and arrhythmicity index in milliseconds (ms). Arrhythmicity index was defined as the standard deviation of the heart periodicity. * $P < 0.05$, ** $P < 0.01$, *** $P < 0.001$ vs. WT NP, # $P < 0.05$ vs. WT TP, $n = 26$ *Drosophila* prepupae for WT, $n = 20$ *Drosophila* prepupae for PARP1i. Data are all expressed as mean ± s.e.m. Individual group mean differences were evaluated with the two-tailed Student's *t* test

treated with ABT-888 revealed accelerated recovery at all end-points (Supplementary Figure 8). These findings demonstrate that PARP1 inhibitors not only prevent PARP1 activation, $NAD^+$ depletion, CaT loss, and electrophysiological and ion channel deteriorations, but also accelerate recovery after cessation of TP.

In line with the findings in tachypaced HL-1 cardiomyocytes, TP of isolated adult rat atrial cardiomyocytes significantly induced DNA damage and PAR levels, reduced $NAD^+$ levels and resulted in contractile dysfunction (Fig. 6 and Supplementary Figure 2f–i). Importantly, all these effects were prevented by the PARP1 inhibitors ABT-888 and olaparib (Fig. 6 and Supplementary Figure 7a, b).

**PARP1 inhibition prevents oxidative stress-induced DNA damage.** Since $NAD^+$ depletion is associated with the induction of oxidative stress[22], which may in turn leads to (further) DNA damage, we tested whether PARP1 inhibition protects by reducing oxidative stress-induced DNA damage[23]. TP of HL-1 cardiomyocytes resulted in significant induction of oxidative damage to proteins (Fig. 7a, b, Supplementary Figure 9) and DNA (Fig. 7c, d), as evidenced by formation of 8-oxoguanine, a biomarker for oxidative DNA damage[24]. Inhibition of PARP1 by ABT-888 prevented TP-induced oxidative protein and DNA

damage (Fig. 7a–d). In addition, the TP-induced γH2AX levels were partly reduced by ABT-888 treatment (Fig. 7e, f). Together, these data indicate that PARP1 inhibition precludes the initiation of a vicious circle in which advanced PARP1 activation is driven by depletion of $NAD^+$, causing further DNA damage.

**DNA damage-mediated PARP activation is the cause of $NAD^+$ depletion.** To study whether PARP activation is the cause of $NAD^+$ depletion and contractile dysfunction in cardiomyocytes, cardiomyocytes were gamma-irradiated to induce DNA damage and thereby PARP activation. As expected, irradiation resulted in a significant induction of DNA damage and consequently an increase in PAR levels, reduction in $NAD^+$ levels, and finally loss in CaT in both HL-1 and rat atrial cardiomyocytes (Figs. 8 and 9). The PARP1 inhibitor ABT-888 prevented the increase in PAR levels, $NAD^+$ depletion and CaT loss (Figs. 8 and 9). These findings confirm that DNA damage-mediated PARP activation is the cause of $NAD^+$ depletion and CaT remodeling in atrial cardiomyocytes.

**PARP1 is activated in human AF and correlates with DNA damage.** To extend our findings to clinical AF, we measured DNA damage and PARP1 activation in right and/or left atrial

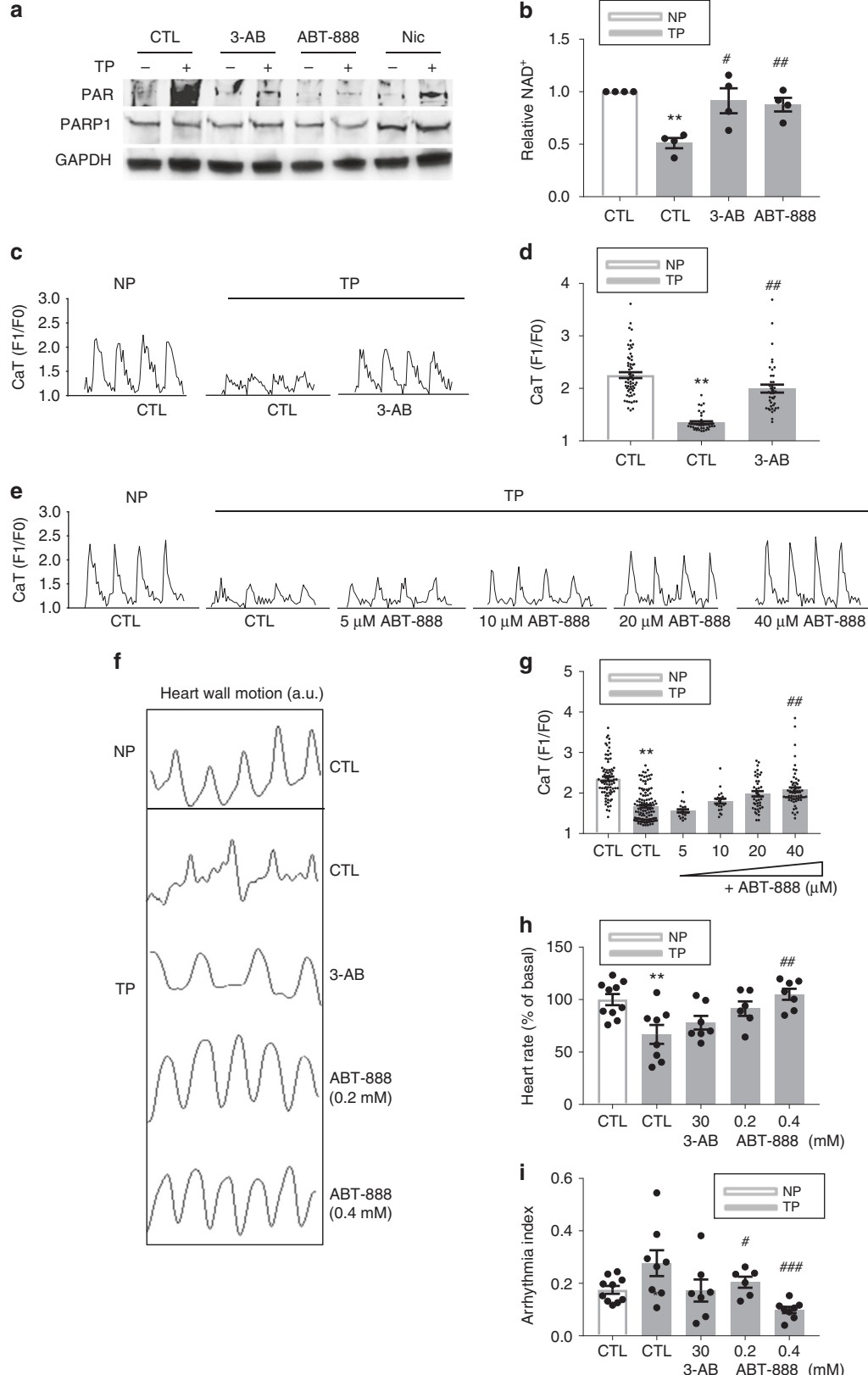

samples (RAA and/or LAA) of (longstanding) persistent AF patients and controls in sinus rhythm (SR). Compared to SR, AF patients demonstrate a significant increase in PAR formation in both RAA and LAA, while both groups show similar PARP1 protein expression (Fig. 10a–c). Furthermore, γH2AX levels were

significantly increased in patients with AF compared to SR (Fig. 10d, e). Moreover, a significant positive correlation was found between the amount of PAR and γH2AX (Fig. 10f, Supplementary Figure 10), indicating that AF patients with high levels of PAR also reveal more DNA damage. In addition, the

**Fig. 4** PARP1 inhibitors dose-dependently protect against contractile dysfunction in HL-1 cardiomyocytes and *Drosophila*. **a** Representative Western blot showing that the PARP inhibitors 3-AB (3 mM), ABT-888 (40 μM), and nicotinamide (Nic, 10 mM) inhibit tachypacing (TP)-induced PAR formation (PARylation), which is an indicator of PARP activity. **b** 3-AB (3 mM) and ABT-888 (40 μM) conserved $NAD^+$ levels after TP. The average value of four independent experiments is shown. **$P < 0.01$ vs. control (CTL) NP, #$P < 0.05$ vs. CTL TP, ##$P < 0.01$ vs. CTL TP. **c, d** Representative CaT traces and quantified CaT amplitude in control non-paced (NP) or tachypaced (TP) HL-1 cardiomyocytes pretreated with 3-AB (3 mM) or vehicle (CTL). **$P < 0.01$ vs. CTL NP, ##$P < 0.01$ vs. CTL TP, $n = 60$ cardiomyocytes for CTL NP, $n = 40$ for CTL TP, $n = 40$ for 3-AB TP. **e, f** Representative CaT and quantified CaT amplitude of nonpaced (NP) and tachypaced (TP) HL-1 cardiomyocytes pretreated with ABT-888 at different doses (5–40 μM) or vehicle DMSO (CTL). **$P < 0.01$ vs. CTL NP, ##$P < 0.01$ vs. CTL TP, $n = 80$ HL-1 cardiomyocytes for CTL NP, $n = 119$ for CTL TP, $n = 20$ for 5 μM ABT-888 TP, $n = 20$ for 10 μM ABT-888 TP, $n = 40$ for 40 μM ABT-888. **g–i** Representative heart wall contraction measurements and quantified relative heart rate and arrhythmicity index of control NP or TP *Drosophila* pretreated with 3-AB (30 mM), ABT-888 (0.2 mM, 0.4 mM), or vehicle (CTL). *$P < 0.05$, **$P < 0.01$ vs. CTL NP, #$P < 0.05$, ###$P < 0.001$ vs. CTL TP, $n = 10$ *Drosophila* prepupae for CTL, $n = 7$ for 3-AB, $n = 6$ for ABT-888 (0.2 mM), $n = 7$ for ABT-888 (0.4 mM). Data are all expressed as mean ± s.e.m. Individual group mean differences were evaluated with the two-tailed Student's *t* test

amount of another DNA damage marker, 53BP1, was significantly increased in AF patients compared to control SR patients (Fig. 10g, h). Finally, we examined nuclear circularity, a marker for oxidative stress-induced DNA damage[25], showing that nuclear circularity was significantly decreased in patients with AF compared to controls in SR (Fig. 10i). Thus, patients with (longstanding) persistent AF showed an increase in levels of PAR, indicative for PARP1 activation, markers of DNA damage, including γH2AX, 53BP1, and reduced levels of nuclear circularity. The features found in patients thus match the observations in tachypaced cardiomyocytes and *Drosophila*, indicating the clinical significance of PARP1 activation in (longstanding) persistent AF.

## Discussion

In the current study, we identified PARP1 activation as a key process in experimental AF by conferring depletion of the cellular content of $NAD^+$, an important component for cell function. Our results show that AF is associated with DNA damage and subsequent PARP1 activation. Activated PARP1 synthesizes PAR and in turn consumes $NAD^+$, resulting in functional loss in tachypaced cardiomyocytes and *Drosophila*. Accordingly, both inhibition of PARP1 and replenishment of $NAD^+$ protect against TP-induced $NAD^+$ depletion, oxidative DNA damage and contractile dysfunction in atrial cardiomyocytes and *Drosophila*. Consistent with these findings, PARP1 is also activated in atrial tissue of (longstanding) persistent AF patients, which correlates with the level of DNA damage. Taken together, these findings uncover a dominant role of PARP1 in TP-induced contractile dysfunction and cardiomyocyte remodeling and disease progression, thus implicating PARP1 as a possible therapeutic target in AF. We found PARP, specifically PARP1, to have a prominent role in AF progression. Both in tachypaced atrial cardiomyocytes and RAA/LAA tissue from persistent AF patients, we observed that PARP1 activation is caused by DNA damage. Moreover, in tachypaced atrial cardiomyocytes we showed that PARP1 activation results in the consumption of $NAD^+$ to such an extent that it depletes intracellular $NAD^+$ levels, thereby exacerbating oxidative damage to proteins and DNA. Activation of this sequel is likely triggered by a substantially increase in myocardial energy demand resulting from the four to sixfold increase in electrical and contractile activity during AF episodes. Subsequent failure to meet the increased energy demand results in progressive dysfunction of mitochondria and oxidative damage to proteins and DNA. DNA damage then activates PARP1 initiating the depletion of $NAD^+$. A unifying concept exists that, dependent on the amount of DNA damage, PARP1 activation initiates one of three major pathways[26]. Mild stress facilitates PARP1 activation to initiate DNA repair, without depleting $NAD^+$ levels. Intermediate stress conditions which induce more DNA damage, however, lead to excessive activation of PARP1 and depletion of $NAD^+$

resulting in energy depletion and functional loss, while even more severe stress triggers PARP1 cleavage and programmed cell death via apoptosis[15]. Importantly, both mild- and severe-stress conditions are not accompanied with cellular $NAD^+$ depletion. Because of the notable decrease in $NAD^+$ after TP of atrial cardiomyocytes, our observations thus indicate that persistence of AF represents an extensive stress condition. Interestingly, PARP1 cleavage was not observed at any stage in tachypaced cardiomyocytes and clinical AF, which likely explains the absence of apoptotic and/or necrotic cell death under these conditions[27]. This is in line with the observation that AF induces hibernation (myolysis) of the cardiomyocyte instead of cell death[28]. Our data from tachypaced atrial cardiomyocytes reveal that excessive activation of PARP1 and depletion of cellular $NAD^+$, a key coenzyme in cell metabolism[21], induce further DNA damage, and structural damage, and consequently electrophysiological and ion channel deterioration and functional loss[12]. These findings offer a novel paradigm to be tested in (longstanding) persistent AF patients. In addition, our findings are consistent with previous findings showing that structural remodeling underlies electrophysiological deterioration, including prolongation of APD (possibly via the reduction in potassium channel expression)[29–33], reduction in cardiomyocyte excitability and increased ADP dispersion, thereby creating a substrate for further arrhythmogenesis[34–37]. Although APD shorting was previously recorded in models for TP-induced AF, APD prolongation was observed in patients with lone paroxysmal AF, in atrial tissue of patients predisposed to AF and in various patient and animal studies for AF with underlying heart failure and structural changes in the atria[29–32,38,39], which is consistent with our current findings. Taken together, these studies provide compelling evidence that the predominant contributors to the substrate underlying AF are the structural and associated conduction abnormalities rather than changes in refractoriness. In addition, the studies may explain why current drug treatment directed at modulation of refractoriness shows limited efficacy, while its usage is further limited by a pro-arrhythmic action and noncardiovascular toxicity[40]. As such, PARP1-induced depletion of $NAD^+$ apparently functions as a key feed-forward switch in this chain of events, as PARP1 inhibition fully conserves $NAD^+$ levels, precludes oxidative protein and DNA damage and preserves structural, and therefore electrical and contractile function in tachypaced atrial cardiomyocytes. Consequently, in heart conditions associated with extensive PARP1 activation and $NAD^+$ depletion, as disclosed here in experimental AF, the pharmacological inhibition of PARP1 may offer substantial therapeutic benefits.

The prominent role of PARP1 in experimental AF progression thus extends previous observations in models of other cardiovascular disease, including heart failure models in mice, dogs, and rats, where activation of PARP1-induced endothelial dysfunction, myocardial hypertrophy, and remodeling[41,42]. In addition,

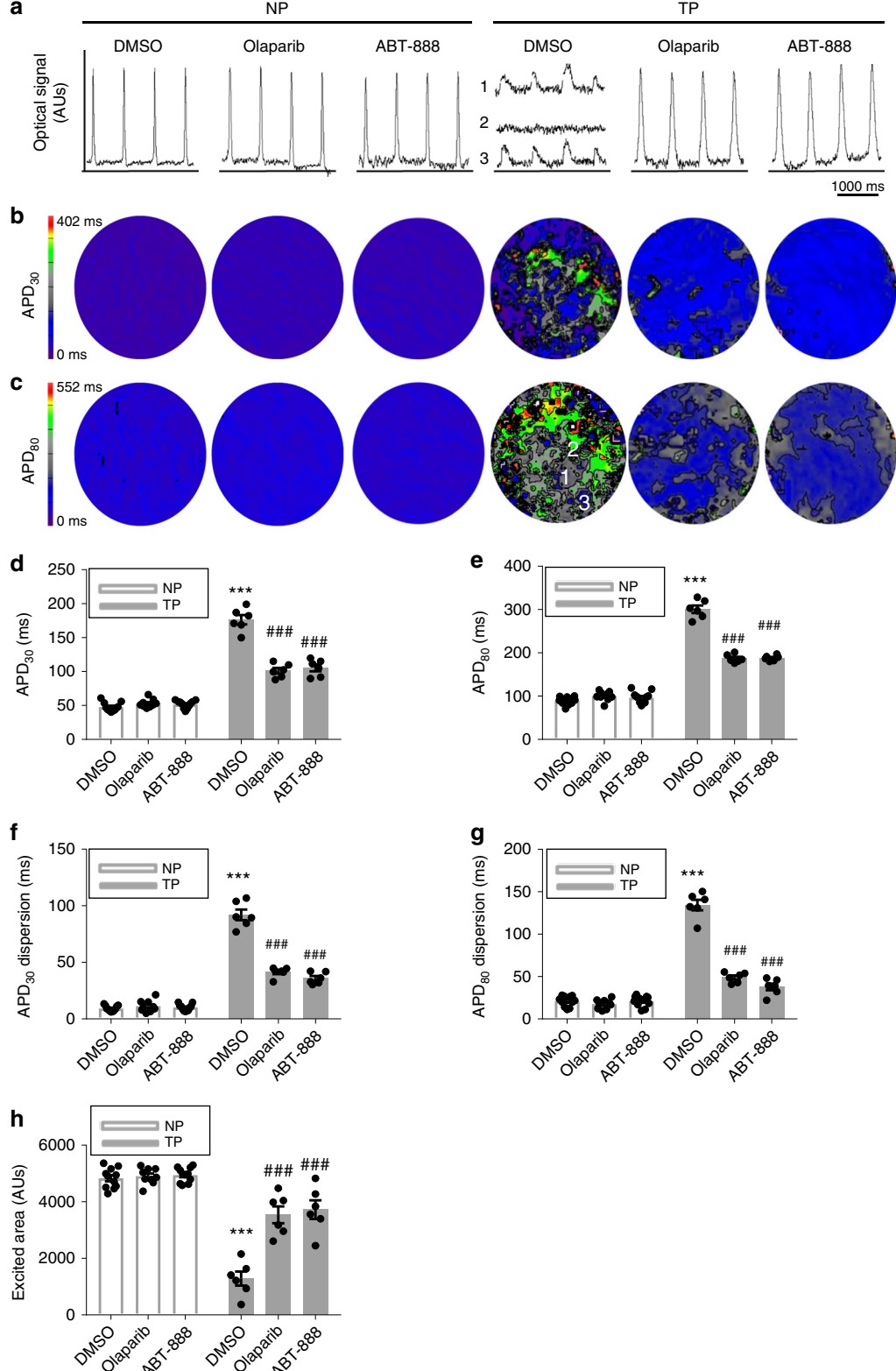

cardiac function in mouse models of diabetic cardiomyopathies showed marked improvement by the knockout of PARP1[16,43]. Importantly, previous studies in biopsy material from patients with heart failure reported increased expression and activation of PARP1 to contribute to disease progression[17,44]. Thus, the findings from the current study contribute to a further appreciation of the importance of PARP1 activation in cardiovascular diseases.

Our study implicates PARP1 inhibitors as potential therapeutics in AF. Early PARP1 inhibitors, such as 3-AB, may be unsuited for the treatment of patients as they compete with $NAD^+$ for the enzyme and consequently, inhibit PARP1 and other members of the PARP family, as well as mono-ADP-ribosyl-transferases and sirtuins, which are cardiac protective enzymes[13]. However, recently developed PARP inhibitors, such as ABT-888 and olaparib, exhibit

**Fig. 5** PARP1 inhibitors significantly attenuated tachypacing-induced electrophysiological deterioration in HL-1 cardiomyocytes. **a–h** Optical voltage mapping of HL-1 cardiomyocyte monolayers following 1-Hz electrical stimulation in control nonpaced (NP) or 8 h tachypaced (TP) HL-1 cardiomyocytes with 20 μM olaparib, 40 μM ABT-888 or vehicle DMSO 12-h pretreatment before tachypacing. **a** Representative filtered optical signal traces. To indicate electrical heterogeneity, three tracers which vary in time and space [1 and 3] to excitation block [2] in the TP DMSO group are depicted **b** typical $APD_{30}$ and **c** $APD_{80}$ maps for indicated groups. **d–h** Corresponding quantitative analysis of $APD_{30}$, $APD_{80}$, $APD_{30}$ dispersion, $APD_{80}$ dispersion and excited cell surface area, showing that TP resulted in significant APD prolongation (**a**, **d**, **e**), an increase in APD dispersion (**b**, **c**, **f**, **g**) and a significant decrease of excited cell surface area (**h**) in HL-1 cardiomyocyte monolayers. Pretreatment of HL-1 cultures with ABT-888 or olaparib significantly prevented the tachypacing-induced electrophysiological deteriorations (**a–h**). $^{***}P < 0.001$ vs. DMSO NP, $^{###}P < 0.001$ vs. DMSO TP. $n = 11$ for NP DMSO, $n = 9$ for NP olaparib TP, $n = 11$ for NP ABT-888, $n = 6$ for TP DMSO, $n = 6$ for TP olaparib, $n = 6$ for TP + ABT-888. $n =$ number of experiments. Data are all expressed as mean ± s.e.m. Individual group mean differences were evaluated with the two-tailed Student's $t$ test

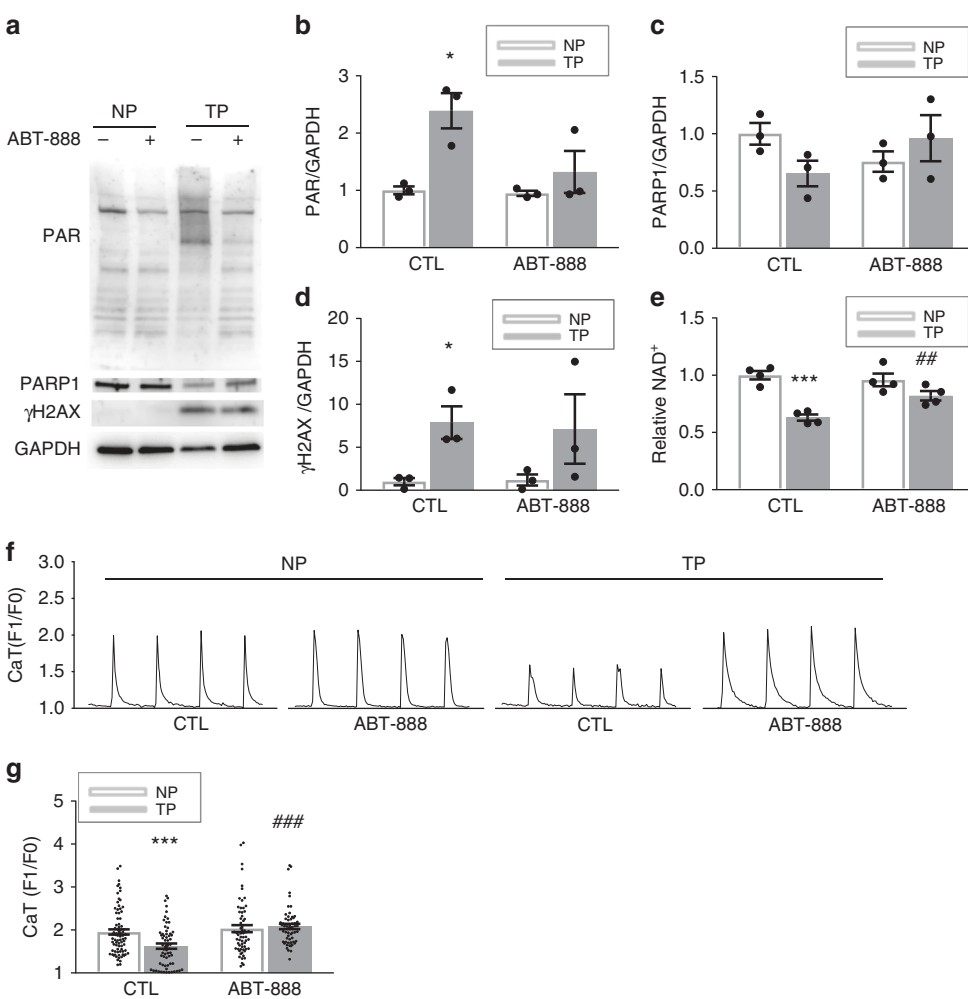

**Fig. 6** The PARP inhibitor ABT-888 attenuates tachypacing-induced PARP1 activation, $NAD^+$ depletion and CaT loss in adult rat atrial cardiomyocytes. **a–d** Representative Western blot and quantified data of PAR, PARP1, and γH2AX expression levels in rat atrial cardiomyocytes. Tachypacing (TP) significantly increased PAR levels, which was inhibited by the PARP inhibitor ABT-888. PARP1 protein levels were not changed by TP. TP significantly increased DNA damage (γH2AX) compared to NP. $^{*}P < 0.05$ vs. control (CTL) NP, $n = 3$ independent experiments. **e** TP reduced $NAD^+$ levels, which was prevented by PARP inhibitor ABT-8888. $^{***}P < 0.001$ vs. CTL NP $^{##}P < 0.01$ vs. CTL TP, $n = 4$ independent experiments. **f**, **g** Representative CaT traces and quantified CaT amplitude in control normal-paced (NP) or TP rat atrial cardiomyocytes pretreated with ABT-888 or vehicle DMSO (CTL). $^{***}P < 0.001$ vs. CTL NP, $^{###}P < 0.001$ vs. CTL TP, $n = 79$ cardiomyocytes for CTL NP, $n = 61$ for ABT-888 NP, $n = 63$ for CTL TP, $n = 57$ for CTL TP. Data are all expressed as mean ± s.e.m. Individual group mean differences were evaluated with the two-tailed Student's $t$ test

increased potency and specificity relative to earlier inhibitors. ABT-888 directly inhibits PARP1 and PARP2 without an action on sirtuins[45]. ABT-888 is currently in phase I and II clinical studies in cancer[46]. In addition to ABT-888, olaparib may represent a suitable candidate. Olaparib is used in phase III clinical trials for the treatment of metastatic breast cancers and has no effect on QT/QTc interval[47,48]. Another potential therapeutic option to protect against

AF-induced remodeling could be to replenish the $NAD^+$ pool by supplementation with $NAD^+$ or its precursors, such as nicotinamide and nicotinamide riboside. Interestingly, nicotinamide is not only a PARP1 inhibitor, but also a $NAD^+$ precursor. Nicotinamide can be converted into $NAD^+$ via the salvage pathway[49]. In heart failure, nicotinamide displayed a similar protective effect in experimental model systems[49], demonstrating a clear benefit of

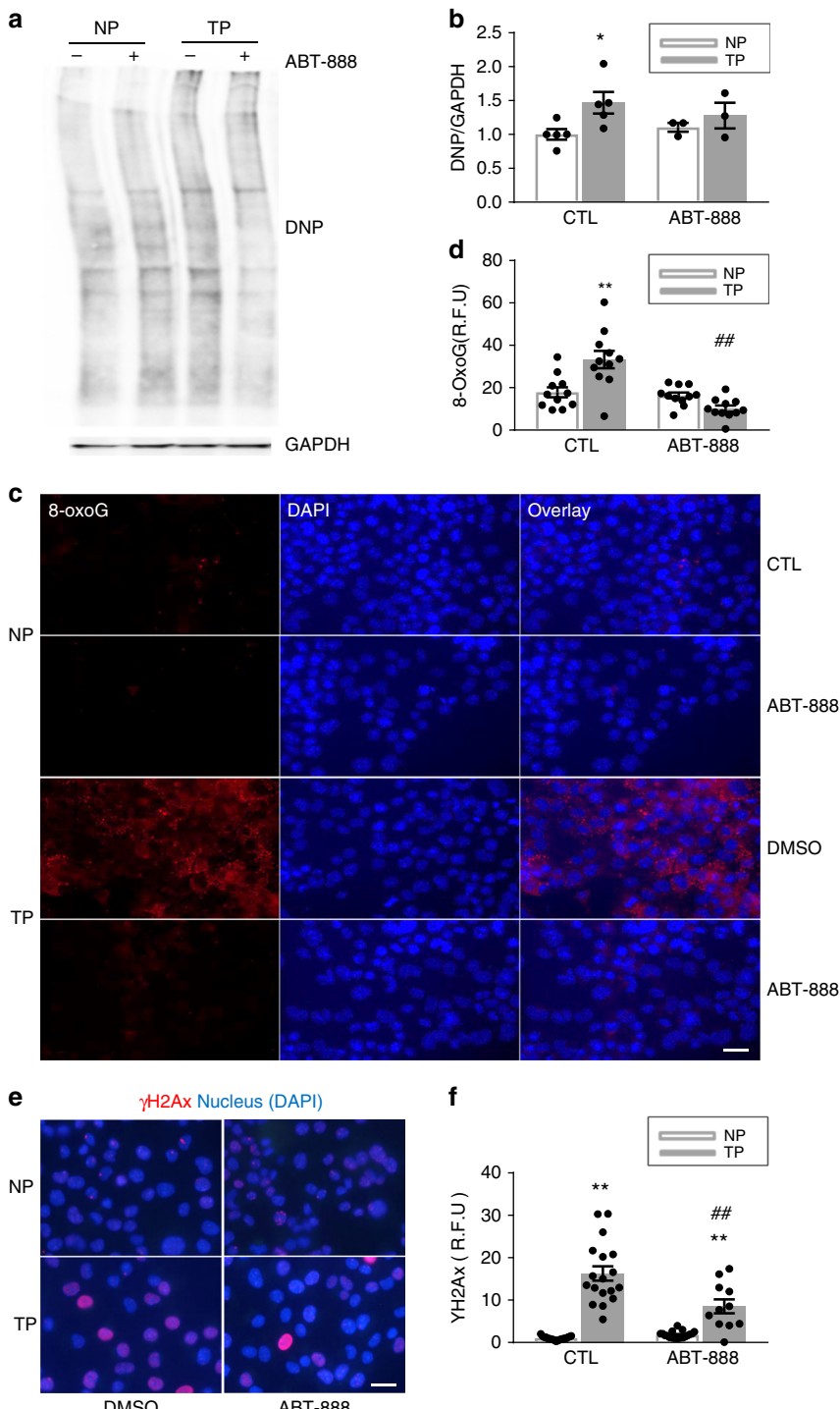

**Fig. 7** ABT-888 inhibits tachypacing-induced oxidative stress in HL-1 cardiomyocytes. HL-1 cardiomyocytes were pretreated with ABT-888 (40 μM) or vehicle DMSO (CTL) 12 h before tachypacing (TP). **a**, **b** Representative Western blot of protein carbonyl oxidation levels by DNP antibody staining and quantified data from $n = 5$ independent experiments for DMSO NP and TP, $n = 3$ independent experiments for ABT-888 NP and TP. *$P < 0.05$ vs. nonpaced (NP) DMSO. **c**, **d** Representative immunofluorescence staining of oxidative DNA damage marker 8-oxoguine (8-OxoG). $n = 11$ images from over 1000 cardiomyocytes. **$P < 0.01$ vs. NP CTL, ##$P < 0.01$ vs. CTL TP. **e**, **f** Representative immunofluorescence staining of DNA damage marker γH2Ax and quantified data from $n = 19$ images for DMSO NP and ABT-888 NP; $n = 18$ images for DMSO TP, $n = 11$ for ABT-888 TP from over 200 cardiomyocytes. **$P < 0.01$ vs NP CTL, ##$P < 0.01$ vs CTL TP. Scalebar is 15 μm. Data are all expressed as mean ± s.e.m. Individual group mean differences were evaluated with the two-tailed Student's $t$ test

normalizing NAD$^+$ levels in failing hearts. The high translational potential and the applicability in humans recently prompted an open-label pharmacokinetics study with nicotinamide riboside (Niagen, Chromadex) in healthy volunteers, showing that nicotinamide riboside stably induced circulating NAD$^+$ and was well tolerated (even up to $2 \times 1000$ mg/day)[50]. Therefore, nicotinamide riboside represents a potential therapy for diseases in which NAD$^+$ depletion has been implicated, such as heart art failure and

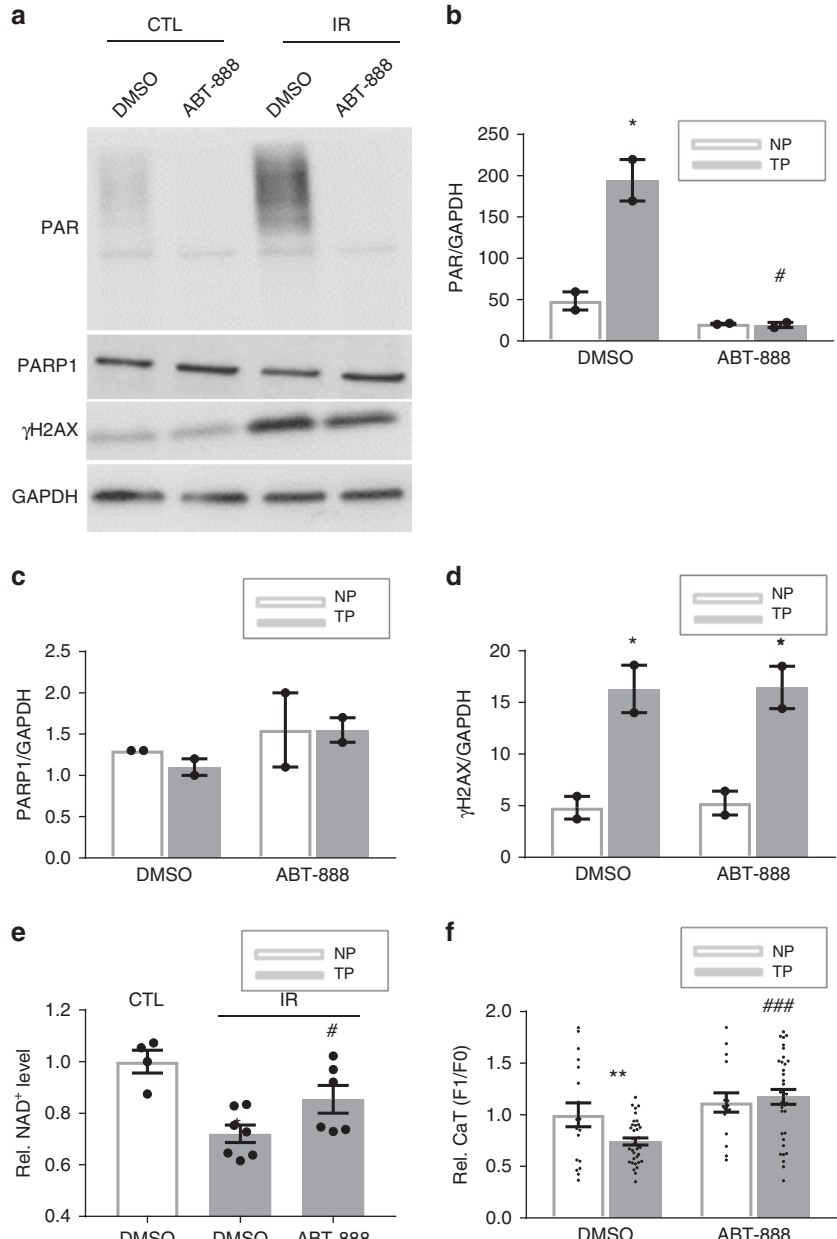

**Fig. 8** Irradiation-induced DNA damage results in PARP1 activation, NAD$^+$ reduction and contractile dysfunction in HL-1 cardiomyocytes. **a** Representative Western blot of PAR and γH2AX in control (CTL) and irradiated (IR) HL-1 cardiomyocytes treated either with vehicle (DMSO) or ABT-888. **b–d** Quantified data of Western blot in **a**, showing significant increase in PAR and γH2AX levels, indicating PARP1 activation and presence of DNA damage, respectively, due to IR. ABT-888 pretreatment protected against PAR induction. $^*P < 0.05$, $^{**}P < 0.01$ vs. CTL. $n = 2$ independent experiments. No significant difference was found in the amount of PARP1. **e** Relative NAD$^+$ levels in CTL and IR HL-1 cardiomyocytes. IR resulted in reduction in NAD$^+$ levels, which was prevented by ABT-888 pretreatment. $^*P < 0.05$ vs. CTL treated with vehicle DMSO, $^\#P < 0.05$ vs. IR treated with vehicle DMSO, $n = 4$ independent experiments for CTL DMSO, $n = 7$ independent experiments for IR DMSO, $n = 6$ independent experiments for IR ABT-888. **f** Quantified CaT amplitude in CTL or IR HL-1 cardiomyocytes pretreated with ABT-888 (3 mM) or vehicle (CTL). ABT-888 protected against IR-induced CaT loss. $^{**}P < 0.01$ vs. CTL DMSO; $^{\#\#\#}P < 0.0001$ vs. IR DMSO, $n = 18$ cardiomyocytes for CTL DMSO, $n = 37$ for IR DMSO, $n = 16$ for CTL ABT-888, $n = 34$ for IR ABT-888. Data are all expressed as mean ± s.e.m. Individual group mean differences were evaluated with the two-tailed Student's $t$ test

AF. Importantly, conduction of clinical trials with drugs directed at PARP1-NAD$^+$ pathway deserves strong priority, particularly to preserve quality of life and to attenuate devastating complications such as heart failure or stroke. Moreover, advancing therapeutic options in AF has substantial economic impact by reducing the number of repetitive hospitalizations and visits to healthcare professionals.

In summary, this study documents the induction of DNA damage, extended activation of PARP1, and subsequent NAD$^+$ depletion, as key events in cardiomyocyte functional loss and experimental AF progression. Importantly, inhibition of PARP1 activation prevents NAD$^+$ depletion and conserves cardiomyocyte function in models of AF, thereby attenuating disease progression. Our findings indicate that inhibition of PARP1 may

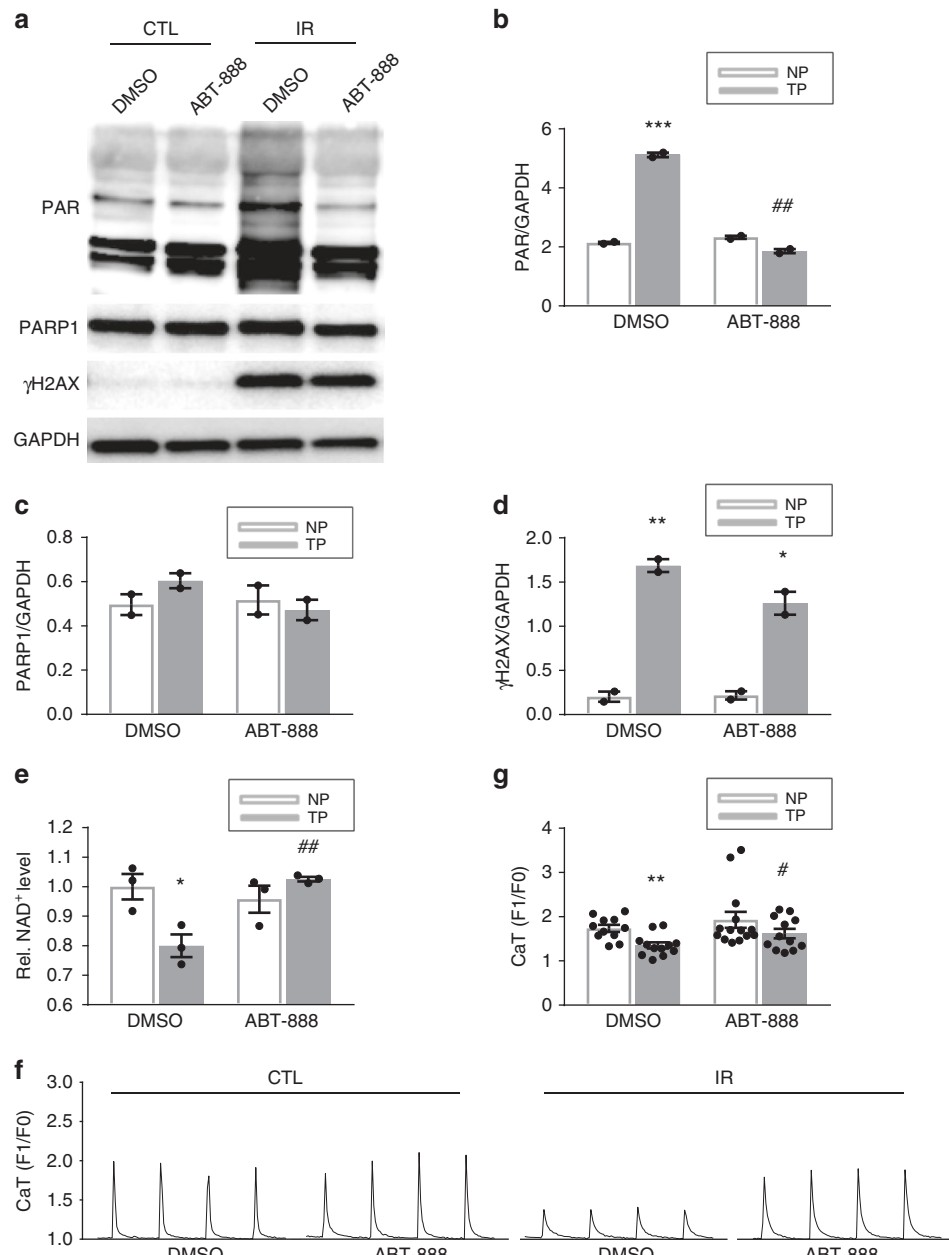

**Fig. 9** Irradiation-induced DNA damage results in PARP1 activation, NAD$^+$ reduction and contractile dysfunction in rat atrial cardiomyocytes.
**a** Representative Western blot showing PAR and γH2AX levels due to irradiation (IR) with and without ABT-888 pretreatment. **b–d** Quantified data of Western blot in **a**, showing significant increase in PAR and γH2AX levels, indicating PARP activation and presence of DNA damage, respectively, due to IR. ABT-888 pretreatment protected against PAR induction. *$P < 0.05$, **$P < 0.01$, ***$P < 0.001$ vs. control nonirradiated (CTL) rat atrial cardiomyocytes treated with vehicle DMSO. ##$P < 0.01$ vs. IR treated with vehicle DMSO, $n = 2$ independent experiments. No significant difference was found in the amount of PARP1. **e** Relative NAD$^+$ levels in CTL and IR rat atrial cardiomyocytes treated with DMSO or ABT. IR resulted in reduction in NAD$^+$ levels which was prevented by ABT-888 pretreatment *$P < 0.05$ vs. CTL treated with vehicle DMSO, ##$P < 0.01$ vs. IR treated with vehicle DMSO, $n = 3$ independent experiments. **f** Quantified CaT amplitude in CTL or IR rat atrial cardiomyocytes pretreated with ABT-888 or vehicle (CTL). ABT-888 protected against IR-induced CaT loss. **$P < 0.01$ vs. CTL DMSO; #$P < 0.05$ vs. IR DMSO, $n = 11$ atrial cardiomyocytes for CTL DMSO, $n = 14$ for CTL ABT-888, $n = 12$ for IR DMSO and IR ABT-888. Data are all expressed as mean ± s.e.m. Individual group mean differences were evaluated with the two-tailed Student's $t$ test

serve as a novel therapeutic target in AF by conserving the cardiomyocyte metabolism.

We uncovered a role for DNA damage-induced PARP1 activation in cardiomyocyte dysfunction in AF by utilizing various experimental model systems, including tachypaced HL-1 cardiomyocyte and *Drosophila* models which are easily accessible to genetic manipulations. The spontenous contraction rate of these cardiomyocytes is ~0.5–1 Hz in a 2D culture dish (instead of 5–7 Hz in in vivo mice/rats), a 5–10-fold rate increase by TP induces various endpoints of human AF remodeling[8,51,52]. Although observations were consistent between different experimental AF models (in vitro HL-1 cardiomyocyte and rat atrial cardiomyocytes, *Drosophila*) and in heart tissue from AF patients, our data do not provide conclusive evidence about involvement of PARP1

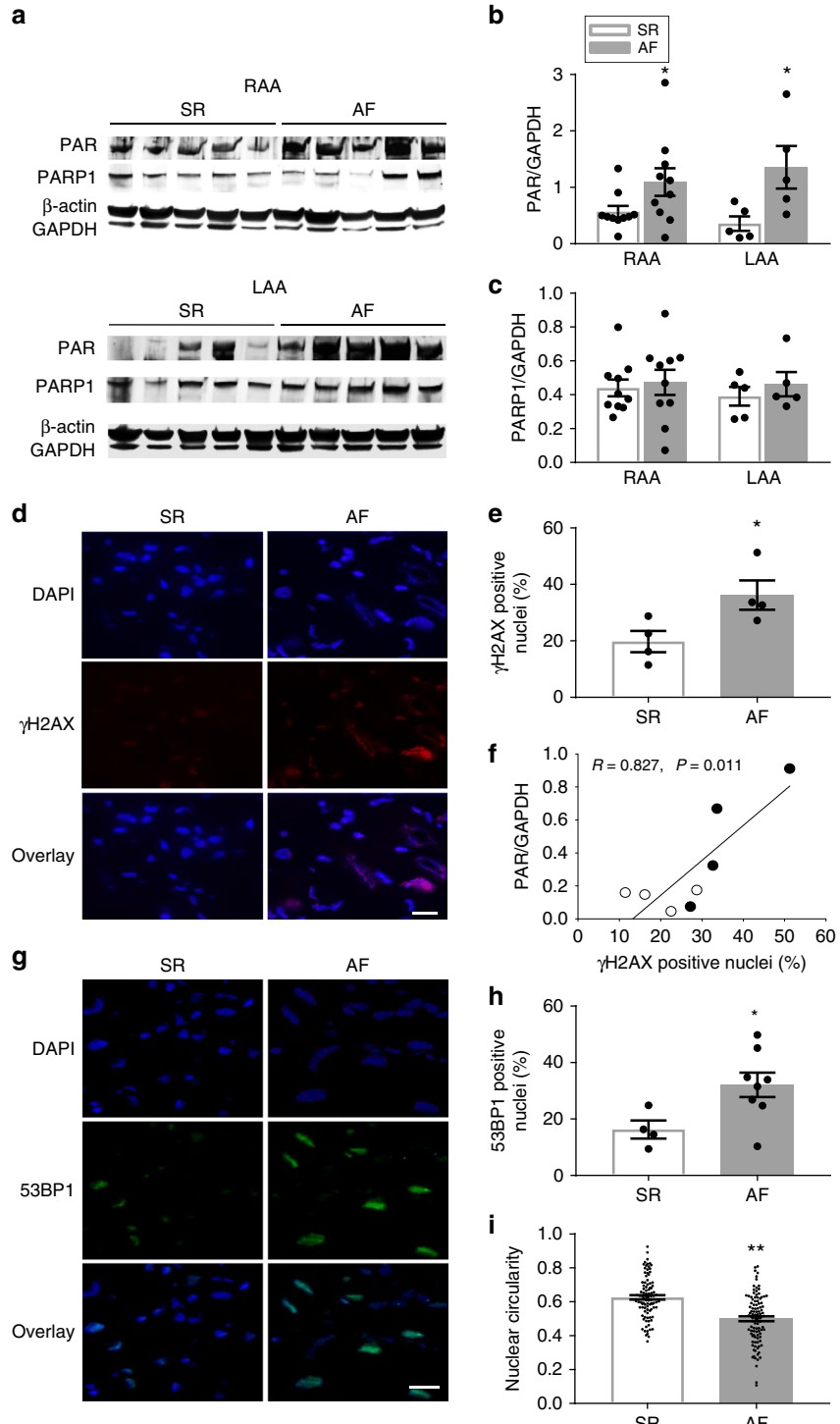

**Fig. 10** Patients with AF reveal DNA damage and PARP1 activation. **a–c** Representative Western blots of PAR and PARP1 levels in RAA and LAA of SR and AF patients with underlying mitral valve disease, showing significant increase in PAR levels in AF patients compared to SR. PARP1 expression levels remain unchanged between AF and SR patients. $n = 10$ for SR RAA, $n = 5$ for SR LAA, $n = 10$ for AF RAA, $n = 5$ for AF LAA *$P < 0.05$ SR RAA vs. AF RAA, SR LAA vs. AF LAA. **d** Representative immunofluorescence staining of γH2AX in RAA of SR and AF patients. **e** Quantified data of positive nuclear γH2AX staining of RAA from SR and AF patients. $n = 4$ for SR, $n = 5$ for AF. **f** PARP1 activity (PAR) correlates significantly with DNA damage (γH2AX positive nuclei). $n = 4$ for each group. SR: open circle and AF: filled circle. **g** Representative immunofluorescent staining of 53BP1 in RAA of SR and AF patients. **h** Quantified data of positive nuclear 53BP1 staining in RAA of SR and AF patients. $n = 4$ for SR, $n = 7$ for AF. **i** Quantification of nuclear circularity in SR and AF patients showing AF patients with elongated nuclei. $n = 94$ nuclei from 4 SR patients and $n = 104$ nuclei from 7 AF patients. **d–i** *$P < 0.05$ SR vs. AF, **$P < 0.01$ SR vs. AF. Scalebar is 40 μm. Data are all expressed as mean ± s.e.m. Individual group mean differences were evaluated with the two-tailed Student's $t$ test

in AF progression in patients. Nevertheless, previous findings on the role of heat shock proteins, HDAC6 and autophagy, initially made in HL-1 cardiomyocyte and *Drosophila* models have been confirmed in all instances in the tachypaced dog model and clinical human AF[8,51,52]. Therefore, the tachypaced HL-1 cardiomyocyte and *Drosophila* model may have merit to identify potential signaling pathways involved in AF remodeling. Future research should elucidate the relevance of the DNA damage-induced PARP1 activation pathway in clinical AF with or without underlying heart diseases.

Nevertheless, clinical development of PARP1 inhibitors for AF awaits two further steps. First, the action of recently developed PARP1 inhibitors, such as ABT-888, should be investigated in large animal AF models to substantiate its efficacy in relation to the stage of AF. Secondly, current clinical trials should indicate a favorable safety profile, especially in case the animal studies indicate a beneficial effect of long-term use in halting progression from paroxysmal to persistent AF.

## Methods

**HL-1 cardiomyocyte model, Ca$^{2+}$ measurements and drug treatment**. HL-1 cardiomyocytes derived from adult mouse atria were obtained from Dr. William Claycomb (Louisiana State University, New Orleans) and cultured in complete Claycomb medium (Sigma) supplemented with 10% fetal bovine serum (PAA Laboratories GmbH, Austria), 100 U per ml penicillin (GE Healthcare), 100 μg per ml streptomycin (GE Healthcare), 4 mM L-glutmaine (Gibco), 0.3 mM L-ascorbic acid (Sigma), and 100 μM norepinephrine (Sigma). HL-1 cardiomyocytes were cultured on cell culture plastics or on glass coverslips coated with 0.02% gelatin (Sigma) in a humidified atmosphere of 5% CO$_2$ at 37 °C. The cardiomyocytes, which have a basal spontaneous contraction rate of ~0.5–1 Hz[4], were subjected by TP to a 5–10-fold rate increase as observed in clinical AF (5 Hz, 40 V, pulse duration of 20 ms) with a C-Pace100 culture pacer (IonOptix) for 12 h unless stated otherwise. HL-1 cardiomyocytes followed the pacing rate. CaT were imaged by Solamere-Nipkow-Confocal-Live-Cell-Imaging system (based on a Leica DM IRE2 Inverted microscope). A 2 μM of the Ca$^{2+}$-sensitive Fluo-4-AM dye (Invitrogen) was loaded into HL-1 cardiomyocytes by 45 min incubation, followed by 3 times washing with phosphate-buffered saline (PBS). Ca$^{2+}$ loaded cardiomyocytes were excited by 488 nm and emitted at 500–550 nm and visually recorded with a ×40-objective. CaT measurements were performed in a blinded manner by selection of normal-shaped cardiomyocytes with the use of bright field settings, followed by a switch to the fluorescent filter to determine the CaT.

Prior to 12 h TP, HL-1 cardiomyocytes were treated for 12 h with the PARP inhibitors 3-aminobenzamide (3-AB, Sigma-Aldrich), ABT-888 (Selleckchem), olaparib (Selleckchem), beta-nicotinamide adenine dinucleotide hydrate (NAD$^+$, Sigma-Aldrich) or transfected with scrambled siRNA (control, Ambion) PARP1 siRNA (Ambion), or PARP2 siRNA (Santa Cruz) to study the specific role of PARP1 and PARP2, respectively.

**Rat atrial cardiomyocyte model, Ca$^{2+}$ test and drug treatment**. Adult Wistar rats (~200 g) were injected with heparin 15 min before atrial cardiomyocyte isolation, followed by anesthetisation (2% isoflurane and 98% O$_2$). Hearts were excised and placed in ice-cold, oxygenated buffer solution containing (in mM) 134 NaCl, 10 HEPES, 4 KCl, 1.2 MgSO$_4$, 1.2 Na$_2$HPO$_4$, and 11 D-glucose (pH 7.4). Freshly excised rat hearts were mounted on a Langendorff setup and perfused retrogradely through the aorta for 30 min with oxygenated buffer solution of 37 °C, to which 66.7 mg perL librase (Roche) was added. Following Langendorff perfusion, the atria were cut off the heart and rinsed in isolation solution containing (in mM): 100 NaCl, 5 Hepes, 20 D-glucose, 10 KCl, 5 MgSO$_4$, 1.2 KH$_2$PO$_4$, 50 Taurin, 0.5% bovine serum albumin (BSA) (pH 7.4), transferred to a 15-ml tube containing 10 ml of isolation solution plus 0.02 mM CaCl$_2$ and 0.02 U per ml DNase, gently triturated for 7 min, and subsequently filtered through a 200μm mesh filter into another 15-ml tube, followed by centrifugation for 1 min at 700 × g. The supernatant was removed and the pellet containing atrial cardiomyocytes was resuspended carefully in 10 ml of isolation solution plus 0.02 mM CaCl$_2$. Next, the Ca$^{2+}$ concentration was increased in 5-min steps from 0.1, 0.2 mM to 0.4 mM Ca$^{2+}$. Atrial cardiomyocytes were left to sink for 20 min and transferred into laminin-coated plates in plating medium (M199 medium plus 5% fetal calf serum) for 2 h followed by replacement with M199 medium plus Insulin-Transferrin-Sodium Selenite Supplement (Sigma). The isolated adult rat atrial cardiomyocytes have a basal spontaneous contraction rate of ~0.5–1 Hz in vitro. The rat experiments complied with all relevant ethical regulations and theVUmc approved the study protocol (DEC FYS 14-06).

Prior to TP, atrial cardiomyocytes were treated for 2 h with the PARP inhibitors ABT-888 (Selleckchem) or olaparib (Selleckchem), followed by 2 h TP at 5 Hz, 30 V with a pulse duration of 2 ms. Control atrial cardiomyocytes were either nonpaced (NP) or paced for 2 h at 1 Hz, 30 V and pulse duration of 2 ms. Atrial

cardiomyocytes followed the pacing rate. CaT measurement was performed according to previous studies with minor changes[2,53]. In short, atrial cardiomyocytes were washed twice with M199 medium, incubated with the Ca$^{2+}$ dye Fluo-4 (1 μg per ml) in M199 medium for 15 min, and rinsed twice again with M199 medium. The Fluo-4-loaded cardiomyocytes were excited at 488 nm and the light emitted at 500–550 nm and recorded with a high-speed confocal microscope (Nikon A1R). Bright field settings were used to randomly select normal-shaped cardiomyocytes, followed by a switch to the fluorescent filter to determine the CaT. As such, CaT measurements were conducted in a blinded manner.

**Drosophila stocks, TP, and heart contraction assays**. The wild-type *Drosophila melanogaster* strain w1118 strain was used for all drug screening (PARP inhibitors or NAD$^+$) experiments. Hereto, female and male adult flies were crossed. After 3 days, flies were removed from the embryos-containing tubes and drugs or the same amount of vehicle (DMSO) were added to the food. *Drosophila* was incubated at 25 °C for 48 h, with larvae consuming the drug/vehicle prior to entering the prepupae stage. The *Drosophila* prepupae were collected and subjected to TP for 20 min (4 Hz, 20 V, pulse duration of 5 ms) and heart wall functions were measured as described in detail below. See Supplementary Table 1 for the applied doses of 3-AB, ABT-888 and NAD$^+$.

To create the knockdown of PARP1 in *Drosophila*, two PARP1 UAS-RNAi *Drosophila* lines, from the Vienna *Drosophila* RNAi Center (VDRC, ID:330230) and Bloomington *Drosophila* Stock Center (BDSC, ID:34888), were utilized. Both RNAi lines were crossed with a Hand-GAL4 driver strain (kind gift of Prof. Dr. Achim Paululat)[54]. As control, wild-type flies w1118 were crossed with Hand-GAL4 driver flies. Prepupae of F1 offspring were tachypaced as previously described[8].

Heart wall contractions were measured utilizing high-speed digital video imaging (100 frames per s) before and after TP in at least duplicated 10 s-movies. Movies were used to prepare heart wall traces and M-mode cardiography. Hereto, 1-pixel width lines were drawn across the heart wall, followed by determination of Plot-Z axis profile (based on contrast changes) to generate heart wall traces or kymographs (via the kymograph plugin of Image J) for M-mode cardiography. To determine the heart rate and arrhythmicity index (defined as the standard deviation of the heart period normalized to the median heart period of each fly followed by averaging across flies)[55], the heart wall traces were further analyzed with the use of Drosan software, which was modified from the software originally developed to determine human heart rate and arrhythmicity[56,57]. The detailed algorithm of the Drosan software is described in the Supplementary Methods section and overview of the outcome parameters is presented in Supplementary Table 2.

**Patients**. Before surgery, patient characteristics were collected (Supplementary Table 3). RAA and/or LAA tissue samples were obtained from patients with coronary artery and/or valvular heart disease having SR or (longstanding) persistent AF. After excision, atrial appendages were immediately snap-frozen in liquid nitrogen and stored at −80 °C. The study conformed to the principles of the Declaration of Helsinki and complied with all relevant ethical regulations. The Erasmus Medical Center Review Board approved the study (MEC-2014-393), and all patients gave written informed consent.

**Protein extraction and Western blot analysis**. HL-1 cardiomyocytes or human tissue samples were lysed in radioimmunoprecipitation assay (RIPA) buffer containing PBS, Igecal ca-630, eoxycholic acid, and sodium dodecyl sulfate (SDS)[2,8]. In short, equal amounts of protein homogenates were separated by SDS-polyacrylamide gel electrophoresis (SDS-PAGE), transferred onto nitrocellulose membranes, and probed with antibodies directed against poly (ADP-Ribose) (PAR, 1:1000, BD bioscience, 551813), PARP1 (1:500, Santa Cruz, sc-25780), γH2AX (1:1000, Millipore, 05–636), Cav1.2 (1:200, Alomone Labs, ACC-003), Kv11.1 (1:400, Alomone Labs, APC-062), Kir3.1 (1:200, Alomone Labs, APC-005), β-actin (1:1000, Santa Cruz, sc-47778), or GAPDH (1:5000, Fitzgerald, 10R-G109A). Membranes were subsequently incubated with horseradish peroxidase-conjugated goat anti-mouse or goat anti-rabbit secondary antibodies (Dako). Signals were detected by the ECL detection method (Amersham) and quantified by densitometry (Syngene, Genetools). Original uncropped blots are available at the Supplementary Information section (Supplementary Figure 12).

**NAD assay**. NAD and NADH levels, in which NAD represents the sum of NAD$^+$ and NADH, were measured according the manufacturer's instructions of the assay kit (Abcam, ab65348). In short, HL-1 cardiomyocytes were lysed in NAD extraction buffer and the protein concentration was determined (BioRad Laboratories). To measure NAD, after equalizing the protein concentration, 50 μl of each sample was mixed with 100 μl NAD cycling buffer and incubated at room temperature (RT) for 5 min to convert NAD$^+$ to NADH, followed by the addition of 10 μl NADH developer buffer and 2 h incubation at RT. NAD/NADH levels were measured at 450 nm (BioTek Synergy 4 plate reader). To measure NADH, NAD$^+$ in each sample was decomposed by incubation at 60 °C for 30 min before measurement. Notably, in accordance with previous findings, the NADH amount in

cultured cardiomyocytes and tissue was below the detection limit[58]. Therefore, the NAD+ amount per µg of total protein was used as endpoint.

**Comet assay.** To evaluate DNA damage in cardiomyocytes, an alkaline comet assay kit (Trevigen) was utilized according to the manufacturer's instructions with minor changes. HL-1 cardiomyocytes were trypsinized, harvested by centrifugation, suspended at $2 \times 10^5$ cells per ml in PBS, combined with 45 µl melted LAM agarose at ratio of 1:10 (v:v) and immediately pipetted onto CometSlides. Slides were dried for 30 min at 4 °C, incubated firstly in lysis solution for 1 h and then in freshly prepared alkaline unwinding solution (pH > 13) for 1 h. After placing the slides in 4 °C alkaline electrophoresis solution, electrophoresis at 21 V for 30 min was performed. After incubation for 2 times 5 min in demineralized $H_2O$ and once for 5 min in 70% ethanol, slides were dried at 37 °C, stained with SYBR Gold for 30 min at RT in the dark, rinsed in water and dried again at 37 °C. Finally, comets were visualized after excitation at 496 nm by fluorescence microscopy (Leica Microsystems) at 522 nm. DNA damage was quantified by scoring the percentage of DNA in the tail, using the Image J Marco "Comet_Assay" based on an NIH Image Comet Assay developed by Herbert M. Geller (1997).

**Irradiation of cardiomyocytes.** To induce DNA damage, HL-1 atrial cardiomyocytes received 10 Gy and rat atrial cardiomyocytes 40 Gy of irradiation with a dose rate of 0.0562 Gy per second by utilizing a cobalt-60 gamma-source (Gammacell 220 Research Irradiator, MDS Nordion, Canada). HL-1 and rat atrial cardiomyocytes were treated with 40 µM ABT-888 (12 h) or 5 µM ABT-888 (2 h), respectively, prior to the irradiation. After irradiation, cardiomyocytes were either prepared for Western blot analyses, NAD+ level measurements or CaT recordings.

**Protein oxidation detection.** To evaluate oxidative stress in cardiomyocytes, OxyBlot protein oxidation detection kit (Millipore, S7510) was used, following the company's instructions. In short, cardiomyocytes were lysed in RIPA buffer containing 1% beta-mercapto-ethanol (Sigma). A 10 µg of protein was denatured in 6% SDS, derivatized by incubation for 15 min in 2,4-dinitrophenylhydrazine (DNPH) solution, followed by the addition of neutralization solution. After neutralization, protein samples were subjected to SDS-PAGE, transferred onto nitrocellulose membranes and probed with anti-dinitrophenyl (DNP) antibody (1:150) for 1 h at RT. Horseradish peroxidase-conjugated goat anti-rabbit IgG (1:300) was used as secondary antibody. All reagents were included in the kit. Signals were detected by the ECL detection method (Amersham) and quantified by densitometry (Syngene, Genetools).

**Quantitative reverse transcription PCR.** Total RNA was isolated from HL-1 cardiomyocytes utilizing the nucleospin RNA isolation kit (Machery-Nagel). First strand cDNA was generated by M-MLV reverse transcriptase (Invitrogen) and random hexamer primers (Promega). Relative changes in transcription level were determined using the CFX384 Real-time system C1000 thermocycler in combination with SYBR green supermix (both from BioRad Laboratories). Calculations were performed using the comparative computed tomography method according to User Bulletin 2 (Applied Biosystems). Fold inductions were adjusted for GAPDH levels. Primer pairs used included PARP1 F: CACCTTCCAGAAGCAG GAGA and R: GCAAGAAATGCAGCGAGAGT; PARP2 F: TCCTCTGGGCATC ATCTTCT and R: AAGCTGGGAAAGGCTCATGT. CACNA1C F: CAAACAAC AGGTTCCGCCTG and R: ATCTTTAGAGCAATTTCAATGGTGA. KCNQ1 F: GCCTCACTCATCCAGACTGC and R: GGACAGAAGCGTGTGACTCC. KCNH2 F: GGCGTACAGACAAGGACACA and R: CAGGGCCCTCATCTTCA CTG. KCNJ3 F: TTCATCCTCCAACAGCACCC and R: GGCCATAGCTGCTTG CTAGA. GAPDH F: CATCAAGAAGGTGGTGAAGC and R: ACCACCCTGTT GCTGTAG. ACTB F: GGCTGTATTCCCCTCCATCG and R: CCAGTTGGTAA CAATGCCATGT. Primer pairs used in *Drosophila* included PARP1 F: TGGTTT GCGTCAGGTGAAGA and R: TCGCGAAACCTGAAGTAGGC; Actin5C: F: GA GCACGGTATCGTGACCAA and R: GCCTCCATTCCCAAGAACGA.

**Immunofluorescent microscopy of cardiomyocytes.** HL-1 cardiomyocytes were grown on coverslips until 80% confluence and subjected to TP for various time periods, with or without drug treatment. Immediately after pacing, cardiomyocytes were rinsed in PBS and fixed with 4% formaldehyde for 15 min, rinsed twice with PBS, permeabilized by incubation with 0.1% triton X-100 in PBS for 10 min, rinsed twice in PBS and blocked with blocking solution (0.5% BSA and 0.15% glycine in PBS) for 10 min. After blocking, cardiomyocytes were incubated with primary antibodies for 2 h at RT. After rinsing the cardiomyocytes three times with blocking solution, cardiomyocytes were incubated with secondary antibodies for 45 min at RT shielded from light, followed by rinsing with blocking solution three times and PBS twice. Lastly, cardiomyocytes were incubated with mounting media containing DAPI (Vectashield), sealed with nail polish and used for fluorescent microscopy (Leica Microsystems). Antibodies used were: anti-γH2AX (1:100, Millipore, 05-636), anti-PAR (1:200, BD Bioscience, 551813), anti-PARP1 (1:200, Santa Cruz, sc-25780), anti-oxoguanine 8 (1:100, Abcam, ab64548), goat anti-rabbit FITC (1:200, Invitrogen, 65-6111), and goat anti-mouse TRITC (1:200, Southern Biotech, 1021-03). For quantification, Image Pro software was used to calculate the total fluorescent (green for FITC and red for TRITC) signal per image as well as the DAPI

signal. The total fluorescent signals, corresponding to the expression of PARP1, PAR or γH2AX, were divided by the respective blue signals (DAPI), representing the cell number.

**Immunofluorescent microscopy and nuclear shape analysis.** The frozen RAA samples of SR and AF patients were used for staining of γH2AX and 53BP1. Frozen sections were cut into 5 µm slices. Sections were air dried for 30 min, fixed in 4% formaldehyde for 10 min at RT, washed 3 times with PBS for 10 min, then permeabilized with 0.3% Triton X-100 (in PBS) for 10 min at RT and washed 3 times for 5 min with PBS. After blocking of the sections with 1% BSA blocking solution for 30 min at RT, sections were incubated with primary antibodies directed against γH2AX (1:100; Millipore, 05-636) or 53BP1 (1:100; Santa Cruz Biotechnology, sc-22760) overnight at 4 °C. After washing with PBS for 3 times 10 min, slides were incubated with secondary antibodies and 1% human serum, TRITC labeled goat anti-mouse (1:200; Southern Biotech, 1021-03) and FITC labeled goat anti-rabbit (1:200, Invitrogen, 65-6111) for 1 h at RT and protected from light. Following 3 washes of 10 min, DAPI mounting medium (Vectashield) was applied to the sections, after which they were covered with coverslips and sealed. Slides were stored at 4 °C for a few hour and subsequently used for fluorescent microscopy (Leica Microsystems). γH2AX and 53BP1 positive nuclei were expressed as the percentage of the total number of nuclei (typically about 200).

The nuclear shape of cardiomyocytes in RAAs of SR and AF patients was determined by measuring its circularity (form factor) with Image J 1.48 software (US National Institute of Health). Hereto, 8-bit images of DAPI-stained nuclei were converted to binary photos by the method of "make binary" in ImageJ, traced by hand and the circularity was calculated by the formula $4\pi*A$ per $P^2$, in which $A$ denotes the surface area and $P$ the perimeter. The circularity of a perfect round circle and a line segment are 1 and 0, respectively[59].

**Statistical analysis.** Results are expressed as mean ± standard error of the mean. Biochemical analyses were performed at least in duplicate. Individual group mean differences were evaluated with the two-tailed Student's $t$ test. Correlation was determined with the Spearman correlation test. To compare continuous variables with a skewed distribution, the Mann–Whitney $U$ test was applied. All $P$ values were two sided. Values of $P < 0.05$ were considered statistically significant. SPSS version 20 (IBM Analytics) was used for all statistical evaluations.

**Reporting summary.** Further information on experimental design is available in the Nature Research Reporting Summary linked to this article.

## Data availability

All the data used in this study are available within the article, Supplementary Information, or available from the authors upon request.

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

## Acknowledgments

This research has been supported by LSH-TKI (40-43100-98-008), the Dutch Heart Foundation (2013T096, 2013T144, and 2017T029) and the Netherlands Organization for Scientific Research (NWO), as part of their joint strategic research program: "Earlier recognition of cardiovascular diseases" (AFFIP: 14728). This project is partially financed by the PPP allowance made available by TOP Sector Life Sciences & Health to the Dutch Heart foundation to stimulate public-private partnerships. Furthermore, we acknowledge Max Goebel, Diederik Kuster and Elza van Deel for their technical assistance.

## Author contributions

D.Z., D.A.P., A.A.F.V., M.W., D.M.S.M., M.T., E.A.H.L., and N.M.S.d.G designed the experiments; A.M.v.R. designed the Drosan software; D.Z., X.H, Jin.L., J.L, L.B.t.b., N.M., and F.H.B. performed experiments; D.Z., X.H, Jin.L., J.L., and B.J.J.M.B. analyzed data; D.Z., X.H., J.L., A.M.v.R., R.H.H., and B.J.J.M.B. wrote the paper.

## Additional information

**Competing interests:** The authors declare no competing interests.

