## [Peer Review File · Nature Communications]

Reviewers' comments:

Reviewer #1 (expert in PARPs)

Remarks to the Author:

Major comments

While the in vitro cell line data and the drosophila data are clear, it should be at least acknowledged that some sort of in vivo model of fibrillation (rodent or large animal) would be an additional, important connecting step prior to considering clinical translation. Of course, it would be even better if the authors could add some in vivo rodent or large animal data to the paper.

When considering translation (repurposing) of PARP inhibitors, the compound of choice is olaparib (already approved and in patients for ovarian cancer) as opposed to veliparib used here (which is in clinical trials but not approved). Would it be possible to add some olaparib data to some of the key results of the paper?

Minor comments

The western blot insert showing PARP2 silencing is missing from Fig 4B

Reviewer #2 (expert in heart physiology and genetics in invertebrates)

Remarks to the Author:

The manuscript by Zhang et al. report that tachypacing in cardiac-like HL-1 cell 'activates' PARP1 leading to NAD depletion that then causes, as the authors claim, an energy deficit, DNA damage and loss of contractile function. DNA damage was also found in AF patient atrial samples. The authors provide evidence that NAD supplement or PARP1 inhibition can provide some protection from the tachypacing-induced damage in HL-1 cells. The authors also provide some evidence that tachypacing-induced heart rate reduction in Drosophila may be due to NAD depletion. The idea that

PARP1 is involved in AF remodeling and cardiomyocyte damage is interesting but somewhat underdeveloped, as no mechanism is provided how lowering of NAD by presumably PARP1 activation by tachypacing causes heart damage, if PARP1 is really the culprit. Below are listed the main concerns.

1. The authors judge PARP 'activation' based solely on NAD levels without any direct activity measures. What about other consumers of NAD, or biosynthetic mechanisms of NAD? Could it be that under tachypacing stress there is less NAD synthesized, not just more consumed, if at all? Other measures of NAD synthesis and degradation need to be employed to make a clear point, especially since PARP size upon 'activation' is not modified. This far the evidence is circumstantial.
2. The experimental confirmation in *Drosophila* is a good idea as an *in vivo* model, but 1) the power of genetics was not exploited, such as genetic knockdown or overexpression experiments could have been instrumental to identify the genetic pathways/mechanisms involved, and 2) the critical factors associate with heart damage were not measured, such as contractility (anecdotal heart wall traces are unacceptable), CaT, etc. (in addition there are some issues with the measurements). Reduced heart rate is not a reliable measure of heart damage. Also, were there any arrhythmias in the fly heart as a consequence (this can be measured now, see Cammarato et al., 2014)?
3. Some of the critical experiments in HL-1 cell should also be confirmed with mature cardiomyocytes from mouse or rat, since the HL-1 cells are only marginally approximating actual cardiomyocytes.
4. Since the authors claim that PARP 'activation' is responsible for and inhibition can prevent tachypacing-induced damage, it would help this conclusion if the authors could show that PARP induction could mimic the damage, in cell and flies.
5. There are a number of issues with the consistency and validity of some of the data:
 - a. Line 196-8: "A gradual increase in PAR levels was observed upon tachypacing, which reached significance after 8 hours of tachypacing and remained increased afterwards (Figure 1a-d, Figure S1a), while PARP1 protein expression ('level' rather) was unchanged during tachypacing (Figure 1a, Figure S1b, c). This observation indicates that tachypacing induces PARP activation." This is an overstatement. All that this says is that PAR levels go up.
 - b. Fig. 2c-f: There is discrepancy between westerns and immunolabeled cells. First, cells are not shown at 4 and 8 hours, so statement (Four hours of tachypacing...) is wrong. Second, the 12h time point shows a 2-fold increase on a western, but a 10-fold or more increase in cells (number of cells overall level of staining - how was ROI determined?).
 - c. Fig. 3: It looks like restoring CaT is 10 times more sensitive to NAD than heart rate/ wall motion. How is that discrepancy explained? Also, from TP Ctl in c it is unlikely that a heart rate is reliably discernible, but in d there is no difference between ctl and 5mM NAD... This has to be addressed experimentally and better documentation of what is going on.
 - d. Fig. 3c,d: When, how and how long was NAD administered, and how long was the time between end of administration and heart measurements, same for a and b.?

e. Fig. 4: This should be confirmed in the *Drosophila* model, as this is the key experiment to show indeed PARP function matters. The drug experiments are also supportive but not as convincing in flies as a genetic knockdown/ overexpression, etc. would be.

Reviewer #3 (expert in cardiac metabolism and physiology)

Remarks to the Author:

This is a novel manuscript suggesting that the PARP1 DNA repair pathway leads to depletion of NAD and subsequent down-regulation of calcium transients that might play a role in atrial remodeling and fibrillation.

Critique

The experiments are largely performed in the HL-1 cell line. One limitation of this cell line is that it is an immortalized and constantly dividing, so that its susceptibility to DNA injury and need for repair might differ from adult cardiomyocytes. In addition, the comparison of unpaced cells to those paced at 5 Hz raises concerns about the relevance of the observations to physiological conditions of sinus vs. AF. The mouse atria beat at 400-600 BPM in sinus rhythm and the stress of going from sinus to AF might be less. A comparison to an intermediate pacing cycle length would provide important additional insight.

The human data add clinical relevance to the cell-based findings. However, the groups vary in that the non-AF patients have less mitral valve disease and likely have less hemodynamic stress and atrial remodeling, which constitutes a potentially important confounding factor that could be an important determinant of the changes observed. Data on LA size and pressure are lacking. LA remodeling in the absence of AF might also activate the PARP pathway.

Most atrial fibrillation originates in the left atrium and most of the human tissues sampled were from right atrial appendage. This is limitation. The manuscript also does not address potential differences between the LA and RA, and the relevance of the HL-1 cells to the LA.

The implications that NAD depletion leads to metabolic stress is not substantiated by data. There are no measurements of adenine nucleotides or creatine phosphate, or oxygen consumption to substantiate this hypothesis.

Finally, the *Drosophila* data are not presented clearly enough to be meaningful to a general audience. The figures need to be explained or deleted.

Reviewer #4 (expert in atrial fibrillation and cardiac electrophysiology)

Remarks to the Author:

This paper examines the effects of rapid pacing of HL-1 on excessive poly(ADP)-ribose polymerase 1 (PARP1) activity, DNA damage and nicotinamide adenine dinucleotide (NAD⁺). Adding NAD⁺ or inhibiting PARP1 activity precludes tachypacing-induced changes in Ca²⁺ transients and wall motion in HL-1 and prepupae *Drosophila* hearts. Cardiomyocytes of patients with persistent AF show significant DNA damage, which correlates with PARP1 activity. The authors conclude that “tachypacing impairs cardiomyocyte function and implicates PARP1 as a therapeutic target to preserve cardiomyocyte function in clinical AF”.

Assessment

The paper proposes an interesting hypothesis but key data are missing to support the conclusion and relevance of the studies. While the data HL-1 cells and prepupae *Drosophila* hearts are intriguing, the connection with AF in humans (and animal models) is absent. Moreover the experimental design ignores a number of critical features of AF (and its progression) making it impossible to endorse the strong conclusions of the paper.

Specific comments

1. The experimental design of HL-1 cell pacing appears to be a problem, although this is criticism is uncertain since there are absolutely no details on how the HL-1 studies were performed. From my reading of the paper, it seems that cells were either paced “rapidly” (at 5Hz) or not paced at all. So many questions arise. What is the intrinsic beating rate of the cells? How are the cells paced (voltage, duration, waveform etc)? What are the effects of electrical stimulation alone, which is not (itself) benign? Are all cells captured by pacing? More important possibly, mouse atrial cardiomyocytes (CMs) normally beat at 10Hz, so how can 5 Hz be considered “rapid pacing”? The

authors need to study various pacing rates and assess question of connectivity between CMs and cellular uniformity in their cultures.

2. AF is ultimately an electrical phenomenon and, while coincident changes in atrial function may also occur once rapid electrical activity is initiated, no electrical endpoints are examined in the paper. In this regard, it is critical for the authors to acknowledge and consider that tachypacing-mediated electrical changes, which appear from many previous studies to be essential for the establishment of persistent AF, are rapidly reversed in animal models and humans once sinus rhythm is restored. Yet, the authors argue and conclude that DNA damage, which I would think is fairly permanent, is a critical event in the march towards permanent AF. Therefore, it would seem inescapable to assume that the mechanism proposed by the authors is largely irreversible. Clearly, electrical measurements and assessments should be performed and discussed in the context of previous clinical and whole-body studies. More important, a critical set of studies is missing in the experimental design. The authors need to explore issues of reversibility in their model systems. How does function as well as various biochemical measures change once pacing is reduced or eliminated? If DNA damage is a critical factor, then the effects should be irreversible, which would put the findings at odds with previous EP studies.

3. Multiple cell types are involved in the atrial changes associated with AF. Directly related to the comments above, the authors ignore the fact that fibrosis and inflammatory cell infiltrates are also critical aspects of AF progression, with fibrosis (due to its irreversible nature) more likely to underlie the progression toward persistent AF. The results and mechanisms need to be extended to fibroblasts.

4. The underlying logic of the experimental design and the interpretation of the studies are quite confusing at times. For example, the authors assert that DNA damage induces the activation and expression of PARPs which consumes NAD⁺ leading to depleted energy and oxidative stress. Yet, Figure 7 summarizes studies that reverse the logic by showing that PARP inhibition prevents DNA damage and therefore PARP activation, creating a chicken and egg dilemma.

5. NAD⁺ is largely a mitochondrial compound. It would be helpful to measure NAD⁺ in mitochondria.

6. Details of how the Ca²⁺ transients were measured and analyzed is required. I am 110% sure that Ca²⁺-transient amplitude vary greatly from cell to cell and from culture to culture. Details of how this inherent variability is addressed are required. These types of studies MUST be done in a blinded manner.

We would like to thank the reviewers for their comments and feedback. We have adapted the manuscript accordingly as detailed below.

Entire report of Reviewer #1 (expert in PARPs)

Major comments

While the in vitro cell line data and the drosophila data are clear, it should be at least acknowledged that some sort of in vivo model of fibrillation (rodent or large animal) would be an additional, important connecting step prior to considering clinical translation. Of course, it would be even better if the authors could add some in vivo rodent or large animal data to the paper.

When considering translation (repurposing) of PARP inhibitors, the compound of choice is olaparib (already approved and in patients for ovarian cancer) as opposed to veliparib used here (which is in clinical trials but not approved). Would it be possible to add some olaparib data to some of the key results of the paper?

Minor comments

The western blot insert showing PARP2 silencing is missing from Fig 4B

Piont to point responses to Comments of Reviewer #1

Major comments

1. While the in vitro cell line data and the drosophila data are clear, it should be at least acknowledged that some sort of in vivo model of fibrillation (rodent or large animal) would be an additional, important connecting step prior to considering clinical translation. Of course, it would be even better if the authors could add some in vivo rodent or large animal data to the paper.

We agree with the reviewer and acknowledged that an *in vivo* animal model of atrial fibrillation would be an additionstep, we conducted key experiments in adult atrial cardiomyocytes isolated from rats. Rat atrial cardiomal step prior to considering clinical translation (Line 407-409, page 13). As a first translational yocytes were subjected to tachypacing with or without treatment of the PARP1 inhibitor ABT-888, followed by biochemical analyses (levels of PARylation, PARP1, γ H2AX, NAD⁺) and calcium transient (CaT) measurements. As found in HL-1 atrial cardiomyocytes, tachypacing of adult rat atrial cardiomyocytes induced DNA damage, PARP1 activation and consequently NAD⁺ depletion which resulted in CaT loss. Furthermore, PARP inhibition by ABT-888 fully protected against tachypacing-induced PAR induction, NAD⁺ depletion and CaT loss. These results demonstrate that tachypaced isolated adult atrial cardiomyocytes from rats show identical changes as originally reported for tachypaced HL-1 cardiomyocytes.

To answer the question whether DNA damage-mediated PARP1 activation is causative for the downstream NAD⁺ depletion and CaT loss, we initiated DNA damage by gamma irradiation of rat atrial cardiomyocytes and HL-1 cardiomyocytes. Indeed, irradiation invoked DNA damage-induced protein PARylation, indicating PARP1 activation, with subsequent NAD⁺ depletion and CaT loss. Again, findings in

rat atrial cardiomyocytes and HL-1 cardiomyocytes were identical. Collectively, these results confirm that DNA damage is an upstream trigger for contractile dysfunction in cardiomyocytes.

We added to the methods section (Line 81-109, page 4 and 5):

'Adult rat atrial cardiomyocyte model, calcium transient measurements and drug treatment

Adult Wistar rats (~ 200 g) were injected with heparin 15 min before atrial cardiomyocyte isolation, followed by anaesthesia (2% isoflurane and 98% O₂). Hearts were excised and placed in cool, oxygenated buffer solution containing (in mM) 134 NaCl, 10 HEPES, 4 KCl, 1.2 MgSO₄, 1.2 Na₂HPO₄, and 11 D-glucose (pH 7.4). Freshly excised rat hearts were mounted on a Langendorff setup and perfused retrogradely through the aorta with the buffer solution described above (37°C) containing 66.7 mg/L librase (Roche) for 30 min until the heart was soft. Following Langendorff perfusion, the atria were cut off the heart and rinsed in isolation solution (in mM): 100 NaCl, 5 Hepes, 20 D-glucose, 10 KCl, 5 MgSO₄, 1.2 KH₂PO₄, 50 Taurin, 0.5% BSA (pH 7.4), transferred into a 15 ml tube containing 10 ml isolation solution plus 0.02 mM CaCl₂ and 0.02 U/ml DNase, gently triturated for 7 min, and subsequently filtered through a 200 µm mesh filter into a 15 ml tube, followed by centrifugation for 1 min at 700 g. The supernatant was removed and the pellet containing atrial cardiomyocytes was resuspended carefully in 10 ml isolation solution plus 0.02 mM CaCl₂. Finally, the Ca²⁺ concentration was increased in incremental steps (adjustment time of 5 min between steps) from 0.1 mM, to 0.2 mM until 0.4 mM Ca²⁺. Atrial cardiomyocytes were left to sink for 20 min and transferred into laminin-coated plates in plating medium (M199 medium plus 5% fetal calf serum) for 2 h followed by replacement with M199 medium plus Insulin-Transferrin-Sodium Selenite Supplement (Sigma).

Prior to tachypacing, atrial cardiomyocytes were treated for 2 h with the PARP inhibitors ABT-888 (Selleckchem) or olaparib (Selleckchem), followed by tachypacing at 5 Hz, 30 V, 2 ms pulse duration for 2 h. Control atrial cardiomyocytes were either non-paced (NP) or paced at 1 Hz, 30 V, 2 ms pulse duration for 2 h. CaT measurement was performed according to previous studies with minor changes^{2,19}. In short, atrial cardiomyocytes were washed twice with M199 medium, incubated with calcium dye Fluo-4 (1 µg/ml) in M199 medium for 15 min, and rinsed twice again with M199 medium. The Fluo-4 loaded cardiomyocytes were excited at 488 nm and emitted at 500-550 nm and visually recorded with high speed confocal microscopy (Nikon A1R). Bright field settings were used to randomly select normal shaped cardiomyocytes, followed by a switch to the fluorescent filter to determine the CaT. As such, CaT measurements were conducted in a blinded manner.'

We added to the method section (Line166-172, page 6):

'Irradiation of cardiomyocytes

To induce DNA damage, HL-1 atrial cardiomyocytes received 10 Gy and rat atrial cardiomyocytes 40 Gy of irradiation with a dose rate of 0.0562 Gy/second by utilizing a cobalt-60 gamma-source (Gammacell 220 Research Irradiator, MDS Nordion, Canada). HL-1 and rat atrial cardiomyocytes were treated with 40 µM ABT-888 (12 h) or 5 µM ABT-888 (2 h), respectively, prior to the irradiation. After irradiation, cardiomyocytes were either prepared for Western blot analyses, NAD⁺ level or CaT measurements.'

We added to the results section (Line 307-310, page 10).

'In line with the findings in tachypaced HL-1 cardiomyocytes, tachypacing of isolated adult rat atrial cardiomyocytes significantly induced DNA damage and PAR levels, reduced NAD⁺ levels and resulted in

contractile dysfunction (Figure 7 and S2f-i). Importantly, all these effects were prevented by the PARP1 inhibitors ABT-888 and olaparib (Figure 7 and S6a, b).’

We added to the results section (Line 323-332, page 11).

‘DNA damage-mediated PARP activation is the cause of NAD⁺ depletion and contractile dysfunction in cardiomyocytes

To study whether PARP activation is the cause of NAD⁺ depletion and contractile dysfunction in cardiomyocytes, we gamma-irradiated cardiomyocytes to induce DNA damage and thereby PARP activation. As expected, irradiation resulted in a significant induction of DNA damage and consequently an increase in PAR levels, reduction in NAD⁺ levels, and finally loss in CaT in both HL-1 and rat atrial cardiomyocytes (Figure 9 and 10). The PARP1 inhibitor ABT-888 prevented the increase in PAR levels, NAD⁺ depletion and CaT loss (Figure 9 and 10). These findings confirm that DNA damage-mediated PARP activation is the cause of NAD⁺ depletion and calcium transient remodeling in atrial cardiomyocytes.

2. When considering translation (repurposing) of PARP inhibitors, the compound of choice is olaparib (already approved and in patients for ovarian cancer) as opposed to veliparib used here (which is in clinical trials but not approved). Would it be possible to add some olaparib data to some of the key results of the paper?

We fully agree with the reviewer and added findings from tachypaced HL-1 cardiomyocytes and *Drosophila* prepupae treated with olaparib. Comparable to ABT-888, olaparib significantly protected against tachypacing-induced functional impairment (loss of CaT and heart wall function) compared to non-treated cardiomyocytes or prepupae.

We added the olaparib data to the results section (Line 289-299, Line 307-310, page 10):

‘Inhibition of PARP1 prevents NAD⁺ depletion and contractile dysfunction in atrial cardiomyocytes and *Drosophila*

To further substantiate that PARP1 represents a drug target to mitigate tachypacing-induced functional remodeling, the action of PARP1 inhibitors was examined in HL-1 cardiomyocytes. PARP1 inhibitors comprised the general inhibitors, nicotinamide and 3-AB, and the specific PARP1/2 inhibitors ABT-888 and olaparib. Both general and specific inhibition of PARP1/2 precluded tachypacing-induced PARylation of proteins and decrease in NAD⁺ levels (Figure 5a, b, and Figure S5). Furthermore, the PARP1 inhibitors ABT-888 and olaparib also significantly attenuated tachypacing-induced contractile dysfunction in HL-1 cardiomyocytes and *Drosophila* without influencing the contractile function in cardiomyocytes paced at normal rate (Figure 6a-g and Figure S3, Figure S6c-f), as previously observed for nicotinamide⁸.

.. ‘In line with the findings in tachypaced HL-1 cardiomyocytes, tachypacing of isolated adult rat atrial cardiomyocytes significantly induced DNA damage and PAR levels, reduced NAD⁺ levels and resulted in contractile dysfunction (Figure 7 and S2f-i). Importantly, all these effects were prevented by the PARP1 inhibitors ABT-888 and olaparib (Figure 7 and S6a, b).’

We added to the discussion section (Line 402-406, page 13):

‘However, recently developed PARP inhibitors, such as ABT-888 and olaparib, exhibit increased potency and specificity relative to earlier inhibitors. ABT-888 directly inhibits PARP1 and PARP2 without an action on sirtuins⁴¹. ABT-888 is currently in phase I and II clinical studies in cancer⁴². In addition to ABT-888,

olaparib may represent a suitable candidate. Olaparib is used in phase III clinical trials for the treatment of metastatic breast cancers and has no effect on QT/QTc interval^{43 44}.

3. Minor comment

The western blot insert showing PARP2 silencing is missing from Fig 4B

We apologize for the misunderstanding. In the new version of the manuscript, we moved figure 4B to the supplemental data section as Figure S4b and clarified the legend of Figure S4 (Line 614-619, page 31) by adding:

'Figure S4: PARP1 knockdown in HL-1 cardiomyocytes and *Drosophila* a) Representative Western blot showing significant knockdown of PARP1 in HL-1 cardiomyocytes transfected with PARP1 siRNA (PARP1i) compared to HL-1 cardiomyocytes transfected with scrambled siRNA (CTL). ** $P < 0.01$ vs CTL. b) Quantified qPCR data showing significant knockdown of PARP2 in HL-1 cardiomyocytes transfected with PARP2 siRNA (PARP2i) compared to HL-1 cardiomyocytes transfected with scrambled siRNA (CTL). ** $P < 0.01$ vs CTL.'

Entire report of Reviewer #2 (expert in heart physiology and genetics in invertebrates)

The manuscript by Zhang et al. report that tachypacing in cardiac-like HL-1 cell 'activates' PARP1 leading to NAD depletion that then causes, as the authors claim, an energy deficit, DNA damage and loss of contractile function. DNA damage was also found in AF patient atrial samples. The authors provide evidence that NAD supplement or PARP1 inhibition can provide some protection from the tachypacing-induced damage in HL-1 cells. The authors also provide some evidence that tachypacing-induced heart rate reduction in Drosophila may be due to NAD depletion. The idea that PARP1 is involved in AF remodeling and cardiomyocyte damage is interesting but somewhat underdeveloped, as no mechanism is provided how lowering of NAD by presumably PARP1 activation by tachypacing causes heart damage, if PARP1 is really the culprit. Below are listed the main concerns.

- 1. The authors judge PARP 'activation' based solely on NAD levels without any direct activity measures. What about other consumers of NAD, or biosynthetic mechanisms of NAD? Could it be that under tachypacing stress there is less NAD synthesized, not just more consumed, if at all? Other measures of NAD synthesis and degradation need to be employed to make a clear point, especially since PARP size upon 'activation' is not modified. This far the evidence is circumstantial.*
- 2. The experimental confirmation in Drosophila is a good idea as an in vivo model, but 1) the power of genetics was not exploited, such as genetic knockdown or overexpression experiments could have been instrumental to identify the genetic pathways/mechanisms involved, and 2) the critical factors associate with heart damage were not measured, such as contractility (anecdotal heart wall traces are unacceptable), CaT, etc. (in addition there are some issues with the measurements). Reduced heart rate is not a reliable measure of heart damage. Also, were there any arrhythmias in the fly heart as a consequence (this can be measured now, see Cammarato et al., 2014)?*
- 3. Some of the critical experiments in HL-1 cell should also be confirmed with mature cardiomyocytes from mouse or rat, since the HL-1 cells are only marginally approximating actual cardiomyocytes.*
- 4. Since the authors claim that PARP 'activation' is responsible for and inhibition can prevent tachypacing-induced damage, it would help this conclusion if the authors could show that PARP induction could mimic the damage, in cell and flies.*
- 5. There are a number of issues with the consistency and validity of some of the data:*
 - a. Line 196-8: "A gradual increase in PAR levels was observed upon tachypacing, which reached significance after 8 hours of tachypacing and remained increased afterwards (Figure 1a-d, Figure S1a), while PARP1 protein expression ('level' rather) was unchanged during tachypacing (Figure 1a, Figure S1b, c). This observation indicates that tachypacing induces PARP activation." This is an overstatement. All that this says is that PAR levels go up.*
 - b. Fig. 2c-f: There is discrepancy between westerns and immunolabeled cells. First, cells are not shown at 4 and 8 hours, so statement (Four hours of tachypacing...) is wrong. Second, the 12h time point shows a 2-fold increase on a western, but a 10-fold or more increase in cells (number of cells overall level of staining - how was ROI determined?).*
 - c. Fig. 3: It looks like restoring CaT is 10 times more sensitive to NAD than heart rate/ wall motion. How is that discrepancy explained? Also, from TP Ctl in c it is unlikely that a heart rate is reliably discernible, but in d there is no difference between ctl and 5mM NAD... This has to be addressed experimentally and better documentation of what is going on.*

d. Fig. 3c,d: When, how and how long was NAD administered, and how long was the time between end of administration and heart measurements, same for a and b.?

e. Fig. 4: This should be confirmed in the Drosophila model, as this is the key experiment to show indeed PARP function matters. The drug experiments are also supportive but not as convincing in flies as a genetic knockdown/ overexpression, etc. would be.

Piont to point responses to comments of reviewer 2

1. The authors judge PARP 'activation' based solely on NAD levels without any direct activity measures. What about other consumers of NAD, or biosynthetic mechanisms of NAD? Could it be that under tachypacing stress there is less NAD synthesized, not just more consumed, if at all? Other measures of NAD synthesis and degradation need to be employed to make a clear point, especially since PARP size upon 'activation' is not modified. This far the evidence is circumstantial.

The main substantiation of enhanced enzymatic activity of PARP was obtained by quantification of the protein polyADP-ribosylation (i.e. PARylation) of PARP to itself and other proteins, as described before (Fang EF, et al. Cell, 2014). Notably, we demonstrated tachypacing to induce protein PARylation in HL-1 and rat atrial cardiomyocytes, which is inhibited by PARP inhibitors (Figure 5a and Figure 7a, b). Given the sole dependence of PARylation on PARP activity, these analyses substantiate enhanced enzymatic activity.

To study whether PARP activation is the main or sole driver of enhanced NAD⁺ consumption, we employed different PARP inhibitors (3-AB, ABT-888 and olaparib). Clearly, these experiments show that tachypacing of cardiomyocytes results in inhibition of PARP (evidenced by absence of PARylation) and consequently conservation of NAD⁺ levels (Figure 5a, b and Figure S5). This result implies that depletion of NAD⁺ is for the vast majority (if not totally) due to enhanced PARP activation/PARylation. Consequently, additional analysis of NAD⁺ flux is not expected to provide important additional insights into the mechanism.

Nevertheless, we agree with the reviewer that phrasing of the above reasoning should be improved. Therefore, we changed the results section (Line 247-249, page 9):

'As nicotinamide is also known to inhibit the activation of PARP^{11, 12}, we tested the level of PARP activity by measuring the amount of PAR synthesis in normal and tachypaced cardiomyocytes.'

Line 251-253, page 9:

'This observation indicates that tachypacing induces PAR synthesis, suggesting induction of PARP activation.'

2. The experimental confirmation in Drosophila is a good idea as an in vivo model, but 1) the power of genetics was not exploited, such as genetic knockdown or overexpression experiments could have been instrumental to identify the genetic pathways/mechanisms involved, and 2) the critical factors associate with heart damage were not measured, such as contractility (anecdotal heart wall traces are unacceptable), CaT, etc. (in addition there are some issues with the measurements). Reduced heart rate is not a reliable measure of heart damage. Also, were there any arrhythmias in the fly heart as a consequence (this can be measured now, see Cammarato et al., 2014)?

In reply to these comments, we sought to exploit the power of genetics in the *Drosophila* model by creating cardiac PARP1 knockout *Drosophila* strains by crossing Hand-GAL4 strains with UAS siRNA of PARP1 and confirmed suppression of PARP1 by qPCR and Western blot (Figure S4c, d). Furthermore, in addition to heart rate, we also included the arrhythmicity index as a determinant of heart function²¹⁻²³. Using this model, we now show tachypacing to reduce heart rate and increase arrhythmicity index in *Drosophila* prepupae, which was prevented by genetic suppression of PARP1 (Figure 4c-e, Figure S4e). We added these findings to the manuscript.

Methods section (Line 115-122, page 5):

'To create the knockdown of PARP1 in *Drosophila*, PARP1 UAS-RNAi *Drosophila* from the Vienna *Drosophila* RNAi Center, were crossed with a Hand-GAL4 driver strain (kind gift from Prof. Dr. Achim Paululat)²⁰. As control, wild-type *Drosophila* W1118 were crossed with Hand-GAL4 driver *Drosophila*. Prepupae of F1 offspring were tachypaced as previously described⁸. Heart wall contractions were measured utilizing high-speed digital video imaging (100 frames/second) before and after tachypacing, followed by the generation of heart wall traces. Traces were used to determine cardiac parameters including heart rate and arrhythmicity index (defined as the standard deviation of the heart period)²¹⁻²³.'

Results section (Line 282-286, page 10):

'To confirm that PARP1 is the key PARP enzyme driving tachypacing-induced contractile dysfunction, PARP1 expression was suppressed specifically in the heart of *Drosophila*, as confirmed by Western blotting and qPCR (Figure S4c, d). In line with the findings in HL-1 cardiomyocytes, suppression of PARP1 resulted in protection against tachypacing-induced heart wall dysfunction (Figure 4c-e, Figure S4e).'

3. Some of the critical experiments in HL-1 cell should also be confirmed with mature cardiomyocytes from mouse or rat, since the HL-1 cells are only marginally approximating actual cardiomyocytes.

In response to this comment, we tested PARP1 activation, NAD⁺ depletion and functional endpoints in tachypaced adult atrial cardiomyocytes isolated from rats. Moreover, we investigated protective effects of the PARP1 inhibitors (ABT-888) in rat atrial cardiomyocytes. As found in HL-1 atrial cardiomyocytes, tachypacing of adult rat atrial cardiomyocytes induced DNA damage, PARP1 activation and consequently NAD⁺ depletion which resulted in CaT loss. Furthermore, PARP inhibition by ABT-888 fully protected against tachypacing-induced PAR induction, NAD⁺ depletion and CaT loss. These results demonstrate that tachypaced isolated adult atrial cardiomyocytes from rats show identical changes as originally reported for tachypaced HL-1 cardiomyocytes.

We added to the methods section (Line 81-109, page 4 and 5):

Adult rat atrial cardiomyocyte model, calcium transient measurements and drug treatment

Adult Wistar rats (~ 200 g) were injected with heparin 15 min before atrial cardiomyocyte isolation, followed by anaesthetisation (2% isoflurane and 98% O₂). Hearts were excised and placed in cool, oxygenated buffer solution containing (in mM) 134 NaCl, 10 HEPES, 4 KCl, 1.2 MgSO₄, 1.2 Na₂HPO₄, and 11 D-glucose (pH 7.4). Freshly excised rat hearts were mounted on a Langendorff setup and perfused retrogradely through the aorta with the buffer solution described above (37°C) containing 66.7 mg/L librase (Roche) for 30 min until the heart was soft. Following Langendorff perfusion, the atria were cut off the heart and rinsed in isolation solution (in mM): 100 NaCl, 5 Hepes, 20 D-glucose, 10 KCl, 5 MgSO₄, 1.2 KH₂PO₄, 50 Taurin, 0.5% BSA (pH 7.4), transferred into a 15 ml tube containing 10 ml isolation solution plus 0.02 mM CaCl₂ and 0.02 U/ml DNase, gently triturated for 7 min, and subsequently filtered through

a 200 µm mesh filter into a 15 ml tube, followed by centrifugation for 1 min at 700 g. The supernatant was removed and the pellet containing atrial cardiomyocytes was resuspended carefully in 10 ml isolation solution plus 0.02 mM CaCl₂. Finally, the Ca²⁺ concentration was increased in incremental steps (adjustment time of 5 min between steps) from 0.1 mM, to 0.2 mM until 0.4 mM Ca²⁺. Atrial cardiomyocytes were left to sink for 20 min and transferred into laminin-coated plates in plating medium (M199 medium plus 5% fetal calf serum) for 2 h followed by replacement with M199 medium plus Insulin-Transferrin-Sodium Selenite Supplement (Sigma).

Prior to tachypacing, atrial cardiomyocytes were treated for 2 h with the PARP inhibitors ABT-888 (Selleckchem) or olaparib (Selleckchem), followed by tachypacing at 5 Hz, 30 V, 2 ms pulse duration for 2 h. Control atrial cardiomyocytes were either non-paced (NP) or paced at 1 Hz, 30 V, 2 ms pulse duration for 2 h. CaT measurement was performed according to previous studies with minor changes^{2,19}. In short, atrial cardiomyocytes were washed twice with M199 medium, incubated with calcium dye Fluo-4 (1 µg/ml) in M199 medium for 15 min, and rinsed twice again with M199 medium. The Fluo-4 loaded cardiomyocytes were excited at 488 nm and emitted at 500-550 nm and visually recorded with high speed confocal microscopy (Nikon A1R). Bright field settings were used to randomly select normal shaped cardiomyocytes, followed by a switch to the fluorescent filter to determine the CaT. As such, CaT measurements were conducted in a blinded manner.'

We added to the results section (Line 307-310, page 10).

'In line with the findings in tachypaced HL-1 cardiomyocytes, tachypacing of isolated adult rat atrial cardiomyocytes significantly induced DNA damage and PAR levels, reduced NAD⁺ levels and resulted in contractile dysfunction (Figure 7 and S2f-i). Importantly, all these effects were prevented by the PARP1 inhibitors ABT-888 and olaparib (Figure 7 and S6a, b).'

4. Since the authors claim that PARP 'activation' is responsible for and inhibition can prevent tachypacing-induced damage, it would help this conclusion if the authors could show that PARP induction could mimic the damage in cells and flies.

This is an interesting comment. PARP is activated by single and double strand breaks in the DNA. Therefore, we approached this question by reasoning that proof-of-concept for damaging downstream effects of PARP activation in cardiomyocytes can be obtained by inducing DNA breaks in its purest form. Hereto, we subjected rat atrial and HL-1 cardiomyocytes to brief (gamma) irradiation, which results in single- and double-strand DNA breaks and activation of PARP1 (induction of PARylation). Our results unequivocally show irradiation to induce DNA damage and PARP1 activation, which is accompanied by NAD⁺ depletion and loss in CaT, both in HL-1 and rat atrial cardiomyocytes. Importantly, as observed in tachypaced cardiomyocytes, all changes were prevented by ABT-888 treatment, substantiating the key role of PARP1 activation in cardiomyocyte damage.

We added these findings to the methods section (Line 166-172, page 6):

'Irradiation of cardiomyocytes

To induce DNA damage, HL-1 atrial cardiomyocytes received 10 Gy and rat atrial cardiomyocytes 40 Gy of irradiation with a dose rate of 0.0562 Gy/second by utilizing a cobalt-60 gamma-source (Gammacell 220 Research Irradiator, MDS Nordion, Canada). HL-1 and rat atrial cardiomyocytes were treated with 40 µM ABT-888 (12 h) or 5 µM ABT-888 (2 h), respectively, prior to the irradiation. After irradiation, cardiomyocytes were either prepared for Western blot analyses, NAD⁺ level or CaT measurements.'

We added to the results section (Line 323-332, page 11).

'DNA damage-mediated PARP activation is the cause of NAD⁺ depletion and contractile dysfunction in cardiomyocytes

'To study whether PARP activation is the cause of NAD⁺ depletion and contractile dysfunction in cardiomyocytes, we gamma-irradiated cardiomyocytes to induce DNA damage and thereby PARP activation. As expected, irradiation resulted in a significant induction of DNA damage and consequently an increase in PAR levels, reduction in NAD⁺ levels, and finally loss in CaT in both HL-1 and rat atrial cardiomyocytes (Figure 9 and 10). The PARP1 inhibitor ABT-888 prevented the increase in PAR levels, NAD⁺ depletion and CaT loss (Figure 9 and 10). These findings confirm that DNA damage-mediated PARP activation is the cause of NAD⁺ depletion and calcium transient remodeling in atrial cardiomyocytes.'

5. There are a number of issues with the consistency and validity of some of the data:

a. Line 196-8: *"A gradual increase in PAR levels was observed upon tachypacing, which reached significance after 8 hours of tachypacing and remained increased afterwards (Figure 1a-d, Figure S1a), while PARP1 protein expression ('level' rather) was unchanged during tachypacing (Figure 1a, Figure S1b, c). This observation indicates that tachypacing induces PARP activation."* This is an overstatement. All that this says is that PAR levels go up.

Thank you for this remark. We changed the sentence into: 'This observation indicates that tachypacing induces PAR synthesis, suggesting induction of PARP activation.' (Line 251-253, page 9).

b. Fig. 2c-f: *There is discrepancy between westerns and immunolabeled cells. First, cells are not shown at 4 and 8 hours, so statement (Four hours of tachypacing...) is wrong. Second, the 12h time point shows a 2-fold increase on a western, but a 10-fold or more increase in cells (number of cells overall level of staining - how was ROI determined?).*

The Western blot of γ H2AX was performed with protein samples isolated at different time-points (4, 8 and 12 hours after tachypacing). The immunostaining of γ H2AX was used as an additional method to confirm the Western blot findings, therefore we only included the 12 hours endpoint and observed comparable findings, showing that tachypacing induces DNA damage.

ROI for the immunostaining was determined for the whole area of the picture. We quantified total red signal, which reflects the amount of γ H2AX (TRITC-labeled 2nd antibody) and divided this signal by the blue signal (DAPI), which indicates the cell number. The reason of the 2-fold and 10-fold difference is caused by the difference in quantification method (Western blot vs immunostaining).

We amended the method section (Line 208-212, page 7):

'For quantification, Image Pro software was used to calculate the total fluorescent (green for FITC and red for TRITC) signal per image and the DAPI signal. The total fluorescent signal, which indicates the expression of PARP1, PAR or γ H2AX, was divided by the respective blue signal (DAPI), which indicates the cell number.'

c. Fig. 3: *It looks like restoring CaT is 10 times more sensitive to NAD than heart rate/ wall motion. How is that discrepancy explained? Also, from TP Ctl in c it is unlikely that a heart rate is reliably discernible, but in d there is no difference between ctl and 5mM NAD... This has to be addressed experimentally and better documentation of what is going on.*

Indeed, our data suggest a difference in efficacy of NAD⁺ in HL-1 cardiomyocytes and *Drosophila*. However, the concentration difference reflects the concentration of NAD⁺ in the medium of the cardiomyocytes, and the concentration in food of *Drosophila*, respectively. In *Drosophila*, the compound was dissolved in 0.5 ml water and added to the food. Based on our previous experience with multiple drug screening assays, including dose-dependent testing, we have to apply 10x the effective dose found in cultured cardiomyocytes to obtain a comparable (protective) effect in *Drosophila* (Zhang D, et al. *Circulation*, 2014; Hoogstra-Berends F, et al. *Trends Cardiovasc Med*, 2012; Wiersma M, et al. *JAHA* 2017). To improve assessment of cardiac function in *Drosophila*, we now included the arrhythmicity index as an endpoint to determine the effect of drugs. ABT-888 and olaparib were found to protect against tachypacing-induced arrhythmicity in wildtype *Drosophila* strains. Furthermore, the quality of the *Drosophila* heart wall tracings was improved (Figure 3c).

We amended the methods section accordingly (Line 118-122, page 5):

‘Heart wall contractions were measured utilizing high-speed digital video imaging (100 frames/second) before and after tachypacing, followed by the generation of heart wall traces. Traces were used to determine cardiac parameters including heart rate and arrhythmicity index (defined as the standard deviation of the heart period)²¹⁻²³.’

d. Fig. 3c,d: When, how and how long was NAD administered, and how long was the time between end of administration and heart measurements, same for a and b.?

Cardiomyocytes were exposed to NAD⁺ for 24 hours and *Drosophila* for at least 48 hours.

We added detailed information to method section:

HL-1 cardiomyocyte model, calcium transient measurements and drug treatment (Line 76-78, page 4)

‘Prior to 12 h tachypacing, HL-1 cardiomyocytes were treated for 12 h with the PARP inhibitors 3-aminobenzamide (3-AB, Sigma-Aldrich), ABT-888 (Selleckchem), olaparib (Selleckchem), beta-nicotinamide adenine dinucleotide hydrate (NAD⁺, Sigma-Aldrich)...’

***Drosophila* stocks, tachypacing, and heart wall contraction assays** (Line 110-114, page 5)

‘The wild-type W1118 strain was used for all drug screening experiments. The *Drosophila* prepupae were pretreated with PARP inhibitors or NAD⁺ for 48 h, followed by subjection to tachypacing for 20 min (4 Hz, 20 V, pulse duration of 5 ms) and heart wall contractions were measured as previously described⁸. See Table 1 for the applied doses of 3-AB, ABT-888 and NAD⁺.’

e. Fig. 4: This should be confirmed in the Drosophila model, as this is the key experiment to show indeed PARP function matters. The drug experiments are also supportive but not as convincing in flies as a genetic knockdown/ overexpression, etc. would be.

Thank you for this suggestion. We agree and conducted experiments by utilizing siRNA for PARP1 as described above (comment 2).

Entire report of Reviewer #3 (expert in cardiac metabolism and physiology)

This is a novel manuscript suggesting that the PARP1 DNA repair pathway leads to depletion of NAD and subsequent down-regulation of calcium transients that might play a role in atrial remodeling and fibrillation.

Critique

The experiments are largely performed in the HL-1 cell line. One limitation of this cell line is that it is an immortalized and constantly dividing, so that its susceptibility to DNA injury and need for repair might differ from adult cardiomyocytes. In addition, the comparison of unpaced cells to those paced at 5 Hz raises concerns about the relevance of the observations to physiological conditions of sinus vs. AF. The mouse atria beat at 400-600 BPM in sinus rhythm and the stress of going from sinus to AF might be less. A comparison to an intermediate pacing cycle length would provide important additional insight.

The human data add clinical relevance to the cell-based findings. However, the groups vary in that the non-AF patients have less mitral valve disease and likely have less hemodynamic stress and atrial remodeling, which constitutes a potentially important confounding factor that could be an important determinant of the changes observed. Data on LA size and pressure are lacking. LA remodeling in the absence of AF might also activate the PARP pathway.

Most atrial fibrillation originates in the left atrium and most of the human tissues sampled were from right atrial appendage. This is limitation. The manuscript also does not address potential differences between the LA and RA, and the relevance of the HL-1 cells to the LA.

The implications that NAD depletion leads to metabolic stress is not substantiated by data. There are no measurements of adenine nucleotides or creatine phosphate, or oxygen consumption to substantiate this hypothesis.

Finally, the Drosophila data are not presented clearly enough to be meaningful to a general audience. The figures need to be explained or deleted.

Point to point response to Comments of Reviewer #3

1. The experiments are largely performed in the HL-1 cell line. One limitation of this cell line is that it is an immortalized and constantly dividing, so that its susceptibility to DNA injury and need for repair might differ from adult cardiomyocytes. In addition, the comparison of unpaced cells to those paced at 5 Hz raises concerns about the relevance of the observations to physiological conditions of sinus vs. AF. The mouse atria beat at 400-600 BPM in sinus rhythm and the stress of going from sinus to AF might be less. A comparison to an intermediate pacing cycle length would provide important additional insight.

Thank you for your comment. Apparently, we did not articulate the nature of normal pacing adequately. We observed that both cultured HL-1 cell line and isolated adult rat atrial cardiomyocytes have a spontaneous activation rate of 0.5-1 Hz. We performed experiments with normal pacing (1 Hz), tachypacing (5 Hz), and non-pacing (0 Hz). No difference was found between non-pacing and normal pacing regarding all the endpoints, including DNA damage, PARP activation, NAD⁺ depletion and contractile function. We choose to include non-paced cardiomyocytes as a control.

We added the findings to supplemental data section figures S2 (Line 596-604, page 29).

We added the findings in the results section (Line 262-264, page 9):

‘Eight hours of tachypacing induced a significant reduction in NAD⁺ levels (Figure 2g). Normal pacing at 1 Hz did not reveal changes at PAR, γ H2AX, or NAD⁺ levels (Figure S2a-e).’

2. The human data add clinical relevance to the cell-based findings. However, the groups vary in that the non-AF patients have less mitral valve disease and likely have less hemodynamic stress and atrial remodeling, which constitutes a potentially important confounding factor that could be an important determinant of the changes observed.

We agree with the reviewer that the underlying heart disease may represent a confounding factor and that patients with mitral valve disease may have excess arrhythmogenic substrate compared to coronary artery disease patients due to atrial dilation. However, we recently reported a comparable degree of electrical remodeling in 253 patients with and without valvular heart disease as assessed by high resolution mapping of the entire epicardial surface (Mouws AE et al, Heart Rhythm, 2017; Mouws AE, Circ AE, 2017). Moreover, both studies show that the extent of electrical remodeling of an individual patient was similar in both left (dilated) and right atrium. These studies question the impact of atrial dilation on cardiomyocyte remodeling, and signify that the observed differences in DNA damage between AF and non-AF patients are the result of the presence of (long-standing) persistent AF and not due to a confounding factor. Findings of these studies are in line with the Western blot data presented in Figure 11a, b (page 27). The data show increased PAR levels in RAA and LAA of patients with persistent AF compared to sinus rhythm, all with mitral valve disease as a common underlying heart disease.

To stress that mitral valve disease was not a confounding factor in our experiments we added to the legend of Figure 11 (Line 575-576): ‘a) Representative Western blots of PAR and PARP1 levels in RAA and LAA of SR and AF patients with underlying mitral valve disease, showing significant increase in PAR levels in AF patients compared to SR patients.’

Data on LA size and pressure are lacking. LA remodeling in the absence of AF might also activate the PARP pathway.

We did not register intra-operative LA pressure as this parameter is continuously changing during the procedure due to slight manipulation of the heart, acute changes in circulating volume or vasopressive drugs and it therefore does not reliably reflect real-life intra-atrial pressure and resulting stretch of the atrial wall. All patients have a certain degree of LA remodeling due to e.g. aging or underlying heart disease. Therefore, the observed presence of DNA damage and PARP activation in both RAA and LAA of patients with persistent AF is unlikely to be caused by LA remodeling due to underlying heart diseases. As mentioned above, Figure 11a and b reveals that the induction of PAR as observed in patients with persistent AF is due to AF and not LA dilatation as all presented patients have mitral valve disease as an underlying heart disease.

Most atrial fibrillation originates in the left atrium and most of the human tissues sampled were from right atrial appendage. This is limitation.

In patients with AF, the triggers originate mainly from the left atrium, yet the arrhythmogenic substrate (structural remodeling) is found in both atria. Prior mapping studies showed that the severity of the arrhythmogenic substrate in patients with persistent AF is comparable between the right and left atria and thus not confined to the left atrium (Allessie, Circ AE 2010; N de Groot Circ 2010). Furthermore, no differences in PAR levels between LAA and RAA were observed (Figure 11a, b), indicating that RAA and LAA findings are comparable.

The manuscript also does not address potential differences between the LA and RA, and the relevance of the HL-1 cells to the LA.

As explained above, we observed by using high-resolution mapping studies that the degree of electrical remodeling of the RAA and LAA are comparable. This is in line with findings from LAA and RAA, showing comparable induction in PAR levels in right and left atrial tissue of patients with persistent AF. Therefore, tachypaced HL-1 and rat (left and right) atrial cardiomyocytes represent findings observed in both LAA and RAA of persistent AF patients.

The implications that NAD depletion leads to metabolic stress is not substantiated by data. There are no measurements of adenine nucleotides or creatine phosphate, or oxygen consumption to substantiate this hypothesis.

The main message of our paper is that AF is associated with DNA damage and downstream PARP activation, NAD⁺ depletion and contractile dysfunction. As other studies revealed that NAD⁺ is important in metabolic stress, we mentioned this relation in the discussion section. To avoid misinterpretation, we removed the word 'metabolic' from the discussion section (Line 374 on page 12).

Finally, the Drosophila data are not presented clearly enough to be meaningful to a general audience. The figures need to be explained or deleted.

We improved the presentation of the *Drosophila* data by amending the methods and Legend section as suggested by reviewer 2:

Methods section (Line 115-122, page 5):

'To create the knockdown of PARP1 in *Drosophila*, PARP1 UAS-RNAi *Drosophila* from the Vienna *Drosophila* RNAi Center, were crossed to a Hand-GAL4 driver strain (kind gift from Prof. Dr. Achim Paululat)²⁰. As control, wild-type *Drosophila* W1118 were crossed with Hand-GAL4 driver *Drosophila*. Prepupae of F1 offspring were tachypaced as previously described⁸. Heart wall contractions were measured utilizing high-speed digital video imaging (100 frames/second) before and after tachypacing, followed by the generation of heart wall traces. Traces were used to determine cardiac parameters including heart rate and arrhythmicity index (defined as the standard deviation of the heart period)²¹⁻²³.'

Legend section:

Figure 4 (Line 490-494, page 20): 'c) Representative traces (10 seconds) prepared from high-speed movies of *Drosophila* prepupae. Movies were made from control non-paced (NP) and tachypaced (TP) *Drosophila* prepupae in wildtype (WT) and PARP1 knockdown (PARP1i) strains. d) and e) Quantified heart rate (bpm: beats per minute) and arrhythmicity index (AI) in milliseconds (ms).'

Figure 6 (Line 513-515, page 22): 'e)-g) Representative heart wall contraction measurements and quantified relative heart rate and arrhythmicity index (AI) of control NP or TP *Drosophila* pretreated with 3-AB (30 mM), ABT-888 (0.2 mM, 0.4 mM), or vehicle (CTL). ** $P < 0.01$ vs control NP, # $P < 0.05$ vs control TP, N=10 *Drosophila* prepupae for CTL, N=7 for 3-AB, N=6 for ABT-888 (0.2 mM), N=7 for ABT-888 (0.4mM). Representative heart wall contraction measurements and quantified relative heart rate and arrhythmicity index (AI) of control NP or TP *Drosophila* pretreated with 3-AB (30 mM), ABT-888 (0.2 mM, 0.4 mM), or vehicle (CTL).'

Figure S4 (Line 619-625, page 31): 'c) Representative Western blot showing significant knockdown of PARP1 in *Drosophila* PARP1i (Hand4-GAL4 crossed with UAS-PARP1 shRNA *Drosophila*) compared to wild-type (WT: Hand4-GAL4 crossed with wild-type W1118). * $P < 0.05$ vs WT, N=3 independent experiments. d) Quantified qPCR data showing significant knockdown of PARP1 in *Drosophila* PARP1i (compared to wild-type * $P < 0.05$ vs WT, N=3 independent experiments. e) Representative traces (10 seconds) prepared from high-speed (100 frames per seconds) movies of *Drosophila* prepupae. Movies were made in *Drosophila* prepupae before TP (NP) or after tachypacing (TP).'

Entire report of Reviewer #4 (expert in atrial fibrillation and cardiac electrophysiology)

This paper examines the effects of rapid pacing of HL-1 on excessive poly(ADP)-ribose polymerase 1 (PARP1) activity, DNA damage and nicotinamide adenine dinucleotide (NAD⁺). Adding NAD⁺ or inhibiting PARP1 activity precludes tachypacing-induced changes in Ca²⁺ transients and wall motion in HL-1 and prepupae Drosophila hearts. Cardiomyocytes of patients with persistent AF show significant DNA damage, which correlates with PARP1 activity. The authors conclude that “tachypacing impairs cardiomyocyte function and implicates PARP1 as a therapeutic target to preserve cardiomyocyte function in clinical AF”.

Assessment

The paper proposes an interesting hypothesis but key data re missing to support the conclusion and relevance of the studies. While the data HL-1 cells and prepupae Drosophila hearts are intriguing, the connection with AF in humans (and animal models) is absent. Moreover the experimental design ignores a number of critical features of AF (and its progression) making it impossible to endorse the strong conclusions of the paper.

Specific comments

1. The experimental design of HL-1 cell pacing appears to be a problem, although this is criticism is uncertain since there are absolutely no details on how the HL-1 studies were performed. From my reading of the paper, it seems that cells were either paced “rapidly” (at 5Hz) or not paced at all. So many questions arise. What is the intrinsic beating rate of the cells? How are the cells paced (voltage, duration, waveform etc)? What are the effects of electrical stimulation alone, which is not (itself) benign? Are all cells captured by pacing? More important possibly, mouse atrial cardiomyocytes (CMs) normally beat at 10Hz, so how can 5 Hz be consider “rapid pacing”? The authors need to study various pacing rates and assess question of connectivity between CMs and cellular uniformity in their cultures.

2. AF is ultimately an electrical phenomenon and, while coincident changes in atrial function may also occur once rapid electrical activity is initiated, no electrical endpoints are examined in the paper. In this regard, it is critical for the authors to acknowledge and consider that tachypacing-mediated electrical changes, which appear from many previous studies to be essential for the establishment of persistent AF, are rapidly reversed in animal models and humans once sinus rhythm is restored. Yet, the authors argue and conclude that DNA damage, which I would think is fairly permanent, is a critical event in the march towards permanent AF. Therefore, it would seem inescapable to assume that the mechanism proposed by the authors is largely irreversible. Clearly, electrical measurements and assessments should be performed and discussed in the context of previous clinical and whole-body studies. More important, a critical set of studies is missing in the experimental design. The authors need to explore issues of reversibility in their model systems. How does function as well as various biochemical measures change once pacing is reduced or eliminated? If DNA damage is a critical factor, then the effects should be irreversible, which would put the findings at odds with previous EP studies.

3. Multiple cell types are involved in the atrial changes associated with AF. Directly related to the comments above, the authors ignore the fact that fibrosis and inflammatory cell infiltrates are also critical

aspects of AF progression, with fibrosis (due to its irreversible nature) more likely to underlie the progression toward persistent AF. The results and mechanisms need to be extended to fibroblasts.

4. The underlying logic of the experimental design and the interpretation of the studies are quite confusing at times. For example, the authors assert that DNA damage induces the activation and expression of PARPs which consumes NAD⁺ leading to depleted energy and oxidative stress. Yet, Figure 7 summarizes studies that reverse the logic by showing that PARP inhibition prevents DNA damage and therefore PARP activation, creating a chicken and egg dilemma.

5. NAD⁺ is largely a mitochondrial compound. It would be helpful to measure NAD⁺ in mitochondria.

6. Details of how the Ca²⁺ transients were measured and analyzed is required. I am 110% sure that Ca²⁺ transient amplitude vary greatly from cell to cell and from culture to culture. Details of how this inherent variability is addressed are required. These types of studies MUST be done in a blinded manner.

Point to point response to Comments of Reviewer #4

Specific comments

1. The experimental design of HL-1 cell pacing appears to be a problem, although this is criticism is uncertain since there are absolutely no details on how the HL-1 studies were performed. From my reading of the paper, it seems that cells were either paced “rapidly” (at 5Hz) or not paced at all. So many questions arise. What is the intrinsic beating rate of the cells? How are the cells paced (voltage, duration, waveform etc)? What are the effects of electrical stimulation alone, which is not (itself) benign? Are all cells captured by pacing? More important possibly, mouse atrial cardiomyocytes (CMs) normally beat at 10Hz, so how can 5 Hz be consider “rapid pacing”? The authors need to study various pacing rates and assess question of connectivity between CMs and cellular uniformity in their cultures.

HL-1 cardiomyocytes were normal- or tachy-paced with use of the C-Pace EP cell stimulation system (IonOptix). The frequency of spontaneous beating of the cardiomyocytes is 0.5-1 Hz and non-paced and normal paced (1 Hz) conditions were used as controls (Figure S2). For tachypacing, we use 5 Hz pacing rate at 40 V and the waveform is bipolar. Duration of 1 electrical pulse is 20 ms. The adult rat atrial cardiomyocytes have a spontaneous beating rate of 0.5-1 Hz. Control rat cardiomyocytes were non-paced or normal paced at 1 Hz, 30 V, with pulse duration of 2 ms. For tachypacing, the rat cardiomyocytes were subjected to 5 Hz, 30 V and pulses of 2 ms. Upon tachypacing, both HL-1 and rat atrial cardiomyocytes capture the pacing rate. Importantly, normal pacing at 1 Hz did not result in changes in PARylation, NAD⁺ and CaT levels (see Supplemental Data Figure S2 on page 29).

We added details on the tachypacing protocol to the method section:

For HL-1 cardiomyocytes (Line 69-73, page 4):

‘HL-1 cardiomyocytes derived from adult mouse atria were obtained from Dr. William Claycomb (Louisiana State University, New Orleans) and cultured as previously described^{2, 19}. The cardiomyocytes were

tachypaced (TP, 5 Hz, 40 V, pulse duration of 20 ms) with a C-Pace100 culture pacer (IonOptix) for 12 h except specifically stated for time-course pacing. HL-1 cardiomyocytes followed the pacing rate.'

For isolated rat atrial cardiomyocytes (Line 99-102, page 4):

'Prior to tachypacing, atrial cardiomyocytes were treated for 2 h with the PARP inhibitors ABT-888 (Selleckchem), or olaparib (Selleckchem), followed by tachypacing at 5 Hz, 30 V and pulse duration of 2 ms for 2 h. Control atrial cardiomyocytes were either non-paced (NP) or paced at 1Hz, 30 V and pulse duration of 2 ms for 2 h. Atrial cardiomyocytes followed the pacing rate.'

2. AF is ultimately an electrical phenomenon and, while coincident changes in atrial function may also occur once rapid electrical activity is initiated, no electrical endpoints are examined in the paper. In this regard, it is critical for the authors to acknowledge and consider that tachypacing-mediated electrical changes, which appear from many previous studies to be essential for the establishment of persistent AF, are rapidly reversed in animal models and humans once sinus rhythm is restored. Yet, the authors argue and conclude that DNA damage, which I would think is fairly permanent, is a critical event in the march towards permanent AF. Therefore, it would seem inescapable to assume that the mechanism proposed by the authors is largely irreversible. Clearly, electrical measurements and assessments should be performed and discussed in the context of previous clinical and whole-body studies. More important, a critical set of studies is missing in the experimental design. The authors need to explore issues of reversibility in their model systems. How does function as well as various biochemical measures change once pacing is reduced or eliminated? If DNA damage is a critical factor, then the effects should be irreversible, which would put the findings at odds with previous EP studies.

Actually, DNA damage is not permanent and DNA repair, by activated PARP, has been recognized in the nineties and extensively studied since then (e.g. Friedberg EC, Nature, 2003). Here we show that NAD⁺ depletion due to upstream DNA damage-induced PARP1 activation is a critical factor in contractile dysfunction of atrial cardiomyocytes. As pharmacological inhibition of PARP1 protects against cardiomyocyte dysfunction, we may argue that PARP1 inhibition helps to restore from contractile dysfunction. Indeed, we observed that post-treatment of tachypaced HL-1 cardiomyocytes with ABT-888 accelerates recovery from CaT loss, NAD⁺ depletion and PAR induction, compared to non-treated HL-1 cardiomyocytes.

We added the data to the results section (Line 299-306, page 10):

'Since AF is a progressive disease, it is of interest to study whether PARP1 inhibition accelerates recovery from tachypacing-induced NAD⁺ depletion and contractile dysfunction. Hereto, HL-1 cardiomyocytes were tachypaced, followed by 24 h recovery under no pacing conditions. In vehicle treated cardiomyocytes, no recovery from tachypacing induced CaT loss, NAD⁺ depletion or increased PAR levels was observed. In contrast, tachypaced HL-1 cardiomyocytes post-treated with ABT-888 revealed accelerated recovery at all endpoints (Figure S7). These findings demonstrate that PARP1 inhibitors not only prevent PARP1 activation, NAD⁺ depletion and CaT loss, but also accelerate their recovery after cessation of tachypacing.'

3. Multiple cell types are involved in the atrial changes associated with AF. Directly related to the comments above, the authors ignore the fact that fibrosis and inflammatory cell infiltrates are also critical aspects of AF progression, with fibrosis (due to its irreversible nature) more likely to underlie the progression toward persistent AF. The results and mechanisms need to be extended to fibroblasts.

As mentioned by this reviewer (major comment 2), AF is an electrical phenomenon and therefore we focused on the role of electrical stimulation-induced molecular remodeling and not on secondary risk factors which may also contribute.

Therefore, we think that it is beyond the scope of our paper to include further mechanistic data by including different cell types. Future studies may elucidate the role of AF-induced DNA damage in different cell types in the heart.

4. The underlying logic of the experimental design and the interpretation of the studies are quite confusing at times. For example, the authors assert that DNA damage induces the activation and expression of PARPs which consumes NAD⁺ leading to depleted energy and oxidative stress. Yet, Figure 7 summarizes studies that reverse the logic by showing that PARP inhibition prevents DNA damage and therefore PARP activation, creating a chicken and egg dilemma.

Perhaps, the reviewer is mistaken that PARP1 inhibition totally prevents DNA damage. Inhibition of PARP by ABT-888 prevented tachypacing-induced oxidative protein and DNA damage (Figure 8a-d). In addition, the tachypacing-induced γ H2AX levels were partly reduced by ABT-888 treatment (Figure 8e, f). Together, these data indicate that PARP1 inhibition precludes the initiation of a vicious circle in which advanced PARP1 activation is driven by depletion of NAD⁺, causing oxidation of proteins and DNA and further DNA damage.

To avoid confusion on this topic we amended Results section (Line 317-322, page 11):

‘Inhibition of PARP1 by ABT-888 prevented tachypacing-induced oxidative protein and DNA damage (Figure 8a-d). In addition, the tachypacing-induced γ H2AX levels were partly reduced by ABT-888 treatment (Figure 8e, f). Together, these data indicate that PARP1 inhibition precludes the initiation of a vicious circle in which advanced PARP1 activation is driven by depletion of NAD⁺, causing further DNA damage.’

5. NAD⁺ is largely a mitochondrial compound. It would be helpful to measure NAD⁺ in mitochondria.

Thank you for your comment. In our study, we focused on the role of tachypacing-induced PARP1 activation and its effect on general (mitochondrial and cytoplasmic) NAD⁺ levels and contractile function in atrial cardiomyocytes. Since NAD⁺ is largely present in the mitochondria, tachypacing will probably result in depletion of mitochondrial NAD⁺ pool, resulting in oxidation of cellular proteins as described before (ref 28 and 29 in the manuscript). As we observed oxidative stress in tachypaced HL-1 cardiomyocytes this suggests depletion of NAD⁺ levels from the mitochondria (Figure 8a-d).

We amended the introduction (Line 59, page 3):

‘During the synthesis of PAR chains, nicotinamide adenine dinucleotide (NAD⁺) is consumed by PARP up to an extent that it depletes cellular NAD⁺, leading to a progressive decline in ATP levels, energy loss and cell death in case of excessive PARP activation¹⁴.’

6. Details of how the Ca²⁺ transients were measured and analyzed is required. I am 110% sure that Ca²⁺-transient amplitude vary greatly from cell to cell and from culture to culture. Details of how this inherent variability is addressed are required. These types of studies MUST be done in a blinded manner.

For Ca²⁺ transient measurements of cardiomyocytes, we made movies by using both bright field and fluorescent settings. Bright field settings were used to randomly select normal shaped cardiomyocytes, after which we switched to the fluorescent filter to determine the CaT. Therefore we do not know the CaT amplitude before cardiomyocyte selection. By using this method, we selected in a blinded manner the cardiomyocytes for CaT measurements. In our hands, the CaT amplitudes (measured at 1 Hz) in HL-1 cardiomyocytes and rat atrial cardiomyocytes are comparable between cardiomyocytes within a group.

We added this information to the method section (Line 73-75, page 4):

'Ca²⁺ transient (CaT) measurements were performed according to previous studies^{2, 19}, and in a blinded manner by selection of normal shaped cardiomyocytes with the use of bright field settings, followed by a switch to the fluorescent filter to determine the CaT.'

Reviewers' comments:

Reviewer #1 (Remarks to the Author):

The responses/changes to my comments are acceptable.

Sincerely,

Dr. Csaba Szabo

Dept. of Anesthesiology and Pharmacology

University of Texas

Galveston

Reviewer #2 (Remarks to the Author):

The revisions by Zhang et al were extensive, adding adult rat atrial cardiomyocytes as an additional validation system to the HL-1 cells and Drosophila pupal heart. This reviewer is satisfied with the revisions, except for essential technical issues with the Drosophila heart function experiments.

1. Although commendable that the authors added RARP1 KD to their Drosophila experiments, the genetic rigor is not quite up to par, since only one RNAi line was tested. This deficiency is augmented by the observed substantial reduction in protein and mRNA upon cardiac-only PARP1 KD (S4), which is unexpected given that the extraction was with whole pupae according to the methods. This puts the entire experiment into question. PARP1 is unlikely expressed in just the heart.

2. Although commendable that authors attempted to add an Arrhythmia Index (AI) to their measurements, a more proximal parameter to AF than HR. However, AI in the cited refs (22,23) is not defined as the authors describe, but rather as the standard deviation of the heart period that is normalized to the median of the heart period of each fly - which can be different from fly to fly, thus need to normalize (see also original ref Ocorr et al. 2007 PNAS; Fink et al. 2009). Besides, AI is certainly not a time measurement. This needs to be recalculated, and properly subjected to statistical analysis. Also, how exactly were the traces "used to determine cardiac parameters including heart rate and arrhythmicity index" (120-22, p.5)? By hand? How?

3. The heart wall tracing method (Image J) remains a crude and still ill-defined way of delineating contractions. Was this done by hand? The procedure should be detailed in the methods section. M-modes seem to be much more convincing. It is unclear how some of the traces shown could yield a reliable heart rate, let alone AI. In addition, most of the experiments have a very low n ($n < 10$) and only 10sec per fly. This makes the reliability of these measurements questionable. Thus, the authors should contemplate to put much of them in supplement.
4. Drug feeding regime should be better explained in the methods since pupae do not eat.

Reviewer #3 (Remarks to the Author):

The revised manuscript addresses the points raised in my original critique. The data are novel and of interest, although the lack of experiments in an animal model of Afib is a limitation to understanding the implications of this study. Although the human studies are consistent with the PARP-1 mechanism, they do not provide direct evidence that the PARP-1 pathway modulates the evolution to persistent Afib. Caution with regards to the conclusions would be appropriate, particularly around whether these data implicate the PARP-1 pathway in Afib progression.

Reviewer #4 (Remarks to the Author):

In my previous review I called into question the relevance of "tachypacing" (5Hz) HL-1 cells to atrial fibrillation. Although the authors responded by including data from cultured rat atrial cardiomyocytes, the same issues remain. In particular normal resting heart rate in rats is 5-6 Hz while being ~10Hz in mice (from which HL-1 cells are derived). So I do not understand the logic of pacing the cultured mouse and rat cells at 5Hz. How can this be purported to be representative of the accelerated stimulation rates seen in atrial fibrillation? Mouse atrial cardiomyocytes are electrically stimulated at 20-30 Hz in mouse AF models. The authors completely ignored this (number 1) criticism in their response and revision. It is my opinion that this is a critical fundamental flaw in the logic of the experimental design and calls into question the relevance of these studies to AF.

Additionally, I previously pointed out that AF is ultimately an electrical phenomenon. Electrical measurements in the model systems are still absent. Again, I have doubts that these largely cellular measurements have any relevance to AF.

Response to reviewers' remarks:

We would like to thank the reviewers for their comments and feedback to help us further improve the manuscript. We have adapted the manuscript accordingly as detailed below.

Complete report of Reviewer #1 (Remarks to the Author):

The responses/changes to my comments are acceptable.

We thank this reviewer for the positive comments.

Complete report of Reviewer #2 (Remarks to the Author):

The revisions by Zhang et al were extensive, adding adult rat atrial cardiomyocytes as an additional validation system to the HL-1 cells and Drosophila pupal heart. This reviewer is satisfied with the revisions, except for essential technical issues with the Drosophila heart function experiments.

1. Although commendable that the authors added RARP1 KD to their Drosophila experiments, the genetic rigor is not quite up to par, since only one RNAi line was tested. This deficiency is augmented by the observed substantial reduction in protein and mRNA upon cardiac-only PARP1 KD (S4), which is unexpected given that the extraction was with whole pupae according to the methods. This puts the entire experiment into question. PARP1 is unlikely expressed in just the heart.

2. Although commendable that authors attempted to add an Arrhythmia Index (AI) to their measurements, a more proximal parameter to AF than HR. However, AI in the cited refs (22,23) is not defined as the authors describe, but rather as the standard deviation of the heart period that is normalized to the median of the heart period of each fly - which can be different from fly to fly, thus need to normalize (see also original ref Ocorr et al. 2007 PNAS; Fink et al. 2009). Besides, AI is certainly not a time measurement. This needs to be recalculated, and properly subjected to statistical analysis. Also, how exactly were the traces "used to determine cardiac parameters including heart rate and arrhythmicity index" (120-22, p.5)? By hand? How?

3. The heart wall tracing method (Image J) remains a crude and still ill-defined way of delineating contractions. Was this done by hand? The procedure should be detailed in the methods section. M-modes seem to be much more convincing. It is unclear how some of the traces shown could yield a reliable heart rate, let alone AI. In addition, most of the experiments have a very low n (n<10) and only 10sec per fly. This makes the reliability of these measurements questionable. Thus, the authors should contemplate to put much of them in supplement.

4. Drug feeding regime should be better explained in the methods since pupae do not eat.

Point to point response to remarks of Reviewer #2

The revisions by Zhang et al were extensive, adding adult rat atrial cardiomyocytes as an additional validation system to the HL-1 cells and Drosophila pupal heart. This reviewer is satisfied with the revisions, except for essential technical issues with the Drosophila heart function experiments.

1. Although commendable that the authors added RARP1 KD to their Drosophila experiments, the genetic rigor is not quite up to par, since only one RNAi line was tested. This deficiency is augmented by the observed substantial reduction in protein and mRNA upon cardiac-only PARP1 KD (S4), which is unexpected given that the extraction was with whole pupae according to the

methods. This puts the entire experiment into question. PARP1 is unlikely expressed in just the heart.

In response to your comment, we performed experiments with an additional heart specific PARP1 RNAi line in *Drosophila*. Both the original RNAi1 strain and the novel RNAi2 strain decreased PARP1 protein abundance by ~ 30% in whole prepupae (Figure S4c), a substantially smaller downregulation than found in siRNA treated HL-1 cardiomyocytes (~80%, Figure S4a). These data suggest that the heart contributes to overall PARP1 expression in the *Drosophila* prepupae. Such is not unprecedented, as mitochondrial proteins, proteins involved in glycolysis, redox, and G-protein signalling are also over-represented in the adult *Drosophila* heart (Cammarato A, PLoS One. 2011; 6(4): e18497). Moreover, we additionally explored functional effects in the PARP1RNAi2 line to conclusively validate the role of PARP1 in tachypacing-induced loss of heart wall contraction. Similar to the RNAi1 line, RNAi2 suppression of PARP1 protected against tachypacing-induced contractile dysfunction, including a reduction in heart rate and increase in arrhythmicity index (Figure S5).

Collectively, two heart specific PARP1 RNAi strains induce a similar ~ 30% reduction in PARP1 protein abundance in whole prepupae and protect against tachypacing-induced contractile dysfunction of the heart wall, substantiating the role of PARP1 in heart wall function.

We added information and results the PARP1 RNAi2 line to the supplemental information section Figure S5:

'Figure S5: PARP1 knockdown protects against tachypacing-induced contractile dysfunction in *Drosophila* **a)** Representative M-mode cardiography (left) and corresponding heart wall motion traces (right) of 10 s high-speed movies (100 f/s) of *Drosophila* prepupa. Movies were obtained from non-paced (NP) and tachypaced (TP) *Drosophila* prepupa in wildtype (WT, N=41) and PARP1 knockdown line PARP1 RNAi2 (N=20). **b)** Quantified heart rate (bpm: beats per minute) and **c)** Quantified arrhythmicity index (AI). AI was calculated as the standard deviation of the heart period in ms. **d)** Quantified AI corrected by median of heart period in the same *Drosophila* prepupae as in c). *P<0.05, **P<0.01, ***P<0.001 vs control WT NP, #P<0.05 ###P<0.001 vs WT TP.'

2. Although commendable that authors attempted to add an Arrhythmia Index (AI) to their measurements, a more proximal parameter to AF than HR. However, AI in the cited refs (22,23) is not defined as the authors describe, but rather as the standard deviation of the heart period that is normalized to the median of the heart period of each fly - which can be different from fly to fly, thus need to normalize (see also original ref Ocorr et al. 2007 PNAS; Fink et al. 2009). Besides, AI is certainly not a time measurement. This needs to be recalculated, and properly subjected to statistical analysis. Also, how exactly were the traces "used to determine cardiac parameters including heart rate and arrhythmicity index" (120-22, p.5)? By hand? How?

Arrhythmia Index (AI) is both calculated as the standard deviation of the heart period without (Karen Ocorr et al, 2007, PNAS) or with normalization to the median of the heart period of each fly (Martin Fink, et al 2009, Biotechniques). Since heart period is quantified in milliseconds (ms), the unit of its standard deviation is ms (i.e. as in the originally reported non-normalized AI).

We found that the heart period of *Drosophila* prepupae are highly comparable both in the control situation and after tachypacing. For this reason, we reported AI findings as averaged standard deviation (i.e. without normalization). Nevertheless, to exclude the potential effect of tachypacing on AI, we now additionally include AI with normalization to the median heart period before or after TP. Expectedly, results of AI with and without normalization are highly comparable, supporting our original conclusions. We added these to the manuscript: see Supplemental Information Figure S5c, d and S7f, g.

In addition, to calculate the AI, movies of heart wall measurements were used to prepare heart wall traces and/or M-mode cardiography with the use of Image J, followed by cardiac parameter analysis by Drosan software, which has been modified previously to specifically analyse *Drosophila* heart rate (Ocorr, K. PNAS 2007; Greaves-Lord, K. Psychiatry Res. 2010; Nolte, I. M. Nat. Commun. 2017).

Detailed information on the calculation of cardiac parameters was added to the method section and the algorithm used by the Drosan software is included in the Supplemental Information section.

Line 131-139: 'Heart wall contractions were measured utilizing high-speed digital video imaging (100 frames/s) before and after tachypacing in at least duplicated 10 s-movies. Movies were used to prepare heart wall traces and M-mode cardiography. Hereto, 1-pixel width lines were drawn across the heart wall, followed by determination of Plot-Z axis profile (based on contrast changes) to generate heart wall traces or kymographs (via kymograph plugin of Image J) for M-mode cardiography. To determine the heart rate and arrhythmicity index (AI, defined as the standard deviation of the heart period)²¹, the heart wall traces were further analysed with the use of Drosan software, which was modified from the software originally developed to determine human heart rate and AI^{22, 23}. The detailed algorithm of the Drosan software is described in the supplemental information section.'

Supplemental Information section:

Algorithm of Drosan software

To process the heart tube signal x , the low pass filter is utilized. The used low pass moving average filter of $2N+1$ points is:

$$y_i = \frac{1}{2N+1} \sum_{k=-N}^N x_{i+k}$$

Default setting is $N=5$. The detection of the start of a beat is the maximum detection in the derivative of the filtered signal y , calculated as:

$$dy_i = \frac{1}{2}(y_{i+1} - y_{i-1})$$

Next, find a dy_i with $dy_i > TriggerLevel$ and then find the first i with $dy_i > dy_{i+1}$. Then, i is the sample at the start of the beat. Start the detection algorithm again after $i+n/nh$ samples. n/nh is the number of samples corresponding with the inhibition period (default setting is 200 ms).

The *TriggerLevel* is set to 1.25 SD_y by default, and can be manually adjusted in the user interface. After detection of the beats, valid signal segments are selected (maximal 3 segments). The outcome parameters listed below, are calculated for each segment and for the combined segments.

Variable	Description	Unit
Nbts	Number of beats	-
mIBI	Mean interbeat interval	ms
sdIBI	SDNN or standard deviation of interbeat interval	ms
minIBI	Shortest interbeat interval	ms
maxIBI	Longest interbeat interval	ms
rMSSD	Root mean square of successive differences of interbeat interval	ms
HR	Heart rate ($=60000/mIBI$)	beats/min
medianIBI	Median of IBI values	ms

Note that heart period (HP) is denoted here as Interbeat Interval (IBI).

In this paper, AI is defined as standard deviation of HP (same as sdIBI) as previously reported². Since AI is also defined as standard deviation of the HP normalized by median HP^{3,4}, we also calculated the AI/median. Comparable results were found (Figure S5c, d, S7f, g). Furthermore, we analyzed heart wall movies using SOHA software developed by Cammarato A et al⁴. Both SOHA and Drosan software showed similar results regarding HR and AI (data not shown).

Supplemental References

1. Bingen, B. O. et al. Atrium-specific Kir3.x determines inducibility, dynamics, and termination of fibrillation by regulating restitution-driven alternans. *Circulation* 128, 2732-2744 (2013).
2. Ocorr, K. et al. KCNQ potassium channel mutations cause cardiac arrhythmias in *Drosophila* that mimic the effects of aging. *Proc. Natl. Acad. Sci. U. S. A.* 104, 3943-3948 (2007).
3. Fink, M. et al. A new method for detection and quantification of heartbeat parameters in *Drosophila*, zebrafish, and embryonic mouse hearts. *BioTechniques* 46, 101-113 (2009).
4. Cammarato, A., Ocorr, S. & Ocorr, K. Enhanced assessment of contractile dynamics in *Drosophila* hearts. *BioTechniques* 58, 77-80 (2015).

3. The heart wall tracing method (Image J) remains a crude and still ill-defined way of delineating contractions. Was this done by hand? The procedure should be detailed in the methods section. M-modes seem to be much more convincing. It is unclear how some of the traces shown could yield a reliable heart rate, let alone AI. In addition, most of the experiments have a very low n (n<10) and only 10sec per fly. This makes the reliability of these measurements questionable. Thus, the authors should contemplate to put much of them in supplement.

We compared results obtained via the tracing-method with the M-mode method. Hereto, we generated M-mode kymograph (via ImageJ) for control, tachypaced and tachypaced+olaparib treated *Drosophila*. We found that heart rate and AI findings derived via M-mode kymograph are comparable to findings derived via traces-method. We added these findings to the supplemental information section Figure S5a and S7d-g.

From heart wall movies, we prepared traces by using Image J and Drosan software (Nolte I et al., *Nature Communication* 2017). We added detailed information on the measurement of heart function parameters in the method section and attached the algorithm of Drosan software in the supplemental information section as mentioned above.

In addition, for each prepupae we determined heart wall measurements at least in duplicate series of 10 seconds each. This measurement is based on a study in human showing that a single 10 second (ECG) measurement already results in a reliable AI with the use of Drosan software (Munoz ML et al., *PLoS One*. 2015). Given the number of bpm, i.e. human 10 sec = 6-8 beats and *Drosophila* 20 sec = 40 beats per measurement, the amount of heart beats used for the *Drosophila* measurements is at least 5-fold more compared to the validated 10 sec method in human. In addition, as outlined above (comment 2), the heart rate is very stable in prepupae, therefore a significant effect of a treatment compared to non-treated control prepupae could be reached with at least an N=7 prepupae. We adapted the method section on heart wall measurements to avoid misinterpretation.

Line 131-139: 'Heart wall contractions were measured utilizing high-speed digital video imaging (100 frames/s) before and after tachypacing in at least duplicated 10 s-movies. Movies were used to prepare heart wall traces and M-mode cardiography. Hereto, 1-pixel width lines were drawn across the heart wall, followed by determination of Plot-Z axis profile (based on contrast changes) to generate heart wall traces or kymographs (via kymograph plugin of Image J) for M-mode cardiography. To determine the heart rate and arrhythmicity index (AI, defined as the standard deviation of the heart period)²¹, the heart wall traces were further analysed with the use of Drosan

software, which was modified from the software originally developed to determine human heart rate and AI^{22, 23}. The detailed algorithm of the Drosan software is described in the supplemental information section.'

4. *Drug feeding regime should be better explained in the methods since pupae do not eat.*

The drugs were added to the food containing larvae. The larvae consumed the drug containing food for at least 48 h until they entered the prepupae stage and were used in experiments. We adapted the method section to clarify drug feeding of *Drosophila* prepupae.

Line 117-125:

'Drosophila stocks, tachypacing, and heart wall contraction assays

The wild-type *Drosophila melanogaster* strain w1118 strain was used for all drug screening (PARP inhibitors or NAD⁺) experiments as described before⁸. In short, female and male adult flies were crossed. After 3 days, flies were removed from the embryos-containing tubes and drugs or the same amount of vehicle (DMSO) were added to the food. *Drosophila* were incubated at 25 °C for 48 h, with larvae consuming the drug/vehicle prior to entering the prepupae stage. The *Drosophila* prepupae were collected and subjected to tachypacing for 20 min (4 Hz, 20 V, pulse duration of 5 ms) and heart wall functions were measured as described previously⁸ and in detail below. See Table 1 for the applied doses of 3-AB, ABT-888 and NAD⁺.'

Complete report of Reviewer #3 (Remarks to the Author):

The revised manuscript addresses the points raised in my original critique. The data are novel and of interest, although the lack of experiments in an animal model of Afib is a limitation to understanding the implications of this study. Although the human studies are consistent with the PARP-1 mechanism, they do not provide direct evidence that the PARP-1 pathway modulates the evolution to persistent Afib. Caution with regards to the conclusions would be appropriate, particularly around whether these data implicate the PARP-1 pathway in Afib progression.

To discuss the limitations of the used experimental model systems, including *Drosophila* model in recapitulating human atrial arrhythmogenesis, we added, as suggested by the editor, to the discussion section: Line 464-479

'Limitations of the study

We discovered the role of DNA damage-induced PARP1 activation in cardiomyocyte dysfunction in AF by utilizing various experimental model systems, including HL-1 cardiomyocyte and *Drosophila* models which are easily accessible to genetic manipulations. Although similar observations were made in different experimental AF models (HL-1 cardiomyocyte, rat atrial cardiomyocytes, *Drosophila*) with consistent data from heart tissue from AF patients, our data do not provide conclusive evidence about involvement of PARP1 in AF progression in patients. Nevertheless, previous findings on the role of heat shock proteins, HDAC6 and autophagy, initially made in HL-1 cardiomyocyte and *Drosophila* models have been confirmed in all instances in the tachypaced dog model and clinical human AF^{8, 47, 48}. Therefore, the tachypaced HL-1 cardiomyocyte and *Drosophila* model has merit to identify potential signaling pathways involved in AF remodeling.

Nevertheless, clinical development of PARP1 inhibitors for AF awaits two further steps. First, the action of recently developed PARP1 inhibitors, such as ABT-888, should be investigated in large animal AF models to substantiate its efficacy in relation to the stage of AF. Secondly, current clinical trials should indicate a favourable safety profile, especially in case the animal studies

indicate a beneficial effect of long-term use in halting progression from paroxysmal to persistent AF.'

Complete report of Reviewer #4 (Remarks to the Author):

In my previous review I called into question the relevance of "tachypacing" (5Hz) HL-1 cells to atrial fibrillation. Although the authors responded by including data from cultured rat atrial cardiomyocytes, the same issues remain. In particular normal resting heart rate in rats is 5-6 Hz while being ~10Hz in mice (from which HL-1 cells are derived). So I do not understand the logic of pacing the cultured mouse and rat cells at 5Hz. How can this be purported to be representative of the accelerated stimulation rates seen in atrial fibrillation? Mouse atrial cardiomyocytes are electrically stimulated at 20-30 Hz in mouse AF models. The authors completely ignored this (number 1) criticism in their response and revision. It is my opinion that this is a critical fundamental flaw in the logic of the experimental design and calls into question the relevance of these studies to AF.

Additionally, I previously pointed out that AF is ultimately an electrical phenomenon. Electrical measurements in the model systems are still absent. Again, I have doubts that these largely cellular measurements have any relevance to AF.

Point to point response to remarks of Reviewer #4

1. In my previous review I called into question the relevance of "tachypacing" (5Hz) HL-1 cells to atrial fibrillation. Although the authors responded by including data from cultured rat atrial cardiomyocytes, the same issues remain. In particular normal resting heart rate in rats is 5-6 Hz while being ~10Hz in mice (from which HL-1 cells are derived). So I do not understand the logic of pacing the cultured mouse and rat cells at 5Hz. How can this be purported to be representative of the accelerated stimulation rates seen in atrial fibrillation? Mouse atrial cardiomyocytes are electrically stimulated at 20-30 Hz in mouse AF models. The authors completely ignored this (number 1) criticism in their response and revision. It is my opinion that this is a critical fundamental flaw in the logic of the experimental design and calls into question the relevance of these studies to AF.

To accommodate the original comments of this reviewer, we added to the methods section information on the spontaneous contraction rate of HL-1 and isolated adult rat atrial cardiomyocytes, which is 0.5-1 Hz *in vitro*. Furthermore, we described that *in vitro* both HL-1 cardiomyocytes and rat atrial cardiomyocytes follow a pacing rate up to 5Hz, signifying a 5-10 fold rate increase, which is comparable with the increase in activation rate of cardiomyocytes in clinical AF. As shown by us and several other research groups, this fold increase in activation rate was successfully utilized to generate *in vitro* experimental models systems to study tachycardia-induced cardiomyocyte remodeling (L.C. Mace et al., J. Mol. Cell. Cardiol 2009; Y.H. Yeh et al., Cardiovasc. Res 2011; N. Kim et al., Sci. Rep. 2017).

We rephrased parts of the methods section to reflect the above. Line: 75-78.

'The cardiomyocytes, which have a basal spontaneous contraction rate of ~ 0.5-1 Hz⁴, were subjected to a 5-10 fold rate increase as observed in clinical AF tachypaced (TP, 5 Hz, 40 V, pulse duration of 20 ms) with a C-Pace100 culture pacer (IonOptix) for 12 h except specifically stated for time-course pacing.'

Line: 104-105

'The isolated adult rat atrial cardiomyocytes have *in vitro* a basal spontaneous contraction rate of ~ 0.5-1 Hz.'

2. Additionally, I previously pointed out that AF is ultimately an electrical phenomenon. Electrical measurements in the model systems are still absent. Again, I have doubts that these largely cellular measurements have any relevance to AF.

Whereas parameters for structural remodeling are an excellent proxy for electrophysiological changes (De Groot N et al., *Circ* 2010; Brundel B et al., *Circ* 2001, *Circ Res* 2006; Ausma J et al., *Circ* 1998, Zhang D et al., *Circ* 2014; Wiersma M et al., *JAHA* 2017), we performed electrophysiological measurements as requested. Hereto, electrophysiological parameters were obtained by optical voltage mapping in control and tachypaced HL-1 cardiomyocytes with and without olaparib or ABT888 treatment, including action potential duration, dispersion of action potential duration and excitable cell surface areas. Expectedly, tachypacing induced changes in electrical signals comparable to changes observed in big animal models for AF due to structural remodelling (Li D et al., *Circulation* 1999; S Verheule S et al., *Circulation* 2003; Cha, YM et al., *Am. J. Physiol. Heart Circ. Physiol.* 2003) and clinical AF (B Brundel et al, *Circulation* 2001) which were precluded by both PARP1 inhibitors.

We added the findings to the supplemental Methods section and Figure S11 and changed the discussion section.

'Supplemental methods

Optical voltage mapping in HL-1 cardiomyocytes

Action potential (AP) generation was investigated on monolayer cultures of HL-1 cardiomyocytes by optical voltage mapping using di-4-ANEPPS (Thermo Fisher Scientific) as fluorescent voltage indicator as described previously¹. In short, HL-1 cardiomyocytes were seeded in 24-well cell culture plates on bovine fibronectin-coated round glass coverslips (diameter of 15 mm). Confluent monolayers of HL-1 cardiomyocytes were incubated for 12 h in medium containing 20 μ M olaparib, 40 μ M ABT-888 or vehicle DMSO, followed by non-pacing or tachypacing (5 Hz, 40 V, 20 ms) for 8 h in the continued presence of treatment. Optical signals were captured using a MiCAM ULTIMA-L imaging system (SciMedia) and analysed using Brain Vision Analyzer 1208 software (Brainvision). Noise artefacts were minimized by averaging of the signals at a selected pixel and its eight nearest neighbours. AP duration (APD) at 30% and 80% repolarization (APD₃₀ and APD₈₀, respectively), and APD dispersions at different percentages of repolarization were determined using HL-1 cultures showing full capture after 1-Hz electrical stimulation via a STG 2004 stimulus generator and MC Stimulus II software (both from Multi Channel Systems). Quantitative analyses of excited area of each monolayer culture were performed with the Java-based image processing program Image J (version 1.50i, National Institutes of Health).

Figure S11: PARP1 inhibitors significantly attenuated tachypacing-induced electrophysiological deterioration in HL-1 cardiomyocytes. a-h) Optical voltage mapping of HL-1 cardiomyocyte monolayers following 1-Hz electrical stimulation in control non-paced (NP) or 8 h tachypaced (TP) HL-1 cardiomyocyte cultures with 20 μ M olaparib, 40 μ M ABT-888 or vehicle DMSO pretreatment 12 h before tachypacing. **a)** Representative filtered optical signal traces, **b)** typical APD₃₀ and **c)** APD₈₀ maps for indicated groups. **d-h)** Corresponding quantitative analysis of APD₃₀, APD₈₀, APD₃₀ dispersion, APD₈₀ dispersion and excited cell surface area, showing that TP resulted in significant APD prolongation (**a**, **d** and **e**), an increase in APD dispersion (**b**, **c**, **f** and **g**) and a significant decrease of excited cell surface area (**h**) in HL-1 cardiomyocyte monolayers. Pretreatment of HL-1 cultures with ABT-888 or olaparib significantly prevented the tachypacing-induced electrophysiological deteriorations (**a-h**). *** $P < 0.001$ vs DMSO NP, ### $P < 0.001$ vs DMSO TP. N=11 for NP DMSO, N=9 for NP olaparib TP, N=11 for NP ABT-888, N=6 for TP DMSO, N=6 for TP olaparib, N=6 for TP+ABT-888. N= number of experiments.

Discussion section line 403-415:

'Our data in tachypaced atrial cardiomyocytes imply that in (longstanding) persistent AF, the excessive activation of PARP1 and depletion of cellular NAD⁺, a key coenzyme in cell metabolism²⁸, induce further DNA damage, which ultimately results in structural damage, and consequently electrophysiological deterioration and functional loss¹². Our finding is consistent with previous findings showing that structural remodeling-induced electrophysiological deterioration, including prolongation of APD³⁶⁻³⁹, reduction in cardiomyocyte excitability and increased ADP dispersion, creates a substrate for further arrhythmogenesis^{40, 41}. Although APD shortening was previously recorded in models for lone AF, APD prolongation was observed in various patient and animal studies for AF with underlying structural remodeling³⁶⁻³⁹, which is consistent with our current finding. As such, PARP1-induced depletion of NAD⁺ apparently functions as a key feed-forward switch in this chain of events, as PARP1 inhibition fully conserves NAD⁺ levels, precludes oxidative protein and DNA damage and preserves structural and contractile function in tachypaced atrial cardiomyocytes.'

Reviewers' comments:

Reviewer #2 (Remarks to the Author):

The revisions made in this second round are satisfactory for this reviewer.

One point to consider for the authors is that AI (based on SOHA) is now commonly used as standard deviation of the heart period of each fly normalized to the median heart period of each fly and then averaged across flies. This AI normalization method should be specified in the methods section and used in the main figures. The reason is that the median heart period can vary from fly to fly and has nothing to do with the actual arrhythmias to be calculated.

Reviewer #3 (Remarks to the Author):

The additional optical voltage mapping data in supplemental figure 11 do show significant electrophysiological impact of tachy-pacing in HL-1 cells, that is relieved by PARP inhibitors.

However, the reported APD prolongation is atypical for large animal and human AF, which is usually characterized by shortening of the in vivo atrial effective refractory period. The manuscript cites literature showing prolongation of the atrial effective refractory period in Afib in the setting of heart failure with fibrotic remodeling in vivo, but the direct relevance of the HL-1 cell autonomous electrophysiological changes observed in vitro to potential in vivo Afib mechanisms is uncertain. Perhaps they have specific relevance to Afib in the setting of heart failure, but the text does not address this important point.

The quality of the optical action potential data is also an issue, in so far as the manuscript lacks analysis of the cellular depolarization velocity, which appears to be slow and is not analyzed or commented upon. The amplitude of the APD also appears diminished in the selected 3 tracings, but it is not clear how these cells were selected and the data are not quantified. I am less familiar with the relevance of cellular APD80 dispersion and excited area to AF in vivo, but at best the relationship to Afib is indirect.

The lack of any molecular correlate of the electrophysiological changes is another important limitation to the data presented, i.e. whether the APD prolongation is associated with changes in K⁺ or Ca⁺⁺ channel expression (or activity). Assessment of channel transcripts would be reasonable to request.

Finally, it is disappointing that the new electrophysiological data, which are an important aspect of the study, are presented in a supplemental figure. The authors should firm up the electrophysiological data and place it in a main figure.

Finally, the new data are striking and support the importance of the PARP pathway during pacing stress in atrial cells, but they also raise additional questions about the relevance to clinical atrial fibrillation, and with only a limited supplemental figure dilute the quality of the manuscript.

The PARP mechanism of atrial cellular dysfunction remains of interest. However, firming up the electrophysiological data and a thorough revision of the text are necessary.

Unfortunately, the text still very much over-interprets the relevance of the cellular studies to Afib. These are simply cell pacing studies and all mention of the implications for Afib should be reserved for the discussion section, where an appropriately balanced consideration will allow the reader to evaluate the relevance of the findings to human disease.

Reviewer #4 (Remarks to the Author):

My criticisms of the manuscript remain. The fundamental problem is that the authors have failed to link rapid pacing in their model systems to atrial fibrillation. The definition of rapid pacing does not apply to the cellular systems; the HL-1 cells are paced at 50% of the normal beating rates of mouse hearts while the cultured rat cardiomyocytes are paced at rates similar to the heart rates of rats. How can the authors claim this is rapid pacing that is somehow representing what occurs in human AF, where beating rates of atria go to 3-5 times baseline rates. Also the interventions are so acute, it is hard to understand that such short periods of pacing are relevant and this also applies to the flies. I am sorry but I just do not get it. Moreover, there is no detailed electrical analyses to determine what electrical changes are occurring. In fact, the optical mapping data shows APD30 prolongation, the opposite of what has been observed in paced dogs and goats (which are rapidly paced).

Reviewers' comments:

Reviewer #2 (Remarks to the Author):

The revisions made in this second round are satisfactory for this reviewer.

One point to consider for the authors is that AI (based on SOHA) is now commonly used as standard deviation of the heart period of each fly normalized to the median heart period of each fly and then averaged across flies. This AI normalization method should be specified in the methods section and used in the main figures. The reason is that the median heart period can vary from fly to fly and has nothing to do with the actual arrhythmias to be calculated.

We like to thank this reviewer for the positive comment related to the revisions made to the manuscript and the suggestion regarding the arrhythmicity index (AI) normalization method to further improve our manuscript.

Indeed, we agree that AI is more commonly defined as the standard deviation of the heart period of each subject normalized to its median heart period with subsequent averaging across (Fink M, BioTechniques 46, 101-113 (2009), ref 21 in the revised manuscript). Therefore, we recalculated AI according to this definition and amended the figures 3e, 4e and 6g as well as related supplemental figures (originally fig S5d and S7g, now figS5c and S7g). Moreover, we outlined the revised calculation of AI in the method section, see Page 5, Line 131-135. We removed text describing AI calculations based on standard deviation in the supplemental methods section (lines 814-816).

Reviewer #3 (Remarks to the Author):

The additional optical voltage mapping data in supplemental figure 11 do show significant electrophysiological impact of tachy-pacing in HL-1 cells, that is relieved by PARP inhibitors.

However, the reported APD prolongation is atypical for large animal and human AF, which is usually characterized by shortening of the in vivo atrial effective refractory period. The manuscript cites literature showing prolongation of the atrial effective refractory period in Afib in the setting of heart failure with fibrotic remodeling in vivo, but the direct relevance of the HL-1 cell autonomous electrophysiological changes observed in vitro to potential in vivo Afib mechanisms is uncertain. Perhaps they have specific relevance to Afib in the setting of heart failure, but the text does not address this important point.

Accumulating evidence from clinical and animal studies provide compelling evidence that the predominant contributors underlying AF are the structural and associated conduction abnormalities rather than changes in refractoriness. This is underscored by various studies showing shortening in APD and also prolongation of APD in AF, the latter likely via reduction in potassium channel function (Olson T, Human Molecular Genetics 2006). Prolongation has been typically observed in patients with 'lone' paroxysmal AF (Allessie M, JACC 2009 and Stiles MK, JACC 2009), but also AF in combination with heart failure and structural changes in the atria. For example, in a canine model of congestive heart failure, an increase in ERP was observed which correlated with structural remodeling and a significant increase in the

duration of AF (Li D, Circ 1999). Also, Verheule et al. observed a propensity for AF despite an increase in ERP in a canine model of mitral regurgitation, again due to abnormalities in conduction and profound structural remodeling (Verheule S, Circ 2003). Clinical studies of the atrial substrate in patients known to be predisposed to AF, but without antecedent arrhythmia, have demonstrated similar prolonged AERP (Morton JB, Circ 2003; Sanders P, Circ 2003, Circ 2004; Kistler P, JACC 2004). Taken together, these clinical and animal experimental studies indicate that the predominant contributors underlying AF are the structural and associated conduction abnormalities rather than changes in refractoriness. Importantly, these observations are in line with the findings from our study and may explain why drug treatment directed at refractoriness shows limited efficacy in AF patients.

To address this important issue, we added to the discussion section (Lines 404-416):

‘Our finding is consistent with previous findings showing that structural remodeling underlies electrophysiological deterioration, including prolongation of APD (possibly via the reduction in potassium channel)³⁶⁻⁴⁰, reduction in cardiomyocyte excitability and increased ADP dispersion, thereby creating a substrate for further arrhythmogenesis⁴¹⁻⁴⁴. Although APD shortening was previously recorded in models for tachypacing-induced AF, APD prolongation was observed in patients with lone paroxysmal AF, in atrial tissue of patients predisposed to AF and in various patient and animal studies for AF with underlying heart failure and structural changes in the atria^{36-39, 45, 46}, which is consistent with our current findings. Taken together, these studies provide compelling evidence that the predominant contributors to the substrate underlying AF are the structural and associated conduction abnormalities rather than changes in refractoriness. In addition, the studies may explain why current drug treatment directed at modulation of refractoriness shows limited efficacy, while its usage is further limited by a pro-arrhythmic action and non-cardiovascular toxicity⁴⁷.’

The quality of the optical action potential data is also an issue, in so far as the manuscript lacks analysis of the cellular depolarization velocity, which appears to be slow and is not analyzed or commented upon. The amplitude of the APD also appears diminished in the selected 3 tracings, but it is not clear how these cells were selected and the data are not quantified. I am less familiar with the relevance of cellular APD80 dispersion and excited area to AF in vivo, but at best the relationship to Afib is indirect.

The reviewer is right about the absence of analysis of cellular depolarization velocity in cells. Our current setup only allows for measurement of depolarization velocity within the monolayer, rather than on the single cell level. Moreover, tachypacing induced a complete paucity in fluorescent signals in areas dispersed across the cell culture, indicating the excessive reduction or even absence of action potential generation. This interfered with proper assessment of depolarization velocity [dF/dt max], as such analysis is critically dependent on excited areas large enough for assessment. Consequently, the electrophysiological consequences of tachypacing prevented such assessment. For a more technical explanation of how conduction velocity is measured from optical mapping data, see Askar et al. Cardiovasc Res, 2011. We describe this item in the revised manuscript supplemental methods section.

Supplemental method section (Page 36, Lines 793-795):

'Tachypacing induced paucity in fluorescent signals in dispersed areas across the cell culture, indicating excessive reduction or even absence of action potential generation, precluding proper assessment of depolarization velocities in the cell monolayer².'

The reviewer is correct about the lower signal amplitudes in control tachypaced HL-1 cardiomyocytes. Such may have been caused both by effects of tachypacing and/or differences in initial loading conditions of the voltage-sensitive dye. For this reason, we choose to measure the effect of tachypacing on electrical function by means of APD maps in monolayers of HL-1 cardiomyocytes. These APD maps reveal that tachypacing increases heterogeneity in APD which was prevented by treatment of PARP inhibitors (ABT888, olaparib).

The selection of 3 traces, as shown in Figure 7, aimed to highlight the electrical heterogeneity due to tachypacing, and is based on the differences in electrical signal observed within a typical example (i.e. ranging from signals varying in time and space [1 and 3] to excitation block [2]), as we address in the revised manuscript (page 26).

Legend section figure 7.

' a) Representative filtered optical signal traces. To indicate electrical heterogeneity, three tracers which vary in time and space [1 and 3] to excitation block [2] in the TP DMSO group are depicted.'

The relevance of APD80 dispersion and excited area to AF is based on their role in the initiation of AF (Andrade, Circ Res 2014 and Bingen, Circ 2013). In these *in vivo* studies it was shown, that heterogeneity in APD contributes to conduction abnormalities, including wave block and break, thereby increasing the probability for disturbances in electrical propagation to occur, including reentry as driver of AF. In a similar way, areas of conduction block are known to create conditions for initiation of AF and even add to its maintenance once established.

We adapted the discussion section on page 13 (line 403-407).

'Our finding is consistent with previous findings showing that structural remodeling underlies electrophysiological deterioration, including prolongation of APD (possibly via the reduction in potassium channel)³⁶⁻⁴⁰, reduction in cardiomyocyte excitability and increased ADP dispersion, thereby creating a substrate for further arrhythmogenesis⁴¹⁻⁴⁴.'

The lack of any molecular correlate of the electrophysiological changes is another important limitation to the data presented, i.e. whether the APD prolongation is associated with changes in K⁺ or Ca⁺⁺ channel expression (or activity). Assessment of channel transcripts would be reasonable to request.

To get more insight whether the electrophysiological changes are associated with changes in ion channel expression, we measured the transcript levels of the α -subunits of L-type Ca²⁺ channel (CNCA1C), and three potassium channels (I_{Ks} KCNQ1, I_{Kr} KCNH2 and I_{KACH} KCNJ3) by qPCR (Figure S11a-d) and confirmed the findings for L-type Ca²⁺ channel (Cav 1.2), and two potassium channels (I_{Kr} Kv11.1 and Kir3.1) by Western blot analyses (Figure S11e-h). Tachypacing of HL-1 cardiomyocytes resulted in a reduction in mRNA and protein level of all ion channel subunits (Figure S11). Importantly, treatment with the PARP1 inhibitor ABT888

precluded the tachypacing induced mRNA reduction of all potassium and L-type Ca²⁺ channel (CNCA1C) subunits (Figure S11). Collectively, these results indicate that inhibition of PARP activity prevents tachypacing-induced ion channel remodeling. Since lower potassium channel expression level explains the prolongation of the APD, protection from its reduction by ABT888 may conserve APD and therefore suppress electrophysiological changes.

We added antibody and primer information to the methods section (Page 6 and 7).

Result section (Page 11, Line 315-319):

'In addition, tachypacing of HL-1 cardiomyocytes resulted in significant electrophysiological deteriorations, including alterations in action potential duration (APD), increased APD dispersions, decreased area of excitability and ion channel remodeling. All tachypacing-induced electrophysiological alterations were prevented by PARP1 inhibitors olaparib and/or ABT-888 (Figure 7 and supplemental methods section and Figure S11).'

Discussion section (Page 13, Line 399-404)

'Our data from in tachypaced atrial cardiomyocytes reveal that excessive activation of PARP1 and depletion of cellular NAD⁺, a key coenzyme in cell metabolism²⁸, induce further DNA damage, and structural damage, and consequently electrophysiological and ion channel deterioration and functional loss¹². These findings offer a novel paradigm to be tested in (longstanding) persistent AF patients.'

We adapted parts of the abstract and discussion section to avoid over-interpretations of the atrial cardiomyocyte findings.

Finally, it is disappointing that the new electrophysiological data, which are an important aspect of the study, are presented in a supplemental figure. The authors should firm up the electrophysiological data and place it in a main figure.

We firmed up the electrophysiological data as mentioned above and placed the electrophysiological data in the main figures (Figure 7). We also detected changes in expression of main ion channels and added it as an extra supplemental figure (Figure S11) to the manuscript.

Finally, the new data are striking and support the importance of the PARP pathway during pacing stress in atrial cells, but they also raise additional questions about the relevance to clinical atrial fibrillation, and with only a limited supplemental figure dilute the quality of the manuscript.

The PARP mechanism of atrial cellular dysfunction remains of interest. However, firming up the electrophysiological data and a thorough revision of the text are necessary.

We believe that the adaptations outlined above substantially firmed up the electrophysiological data by relating it to alterations in expression of ion channels, thus providing additional evidence for a crucial role of PARP in tachypacing-induced cardiomyocyte remodeling.

Unfortunately, the text still very much over-interprets the relevance of the cellular studies to Afib. These are simply cell pacing studies and all mention of the implications for Afib should be reserved for the discussion section, where an appropriately balanced consideration will allow the reader to evaluate the relevance of the findings to human disease.

To balance the relevance of our findings in experimental model system to clinical AF we adapted the abstract and discussion section, including the limitation section (Page 15):

‘We uncovered a role for DNA damage-induced PARP1 activation in cardiomyocyte dysfunction in AF by utilizing various experimental model systems, including tachypaced HL-1 cardiomyocyte and *Drosophila* models which are easily accessible to genetic manipulations. The spontaneous contraction rate of these cardiomyocytes is ~ 0.5-1 Hz in a 2D culture dish (instead of 5-7 Hz in *in vivo* mice/rats), a 5-10 fold rate increase by tachypacing induces various endpoints of human AF remodeling^{8, 19, 59}. Although observations were consistent between different experimental AF models (*in vitro* HL-1 cardiomyocyte and rat atrial cardiomyocytes, *Drosophila*) and in heart tissue from AF patients, our data do not provide conclusive evidence about involvement of PARP1 in AF progression in patients. Nevertheless, previous findings on the role of heat shock proteins, HDAC6 and autophagy, initially made in HL-1 cardiomyocyte and *Drosophila* models have been confirmed in all instances in the tachypaced dog model and clinical human AF^{8, 19, 59}. Therefore, the tachypaced HL-1 cardiomyocyte and *Drosophila* model may have merit to identify potential signaling pathways involved in AF remodeling. Future research should elucidate the relevance of the DNA damage-induced PARP1 activation pathway in clinical AF with or without underlying heart diseases.’

Reviewer #4 (Remarks to the Author):

My criticisms of the manuscript remain. The fundamental problem is that the authors have failed to link rapid pacing in their model systems to atrial fibrillation. The definition of rapid pacing does not apply to the cellular systems; the HL-1 cells are paced at 50% of the normal beating rates of mouse hearts while the cultures rat cardiomyocytes are paced at rates similar to the heart rats of rats. How can the authors claim this is rapid pacing that is somehow representing what occurs in human AF, where beating rates of atria go to 3-5 times baseline rates. Also the interventions are so acute, it is hard to understand that such short periods of pacing are relevant and this also applies to the flies. I am sorry but I just do not get it. Moreover, there is no detailed electrical analyses to determine what electrical changes are occurring. In fact, the optical mapping data shows APD30 prolongation, the opposite of what has been observed in paced dogs and goats (which are rapidly paced).

The purpose of the work using model systems is to identify fundamental cellular adaptations to rapid pacing, not so to say “to establish AF in a dish”. In the current approach, we took advantage of the ease of genetic and pharmacological manipulations in these systems to robustly delineate druggable targets. In analyzing material from patients, we present evidence that such changes are in fact observed in human AF. Evidently,

whether interventions in such pathways will ultimately benefit AF patients awaits further (clinical) studies.

To address that the discrepancies between used tachypacing frequencies *in vitro* and the atrial rates observed *in vivo* during atrial fibrillation we added to the discussion section (limitations of the study):

‘We uncovered a role for DNA damage-induced PARP1 activation in cardiomyocyte dysfunction in AF by utilizing various experimental model systems, including tachypaced HL-1 cardiomyocyte and *Drosophila* models which are easily accessible to genetic manipulations. The spontaneous contraction rate of these cardiomyocytes is ~ 0.5-1 Hz in a 2D culture dish (instead of 5-7 Hz in *in vivo* mice/rats), a 5-10 fold rate increase by tachypacing induces various endpoints of human AF remodeling^{8, 19, 59}. Although observations were consistent between different experimental AF models (*in vitro* HL-1 cardiomyocyte and rat atrial cardiomyocytes, *Drosophila*) and in heart tissue from AF patients, our data do not provide conclusive evidence about involvement of PARP1 in AF progression in patients. Nevertheless, previous findings on the role of heat shock proteins, HDAC6 and autophagy, initially made in HL-1 cardiomyocyte and *Drosophila* models have been confirmed in all instances in the tachypaced dog model and clinical human AF^{8, 19, 59}. Therefore, the tachypaced HL-1 cardiomyocyte and *Drosophila* model may have merit to identify potential signaling pathways involved in AF remodeling. Future research should elucidate the relevance of the DNA damage-induced PARP1 activation pathway in clinical AF with or without underlying heart diseases.’

Furthermore, we obtained more insight whether the electrophysiological changes are associated with changes in ion channel expression, as mentioned above. Hereto, we measured the transcript levels of the α -subunits of L-type Ca^{2+} channel (CNCA1C), and three potassium channels (I_{Ks} KCNQ1, I_{Kr} KCNH2 and I_{KACH} KCNJ3) by qPCR (Figure S11a-d) and confirmed the findings for L-type Ca^{2+} channel (Cav 1.2), and two potassium channels (I_{Kr} Kv11.1 and I_{KACH} Kir3.1) by Western blot analyses (Figure S11e-h). Tachypacing of HL-1 cardiomyocytes resulted in a reduction in mRNA and protein level of all ion channel subunits (Figure S11). Importantly, treatment with the PARP1 inhibitor ABT888 precluded the tachypacing induced mRNA reduction of all three potassium and L-type Ca^{2+} channel (CNCA1C) subunits (Figure S11). Collectively, these results indicate that inhibition of PARP activity prevents tachypacing-induced ion channel remodeling. Since lower potassium channel expression level explains the prolongation of the APD, protection from its reduction by ABT888 may conserve APD and therefore suppress electrophysiological changes.

We added antibody and primer information to the methods section (Page 6 and 7).

Result section (Page 11, Line 315-319):

‘In addition, tachypacing of HL-1 cardiomyocytes resulted in significant electrophysiological deteriorations, including alterations in action potential duration (APD), increased APD dispersions, decreased area of excitability and ion channel remodeling. All tachypacing-induced electrophysiological alterations were prevented by PARP1 inhibitors olaparib and/or ABT-888 (Figure 7 and supplemental methods section and Figure S11).’

Discussion section (Page 13, Line 399-404):

'Our data from tachypaced atrial cardiomyocytes reveal that excessive activation of PARP1 and depletion of cellular NAD⁺, a key coenzyme in cell metabolism²⁸, induce further DNA damage, and structural damage, and consequently electrophysiological and ion channel deterioration and functional loss¹². These findings offer a novel paradigm to be tested in (longstanding) persistent AF patients¹².'

REVIEWERS' COMMENTS:

Reviewer #3 (Remarks to the Author):

The revised manuscript has addressed the prior critique with additional data on ion channel expressions the addition of a limitations paragraph to the discussion.

However, the discussion should more precisely distinguish findings in the experimental models from those based on human atrial tissue analysis in patients with AF. Although the data from patients support the hypothesis that PARP activation occurs in the setting of DNA damage in AF, they do not assess NAD or energy content in AF. Yet the discussion includes the following paragraphs:

"In the current study, we identified PARP1 activation as a key process in AF by conferring depletion of the cellular content of NAD⁺, an important component for cell function.

"We found PARP, specifically PARP1, to have a prominent role in AF progression. Both in tachypaced atrial cardiomyocytes and RAA/LAA tissue from persistent AF patients, we observed that PARP1 activation is caused by DNA damage and results in the consumption of NAD⁺ to such an extent that it depletes intracellular NAD⁺ levels, thereby exacerbating oxidative damage to proteins and DNA. Activation of this sequel is likely triggered by a substantially increase in myocardial energy demand resulting from the four- to six-fold increase in electrical and contractile activity during AF episodes."

I would suggest careful proof-reading to confirm a more precise summary and conclusion.

Reviewers' comments:

Reviewer #3 (Remarks to the Author):

The revised manuscript has addressed the prior critique with additional data on ion channel expressions the addition of a limitations paragraph to the discussion.

However, the discussion should more precisely distinguish findings in the experimental models from those based on human atrial tissue analysis in patients with AF. Although the data from patients support the hypothesis that PARP activation occurs in the setting of DNA damage in AF, they do not assess NAD or energy content in AF. Yet the discussion includes the following paragraphs:

"In the current study, we identified PARP1 activation as a key process in AF by conferring depletion of the cellular content of NAD⁺, an important component for cell function.

"We found PARP, specifically PARP1, to have a prominent role in AF progression. Both in tachypaced atrial cardiomyocytes and RAA/LAA tissue from persistent AF patients, we observed that PARP1 activation is caused by DNA damage and results in the consumption of NAD⁺ to such an extent that it depletes intracellular NAD⁺ levels, thereby exacerbating oxidative damage to proteins and DNA. Activation of this sequel is likely triggered by a substantially increase in myocardial energy demand resulting from the four- to six-fold increase in electrical and contractile activity during AF episodes."

I would suggest careful proof-reading to confirm a more precise summary and conclusion.

To align with the observations in AF patients, we changed parts of the discussion as suggested by this reviewer.

Page 7

'In the current study, we identified PARP1 activation as a key process in experimental AF by conferring depletion of the cellular content of NAD⁺, an important component for cell function.'

Page 7

'We found PARP, specifically PARP1, to have a prominent role in AF progression. Both in tachypaced atrial cardiomyocytes and RAA/LAA tissue from persistent AF patients, we observed that PARP1 activation is caused by DNA damage. Moreover, in tachypaced atrial cardiomyocytes we showed that PARP1 activation results in the consumption of NAD⁺ to such an extent that it depletes intracellular NAD⁺ levels, thereby exacerbating oxidative damage to proteins and DNA. Activation of this sequel is likely triggered by a substantially increase in myocardial energy demand resulting from the four- to six-fold increase in electrical and contractile activity during AF episodes.'